# Elevated plasma complement factor H related 5 protein is associated with venous thromboembolism

Venous thromboembolism (VTE) is a common, multi-causal disease with potentially serious short- and long-term complications. In clinical practice, there is a need for improved plasma biomarker-based tools for VTE diagnosis and risk prediction. Here we show, using proteomics profiling to screen plasma from patients with suspected acute VTE, and several case-control studies for VTE, how Complement Factor H Related 5 protein (CFHR5), a regulator of the alternative pathway of complement activation, is a VTE-associated plasma biomarker. In plasma, higher CFHR5 levels are associated with increased thrombin generation potential and recombinant CFHR5 enhanced platelet activation in vitro. GWAS analysis of ~52,000 participants identifies six loci associated with CFHR5 plasma levels, but Mendelian randomization do not demonstrate causality between CFHR5 and VTE. Our results indicate an important role for the regulation of the alternative pathway of complement activation in VTE and that CFHR5 represents a potential diagnostic and/or risk predictive plasma biomarker.

Venous thromboembolism (VTE), comprising both pulmonary embolism (PE) and deep vein thrombosis (DVT) is a common, multi-causal disease with serious short and long-term complications. VTE has a high mortality rate in the first year, especially within the first 30 days (~30% for PE) and a high risk of recurrence, with a cumulative incidence rate of 25% within 10 years[1–4]. VTE diagnosis is challenging, as the predisposing common risk factors and clinical presentation can be consistent with multiple other conditions, particularly in the case of PE. Current VTE diagnostic workup includes assessment of clinical probability, using clinical decision rules, e.g., the Wells score, in combination with elevated plasma D-dimer levels[5,6]. D-dimer, a clot breakdown product, can be elevated in several other non-VTE conditions (e.g., inflammation, surgery, cancer), and its usefulness is limited to ruling out VTE in low probability cases. In medium and high probability cases, diagnostic imaging is necessary to exclude or confirm diagnosis. However, despite progress in imaging techniques there is an urgent need for more precise biomarker-based tools to confirm or exclude VTE in a hospital setting. Several studies have proposed VTE biomarker candidates (e.g., p-selectin, microvesicles)[7], but none have reached clinical implementation. There is also a need for improved tools and

plasma biomarkers for risk prediction in VTE. Risk scores based on clinical risk factors and D-dimer levels have been developed for recurrence prediction[8–14], but none are routinely integrated into clinical practice. In addition to the genetic variants currently used in clinical assessment of hereditary thrombophilia (e.g., Factor V Leiden, prothrombin mutation) there are other more recently discovered common gene variants that contribute to VTE risk[15–17]. However, even when these are also incorporated into risk scores, they still lack sufficient precision for VTE prediction on an individual basis[18,19]. This likely reflects the interplay between transient and sustained risk factors in disease development, including acquired risk factors, genetics, and environmental exposures[20,21].

VTE is a disease of the intravascular compartment and thus analysis of the blood proteome has the potential to capture the resulting effects of combined genetic, epi-genetic, and environmental contributors to risk variation. So far, a handful of plasma proteomics studies of VTE have been reported, presenting new candidates associated with increased risk of VTE[22–27].

In this work, we aimed to identify biomarkers associated with acute VTE with a potential link to underlying VTE pathogenesis. We identify

✉ e-mail: david-alexandre.tregouet@u-bordeaux.fr; jacob1@kth.se

Complement Factor H Related 5 protein (CFHR5), a regulator of the alternative pathway of complement activation, as a VTE associated plasma biomarker. Our study suggests that CFHR5 could be involved in the underlying pathogenesis of VTE, and that it is a potential clinical biomarker for thrombotic disease diagnosis and/or risk prediction.

## Results

### Affinity plasma proteomics identifies candidates associated with acute VTE

To identify plasma biomarkers for VTE we analysed samples collected as part of our Venous thromboEmbolism BIOmarker Study (VEBIOS)[23]. VEBIOS comprises two study arms; a prospective cohort of patients sampled at the Emergency Room (ER), Karolinska University Hospital, Sweden (VEBIOS ER) and a case/control study with patients sampled at an outpatient coagulation clinic after discontinuation of anticoagulant treatment after a first VTE event (VEBIOS Coagulation). The discovery cohort, VEBIOS ER (Fig. 1a), consisted of patients ($n = 147$) admitted to the ER with the suspicion of DVT in the lower limbs and/or PE. Following admission, both citrate and EDTA whole blood samples were collected from participants. Patient samples were classified as controls ($n = 96$), when a VTE diagnosis was excluded by diagnostic imaging, and/or Wells clinical criteria with a normal D-dimer test, or cases ($n = 51$) when VTE was confirmed by diagnostic imaging and anticoagulant treatment was initiated. A nested case/control sample set of 48 cases and 48 matched controls were selected for plasma protein analysis (Table 1). Target candidates for measurement were selected as previously described[23], based on: (i) indications from the literature, in house data or public repositories of a probable or plausible link to arterial or venous thrombosis (e.g., prior evidence of association with thrombosis or intermediate traits, or known involvement in biological pathways of relevance), including 124 that we predicted to have endothelial enriched expression[28], and (ii) the availability of target specific antibodies in the Human Protein Atlas (HPA) (see Methods). A total of 756 HPA antibodies, targeting 408 candidate proteins, were selected for incorporation into a single-binder suspension bead array (Supplementary information Fig. S1a and Supplementary data 1 [Tab_1]), which was used to analyse plasma generated from the blood samples collected into citrate anticoagulant. The signal generated by antibody HPA059937, raised against the protein target sulfatase 1 (SULF1), was most strongly associated with VTE ($p < 8.34E-06$) (Fig. 1b, green point), with higher relative plasma levels in cases vs. controls (Fig. 1d; left). Signals generated by a further seven antibodies were also associated with VTE ($p < 0.01$) (Supplementary data 1 [Tab_1]). Protein signatures in plasma can be differently affected by the sample matrix; anticoagulants can inhibit specific proteases, influence soluble protein interactions, and modify analyte stability. Thus, the anticoagulant type used has potential consequences for biomarker identification[29]. We therefore replicated the VEBIOS ER discovery screen in EDTA samples drawn in parallel from the same patients (Fig. 1c). Of the eight antibodies that produced signals associated ($p < 0.01$) with VTE in the citrate samples (Supplementary data 1 [Tab_1]), four were replicated in the EDTA samples (Fig. 1b, c): HPA059937 (predicted target SULF1, green point), HPA044659 (predicted target Leukocyte surface antigen CD47 [CD47], blue point), HPA003042 (predicted target Adenosine receptor A2a [ADORA2A], orange point) and HPA002655 (predicted target P-Selectin [SELP], red point). In both anticoagulants, all four target candidates were elevated in cases vs. controls (Fig. 1d, e, Supplementary data 1 [Tab_2]). In all subsequent experiments, citrated blood was used in the analysis.

Previously, in the VEBIOS Coagulation study ($n = 177$)[23], we identified 29 protein candidates in plasma that were associated with prior VTE. This study was composed of patients sampled 1–6 months after discontinuation of anticoagulation treatment (duration 6–12 months) following a first time VTE, or matched controls. Of the four antibodies that generated signals associated with acute VTE in citrate and EDTA

plasma in VEBIOS ER (Fig. 1b, c, marked with coloured circles), only HPA059937 (predicted target SULF1), produced a signal associated with prior VTE in the VEBIOS Coagulation study[23]. As our aim was to identify biomarkers associated with acute VTE that were potentially linked to the underlying disease risk, we prioritised this candidate.

### Complement Factor H Related 5 protein (CFHR5) is associated with VTE

The antibodies used in the single-binder suspension bead arrays passed quality control for antigen binding specificity (see www.proteinatlas.org/), but selective binding to the target protein in context of the complex matrix of plasma requires verification, as antibody specificity and reliability can be a problematic issue[30–32] (Supplementary information Fig. S1c-f). To verify which protein(s) were captured by HPA059937 (predicted target SULF1) we performed immunocapture-mass spectrometry (IC-MS). Two proteins were bound to HPA059937 with a z-score>3 in triplicate experiments; the predicted target, SULF1 (z-score = 4.02, with 1 Peptide Spectrum Match [PSM]), and Complement Factor H Related 5 protein (CFHR5) (z-score = 5.09, with ≥21 PSM) (Fig. 1f and Supplementary data 1 [Tab_3]). These data indicate that CFHR5 was the predominant protein captured by HPA059937 in plasma, whilst not ruling out concurrent binding of SULF1. High levels of CFHR5 have been detected in plasma by mass spectrometry (MS)[33], but SULF1 is below the MS detection threshold[34]. To further verify CFHR5 binding specificity of HPA059937, we developed three dual binder assays (Supplementary information Fig. S1b and Fig. 1g), all with a commercial monoclonal antibody against CFHR5 (MAB3845) as a detection antibody, combined with either: original antibody HPA059937 (Fig. 1g), or one of two independent antibodies raised in house against CFHR5; HPA073894 (Fig. 1g) or HPA072446 (Fig. 1g), as bead coupled capture antibodies. We confirmed that detection antibody (MAB3845) specifically bound CFHR5 in plasma, using IC-MS analysis (Supplementary information Fig. S2a). Western blot analysis showed that both MAB3845 and HPA072446 bound mono and homodimer of recombinant CFHR5, and that HPA072446 detected a band corresponding to CFHR5 size in plasma (Supplementary information Fig. S2b).

When used to re-analyse the VEBIOS ER samples; all three assays consistently detected a higher level of target protein in cases vs. controls (Fig. 1g, $p = 6E-04$, $2.1E-03$, $1E-04$, respectively). Median Fluorescent Intensity (MFI) values from all three strongly correlated with those generated by HPA059937 in the VEBIOS ER discovery screen (Fig. 1g) (Spearman´s ρ = 0.83, 0.75 and 0.82, respectively [all $p < 1E-04$]). We made five dual binder assays that targeted SULF1 using HPA059937 as capture antibody, together with different anti-SULF1 detection antibodies, but none gave a quantitative signal in a plasma dilution series, or buffer containing a dilution series of recombinant SULF1 protein. Western blot analysis showed that the anti-SULF1 HPA059937 detected the monomeric form of recombinant CFHR5 (rCFHR5) under non-reducing, but not in reducing conditions (Supplementary information Fig. S2b), suggesting an off-target binding to an epitope created by a tertiary folded structure of CFHR5. Together, these data are consistent with CFHR5, as opposed to SULF1, being the target protein of HPA059937 associated with VTE in the VEBIOS ER discovery screen.

### CFHR5 is associated with VTE independent of D-dimer or CRP

We used data independent acquisition mass spectrometry (DIA-MS) to perform orthogonal validation of the results obtained from the analysis of CFHR5 plasma levels in VEBIOS ER using the dual binder assay with capture antibody HPA072446 (Fig. 1g). Data from these two independent assays correlated well (ρ = 0.75, $p < 2.2E-16$), and so the dual binder assay was used for quantification of CFHR5 in VEBIOS ER and an extended sample set of the VEBIOS Coagulation study ($n = 284$) (Supplementary data 2 [Tab_1] for cohort descriptive data).

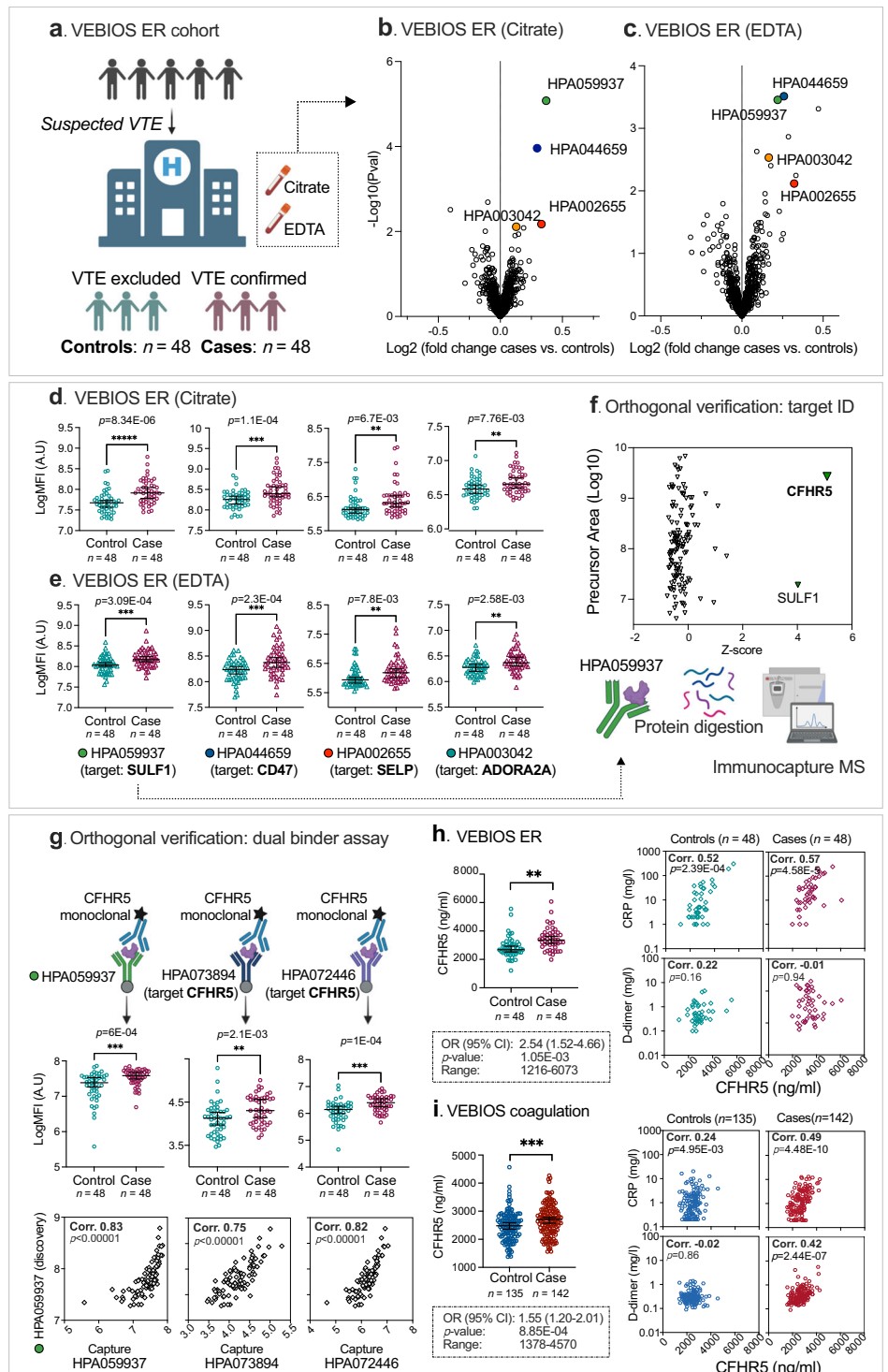

**Fig. 1 | Plasma proteomics profiling identifies CFHR5 associated with VTE.**
**a** Overview of the VEBIOS ER discovery cohort. 756 HPA antibodies, targeting 408 candidate proteins, were used to analyse plasma samples using affinity proteomics. Log fold changes in antibody MFI (median fluorescent intensity) signal were calculated between VTE cases ($n = 48$) and controls ($n = 48$) in (**b**) citrate or (**c**) EDTA anticoagulated plasma; coloured circles indicate antibodies that generated signals significantly associated with VTE in both. MFI signals generated by these antibodies for controls ($n = 48$) and cases ($n = 48$) in (**d**) citrate plasma and (**e**) EDTA plasma. **f** Immunocapture-mass spectrometry identification of protein targets of HPA059937 ($n = 3$ of independent biological replicates). **g** Dual binder assays were developed using an anti-CFHR5 detection antibody, combined with HPA059937 (raised against SULF1), anti-CFHR5 HPA073894 or anti-CFHR5 HPA072446 as capture antibodies. Monoclonal anti-CFHR5 (MAB3845) was applied as detection antibody in the three combinations. CFHR5 levels in the citrated plasma samples were re-analysed, using

the respective dual binder assays, to determine levels (MFI) in controls ($n = 48$) vs. cases ($n = 48$) and the correlation between the signal and those generated by the original single binder assay using HPA059937 [all $p < 1E$-04]. Dual binder assay using capture antibody HPA072446 with a recombinant protein standard and MAB3845 as detection antibody, was used for absolute quantification of CFHR5 in samples from: (**h**) VEBIOS ER and (**i**) VEBIOS Coagulation (controls = 135, cases = 142). CFHR5 concentration was measured in controls and cases, with associated OR (odds ratio per 1 standard deviation increase [**h** and **i**, left]) or used to determine the correlation with C-reactive protein (CRP), or D-dimer concentration [**h** and **i**, right panels]. All data in dot plots (1 **d**–**i**) are represented as median value with 95% CI (confidence interval). Case and control groups was compared using a linear model adjusting for age and sex in **d**, **e**, **g**, **h** and **i**.*****$p < 0.00001$, ****$p < 0.0001$, ***$p < 0.001$, **$p < 0.01$. Two-sided Spearman´s correlation analysis for **h** and **i**. For summary statistics see Supplementary data 1 [Tab_2, Panel A]. Source data are provided as a Source Data file (**b**–**i**).

**Table 1 | Characteristics of VEBIOS ER case control discovery study-clinical variables**

| Variables | Cases (n = 48) | | Controls (n = 48) | | P-value |
|---|---|---|---|---|---|
| **Thrombosis localisation** | | | | | |
| DVT, lower limbs. n; % | 20 | 41.6 | – | – | |
| - Proximal. n; % | 17 | 85 | – | – | |
| PE. n; % | 21 | 43.7 | – | – | |
| DVT and PE. n; % | 7 | 14.5 | – | – | |
| **Sex and biometry** | | | | | |
| Sex; women. n; % | 22 | 45.8 | 22 | 45.8 | 1 |
| Age (years) mean ± SD; [range] | 56.6 ± 18.8 | [19–89] | 56.6 ± 17 | [23–88] | 0.982 |
| BMI (kg/m2) mean ± SD; [range] | 26.7 ± 5.55 | [19.4-44.5] | 28.8 ± 6.67 | [19.8-44.9] | 0.217 |
| - Missing. n; % | 12 | 25 | 26 | 54.2 | |
| Obese (BMI ≥ 30 kg/m2) n; % | 6 | 12.5 | 7 | 14.6 | 0.782 |
| Current smoking ‡ | | | | | |
| - No. n; % | 38 | 79.2 | 37 | 77.1 | 0.309 |
| - Yes. n; % | 4 | 8.3 | 9 | 18.8 | |
| - Missing. n; % | 6 | 12.5 | 2 | 4.2 | |
| **Family history** | | | | | |
| VTE, First degree relative <60 years old | | | | | |
| - No. n; % | 28 | 58.3 | 41 | 85.4 | 0.0388 |
| - Yes. n; % | 14 | 29.2 | 6 | 12.5 | |
| - Missing. n; % | 6 | 12.5 | 1 | 2.1 | |
| **Provoked risk factors (all)†** | 25 | 52.1 | 21 | 43.8 | 0.555 |
| **Estrogen containing contraceptives & hormone replacement therapy\* n; %** | 5 | 10.4 | 5 | 10.4 | 1 |

DVT deep vein thrombosis, PE pulmonary embolism, n numbers, SD standard deviation. P-value was obtained by Student t-test and Pearson's Chi-squared Test for numerical and categorical variables, respectively.

† Within 1 month from diagnosis or index date (immobilization with trauma, surgery, cast and/or orthosis and bedrest >3 days of sickness).

‡ Within the last year.

\* On-going treatment.

In the VEBIOS ER and Coagulation study, mean CFHR5 concentrations in control plasma samples were 2840 ± 756 and 2470 ± 523 ng/ml, respectively (Table 2 and Supplementary data 2 [Tab_1]); levels in the range previously estimated by mass spectrometry (-1900 ng/ml)[33]. In the VEBIOS ER discovery study, the mean absolute CFHR5 concentration was confirmed as higher in patients with confirmed VTE, compared to patients where VTE was ruled out (3430 ng/ml ± 782 [cases] vs. 2840 ng/ml ± 756 [controls], $p = 1.05E{-}03$ [age and sex adjusted]); the odds ratio (OR) for diagnosis of acute VTE associated with one standard deviation (SD) increase of CFHR5 concentration was 2.54 [confidence interval (CI) 1.52-4.66], $p = 1.05E{-}03$ (Fig. 1h, left and Table 2 and 3). Consistent with the relative quantification results in our previous study[23], absolute CFHR5 concentration was associated with prior diagnosis of VTE in the extended VEBIOS Coagulation cohort, compared to controls (mean concentration 2680 ng/ml ± 556 [cases] vs. 2470 ng/ml ± 523 [controls], $p = 8.42E{-}04$ [age and sex adjusted]; the OR for first time VTE was 1.55 [CI 1.20-2.01], $p = 8.85E{-}04$) (Fig. 1i, left, and Table 3 and Supplementary data 2 [Tab_1]).

We next investigated if CFHR5 levels were associated with VTE associated risk factors, such as age, body mass index [BMI] and routine clinical laboratory tests for blood markers associated with thrombosis risk (e.g., D-dimer, c-reactive protein [CRP], thrombocyte count) (Table 2 and Supplementary data 1 [Tab_4]). In VEBIOS ER, CRP levels correlated with plasma CFHR5 concentration, in cases ($\rho = 0.57$, $p = 4.58E{-}05$) and controls ($\rho = 0.52$, $p = 2.39E{-}04$) (Fig. 1h, right panel and Supplementary data 1 [Tab_4, Table A]), but there was no strong correlation between CFHR5 and the other parameters measured, in cases or controls, including D-dimer (Fig. 1h, right panel and Supplementary data 1 [Tab_4, Table A]). Adjusting for CRP, CFHR5 remain significantly associated with acute VTE (OR = 3.31 [CI 1.60-7.73]

$p = 3.15E{-}03$) (Supplementary data 1 [Tab_10]). In the VEBIOS Coagulation study, CFHR5 levels in cases correlated with both CRP ($\rho = 0.49$, $p = 4.48E{-}10$) and D-dimer ($\rho = 0.42$, $p = 2.44E{-}07$) (Fig. 1i, right panel), but in controls these correlations were weak (CRP; $\rho = 0.24$, $p = 4.95E{-}03$) or absent, respectively (Fig. 1i, right panel and Supplementary data 1 [Tab_4, Table B]). The association between CFHR5 and VTE remained significant in VEBIOS Coagulation when adjusted for CRP (OR = 1.55 [CI 1.18-2.03] $p = 1.50E{-}03$) or D-dimer (OR = 1.435 [CI 1.05-1.88] $p = 7.72E{-}03$) (Supplementary data 1 [Tab_10, Table A]).

## CFHR5 measurement can increase diagnostic accuracy in patients with likely VTE

To explore the potential usefulness of CFHR5 as a biomarker to be included in the diagnostic workup of suspected acute VTE, we assessed the discriminatory power of CFHR5 in VEBIOS ER using logistic regression in different models together, with D-dimer dichotomised using current Clinical Decision Rules (CDR) as positive or negative (below age adjusted cut-off [35]) and Wells score (VTE likely (≥2 for DVT and ≥4 for PE) or unlikely) (Supplementary data 1 [Tab_5]). In VEBIOS ER, D-dimer had negative predictive value (NPV) of 100% (0 false negatives) for VTE, while the specificity and positive predictive value (PPV) was only 62.8 and 74% respectively, with 16 false positive cases. Adding CFHR5 to the base model of D-dimer alone resulted in a non-significant improvement in AUC (0.88 vs. 0.82, $p = 0.110$), as did adding Wells score to the base model (AUC 0.85, $p = 0.33$) (Supplementary data 1 [Tab_5]). D-dimer alone performed better than CFHR5 alone (AUC 0.73 vs. 0.82, $p = 0.128$). When stratifying patients based on Wells score, in the group where VTE was considered unlikely based on Wells score ($n = 43$), adding CFHR5 to the base model resulted in a non-significant increased accuracy compared to the base model (AUC 0.84 vs. 0.81, $p = 0.61$).

**Table 2 | Characteristics of VEBIOS ER case control discovery study-blood concentrations**

| Variables | Cases (n = 48) | Controls (n = 48) | P-value |
|---|---|---|---|
| **Concentration of markers measured** | | | |
| CFHR5 (ng/mL) | | | |
| - Mean ± SD | 3430 ± 782 | 2840 ± 756 | <0.001 |
| - Median [Min,Max] | 3360 [1990-6070] | 2680 [1220-5560] | |
| C3 (µg/mL) | | | |
| - Mean ± SD | 668 ± 217 | 660 ± 206 | 0.865 |
| - Median [Min,Max] | 653 [63.7-1210] | 656 [292-1190] | |
| - Missing values (n, %) | 1 (2.1%) | 4 (8.3%) | |
| D-dimer (ng/mL) | | | |
| - Mean ± SD | 4620 ± 4630 | 880 ± 926 | <0.001 |
| - Median [Min,Max] | 3600 [414, 26200] | 521 [92.3, 4410] | |
| - Missing values (n, %) | 2 (4.2%) | 6 (12.5%) | |
| CRP (mg/L) | | | |
| - Mean ± SD | 42.3 ± 56.1 | 23.0 ± 52.4 | 0.1 |
| - Median [Min,Max] | 18.0 [1.00, 246] | 4.50 [1.00, 304] | |
| - Missing values (n, %) | 6 (12.5%) | 2 (4.2%) | |
| LPK (x10⁹/L) | | | |
| - Mean ± SD | 8.68 ± 2.25 | 9.10 ± 5.21 | 0.62 |
| - Median [Min,Max] | 9.10 [3.40, 12.4] | 7.80 [4.10, 36.9] | |
| - Missing values (n, %) | 5 (10.4%) | 3 (6.3%) | |
| Hb (g/L) | | | |
| - Mean ± SD | 134 ± 17.9 | 138 ± 15.2 | 0.349 |
| - Median [Min,Max] | 135 [53.0, 179] | 138 [96.0, 173] | |
| - Missing values (n, %) | 5 (10.4%) | 3 (6.3%) | |
| TPK (x10⁹/L) | | | |
| - Mean ± SD | 221 ± 74.6 | 253 ± 91.4 | 0.073 |
| - Median [Min,Max] | 206 [108,538] | 240 [29.0, 530] | |
| - Missing values (n, %) | 5 (10.4%) | 3 (6.3%) | |

*CFHR5* Complement Factor H-related protein 5, *C3* Complement 3, *CRP* C-reactive protein, *LPK* leucocytes, *Hb* hemoglobin, *TPK* thrombocytes. *P*-value was obtained by Student *t*-test and Pearson's Chi-squared Test for numerical and categorical variables, respectively.

**Table 3 | Association of CFHR5 concentration and VTE**

| | | VEBIOS ER | DFW-VTE | FARIVE | VEBIOS Coag. | RETROVE | Meta-analysis |
|---|---|---|---|---|---|---|---|
| **VTE AL** | OR (1 SD) | 2.54 | 1.80 | 1.24 | 1.55 | 1.29 | 1.35 |
| | (95% CI) | (1.52–4.66) | (1.29–2.58) | (1.10–1.40) | (1.2–2.01) | (1.09–1.53) | (1.23–1.47) |
| | P-value | 1.05E-03 | 7.65E-04 | 3.98E-04 | 8.85E-04 | 2.4E-03 | 1.94E-11 |
| | Cases | 48 | 54 | 582 | 142 | 308 | 1134 |
| | Controls | 48 | 146 | 576 | 135 | 360 | 1265 |
| CFHR5* | (ng/mL) | 1216–6073 | 232–1170 | 450–4904 | 1378-4570 | 364–2341 | 232–6073 |
| **Tertile 1** | OR (1 SD) | Ref. | Ref. | Ref. | Ref. | Ref. | Ref. |
| | P-value | NA | NA | NA | NA | NA | NA |
| | Cases | 10 | 9 | 165 | 39 | 91 | 314 |
| | Controls | 22 | 57 | 221 | 54 | 132 | 486 |
| CFHR5* | (ng/mL) | 1216–2653 | 232–476 | 450–1315 | 1378–2333 | 364-858 | 232–2653 |
| **Tertile 2** | OR (1 SD) | 1.64 | 2.72 | 1.40 | 1.41 | 1.03 | 1.3 |
| | (95%CI) | (0.58–4.80) | (1.15–6.86) | (1.05–1.86) | (0.78–2.55) | (0.69–1.52) | (1.10–1.66) |
| | P-value | 0.36 | 0.03 | 0.02 | 0.25 | 0.88 | 4.38E-03 |
| | Cases | 13 | 21 | 198 | 45 | 94 | 371 |
| | Controls | 19 | 46 | 188 | 47 | 128 | 428 |
| CFHR5* | (ng/mL) | 2675–3338 | 478–601 | 1317–1716 | 2335–2779 | 859–1063 | 478–3338 |
| **Tertile 3** | OR (1 SD) | 9.05 | 2.93 | 1.75 | 2.51 | 1.67 | 1.97 |
| | (95%CI) | (2.93–31.6) | (1.24-7.37) | (1.31–2.33) | (1.39–4.63) | (1.13–2.47) | (1.60–2.42) |
| | P-value | 2.51E-04 | 0.02 | 1.28E-04 | 2.68E-03 | 9.90E-03 | 1.43E-10 |
| | Cases | 25 | 24 | 219 | 58 | 123 | 449 |
| | Controls | 7 | 43 | 167 | 34 | 100 | 351 |
| CFHR5* | (ng/mL) | 3345–6073 | 603–1170 | 1717–4904 | 2782–4569 | 1064–2341 | 603–6073 |

CFHR5*: concentration range (ng/ml). Tertile 1 = reference value 1, *OR* odds ratio, 1 *SD* = 1 standard deviation, *CI* confidence interval. Data adjusted for age and sex.

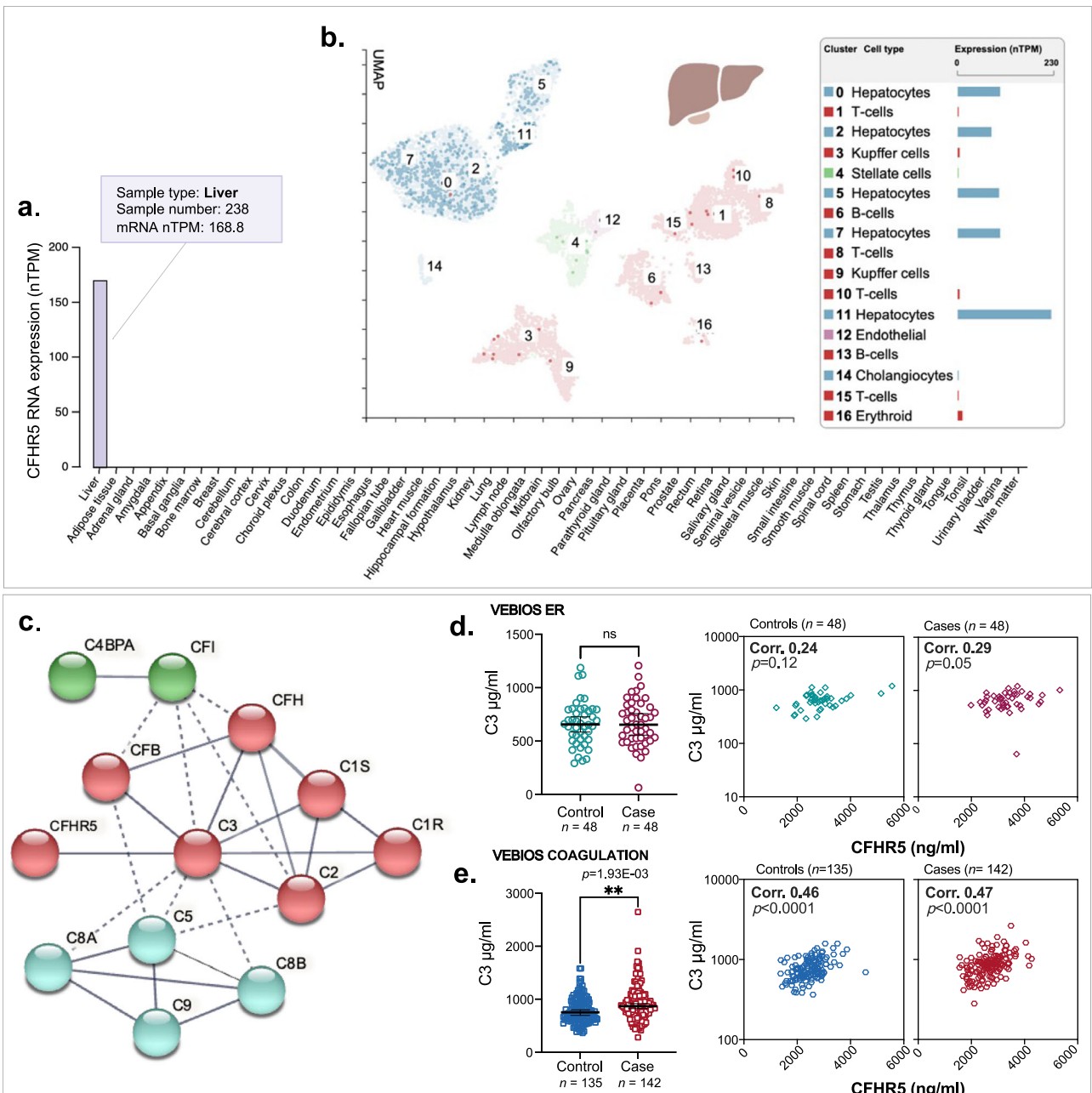

**Fig. 2 | *CFHR5* is expressed in hepatocytes and is VTE-associated independent of C3. a** mRNA expression of *CFHR5* across 55 different human tissue types. **b** Expression of *CFHR5* in different liver cell types, analysed by ssRNAseq. **c** STRING protein-protein interaction analysis for genes identified as potentially co-expressed with *CFHR5* in liver by correlation-based analysis of bulk mRNAseq. Complement component 3 (C3) concentration was measured in plasma from (**d**) VEBIOS ER (*n* = 48 + 48) or (**e**) VEBIOS coagulation (*n* = 135 + 142) to determine: differences between controls and cases (**d** and **e**, left), or correlation with CFHR5 in controls or cases (**d** and **e**, right panels). Case and control groups was compared using a linear

model adjusting for age and sex (**d**, **e**) **\*\****p* < 0.01. Dot plots (**d**, **e**) are represented as median value with 95% CI. Two-sided Spearman´s correlation analysis for 2**d** and 2**e**. Summary of statistical analysis can be found in Supplementary data 1 [Tab_2, Panel B]. Source data are provided as a Source Data file (**d**, **e**). C4BPA complement factor 4 binding protein, CFI complement factor I, CFB complement factor B, CFH complement factor H, C1S complement component 1, C1R complement component 1, C2 complement component 2, C8a complement component C8 alpha chain, C8b complement component C8 beta chain, C5 complement component 5, C9 complement component 9.

However, in the group where VTE was considered likely (*n* = 41), the addition of CFHR5 to the base model resulted in a significantly increased accuracy compared to D-dimer alone (AUC 0.92 *vs.* 0.83; *p* = 0.035) (Supplementary data 1 [Tab_5]).

## CFHR5 is specifically expressed in liver hepatocytes with other complement related genes

To further understand the expression characteristics of *CFHR5*, and to identify possible co-expressed or co-regulated proteins we used a

whole transcriptome analysis approach. In a consensus dataset, consisting of normalized mRNA transcript (nTPM) levels across 55 different human tissue types, generated from Human protein Atlas (HPA) (https://www.proteinatlas.org)[36] and Genotype-Tissue Expression Project (www.gtexportal.org)[37] datasets, *CFHR5* was highly and specifically expressed in the liver (Fig. 2a). Single cell analysis of liver tissue[38], showed that *CFHR5* was specifically expressed in the hepatocyte cellular compartment (Fig. 2b). To identify transcripts potentially co-expressed or co-regulated with *CFHR5* in the liver, we analysed bulk

RNAseq data ($n$ = 226) retrieved from GTEx portal V8. We generated Pearson correlation coefficients between *CFHR5* and all other expressed protein coding transcripts ($n$ = 19,525) (Supplementary data 1 [Tab_6, Table A]) and used gene ontology (GO) and reactome analysis[39,40] to identify over-represented classes and pathways in the top 50 most highly correlated genes (Supplementary data 1 [Tab_6, Table B and C]). Results were consistent with known CFHR5 function; significant GO terms included ´complement activation´ (FDR adjusted $p$-value [PFDR] = 2.4E-16) and ´humoral immune response´ (PFDR = 1.2 E-12) and reactome pathways included ´regulation of complement cascade´ (PFDR = 3.8E-24). We performed an unbiased weighted network correlation analysis (WGCNA)[41] on the same dataset, where correlation coefficients between all transcripts, excluding those classified as non-coding, were calculated and subsequently clustered into related groups, based on expression similarity (Supplementary data 1 [Tab_6, Table A - column F]). Transcripts that fulfilled the criteria of: (i) Pearson correlation with *CFHR5* > 0.65 ($p$ < 0.001) and (ii) annotation to the *CFHR5*-containing gene cluster in WGCNA (group 68), were identified as those most likely to be co-expressed or co-regulated. Of these, 13/18 [72%] were other members of the complement cascade and 15/18 [83%] were also specifically expressed in liver[36] (Supplementary data 1 [Tab_6, Table A - column F]) (Supplementary data 1 [Tab_6, Table A - column B, bold text]). While these data indicate a degree of co-expression with *CFHR5* at the transcriptional level, plasma concentrations of the encoded proteins are subject to several post-transcriptional variables, such as translation efficiency, cellular release dynamics, protein stability and clearance. When these proteins were interrogated using the protein-protein interaction database STRING, v.11[42], 13/18 had high confidence functional and physical associations (Fig. 2c) (Supplementary data 1 [Tab_7, Table A]). CFHR5 was most strongly linked to complement 3 (C3), the central hub of the largest of the three linked cluster groups identified (Fig. 2c, clusters represented by green, red and cyan) (Supplementary data 1 [Tab_7, Table B]).

## CFHR5 is associated with acute VTE independent of C3

Plasma levels of complement component C3 have previously been reported as associated with incident VTE[43]. To determine if the association between CFHR5 and VTE we observed is dependent on the concentration of C3, we developed an in-house dual binder quantitative assay to measure C3 in the VEBIOS ER and VEBIOS Coagulation cohorts. In VEBIOS ER, plasma C3 was not elevated in cases, compared to controls (Fig. 2d, left and Supplementary data 1 [Tab_2, Panel B]), CFHR5 and C3 did not significantly correlate in either group (Fig. 2d, right panel) and C3 was not associated with VTE (OR 1.04 [CI 0.68-1.58], $p$ = 0.86) (Supplementary data 1 [Tab_8]). Furthermore, the association with acute VTE for one SD increase in CFHR5 level remained unchanged (OR 2.65 [CI 1.53-5.01], $p$ = 1.26E-03) when including and adjusting for C3 concentration (together with age and sex), compared to when only adjusting for age and sex ((OR 2.54 [1.52-4.66], $p$ = 0.001), Table 3, Supplementary Data 1 [Tab_8]), demonstrating that CFHR5 is independently associated with acute VTE. In VEBIOS Coagulation, C3 levels were higher in plasma from cases, compared to controls (Fig. 2e, left), and CFHR5 and C3 correlated with each other in both (controls ρ = 0.46 $p$ = <0.0001; cases ρ = 0.47 $p$ = <0.0001) (Fig. 2e, right panel). After adjusting for age and sex, one SD increase in C3 concentration was significantly associated with previous VTE (OR 1.52 [CI 1.18-2.01], $p$ = 1.93E-03). When adjusting for CFHR5 levels (together with age and sex), this no longer reached significance (OR 1.31 [CI 0.99-1.78], $p$ = 0.064). The association with previous VTE for one SD increase in CFHR5 level in VEBIOS Coagulation was still nominally significant when adjusting for C3 levels (OR 1.36 [CI 1.03-1.82], $p$ = 0.032), although weaker compared to adjusting only for age and sex (OR 1.55 [CI 1.2-2.01], $p$ = 8.85E-04, Table 3 and Supplementary data 1 [Tab_8]).

## The CFHR5 association with VTE replicates in additional cohorts

The identification of biomarkers associated with VTE diagnosis, or risk profiling, requires replication in independent cohorts, from different settings with different demographic profiles, to determine feasibility for potential translation to clinical practice. We sourced three independent replication cohorts to test the association of CFHR5 with VTE; the Swedish Karolinska Age Adjusted D-dimer study (DFW-VTE) VTE study ($n$ = 200) consisting of patients with suspected VTE (cases; $n$ = 54, controls; $n$ = 146)[44] (Fig. 3b), the French FARIVE study ($n$ = 1158) consisting of patients sampled during the week following a diagnosis of acute VTE ($n$ = 582), with hospital-based controls ($n$ = 576)[45] (Fig. 3c) and the Spanish Riesgo de Enfermedad TROmboembólica VEnosa (RETROVE) study ($n$ = 668) of patients sampled post anticoagulant treatment ($n$ = 308), with population based controls ($n$ = 360)[46] (Fig. 3e) (for all cohort details see Supplementary data 2 [Tabs_2–4 and 6]). The OR of VTE associated with CFHR5 per 1 SD increase in CFHR5 concentration was significant in all 3 replication cohorts: DFW-VTE (OR 1.80 [CI 1.29-2.58], $p$ = 7.65E-04) (Fig. 3b), FARIVE (OR 1.24 [CI 1.10-1.40], $p$ = 3.98E-04) (Fig. 3c), RETROVE (OR 1.29 [CI 1.09-1.53], $p$ = 2.4E-03) (Fig. 3e) (Table 3). When samples from cases and controls were stratified according to CFHR5 concentration, the association with VTE was most pronounced in the third tertile, in all 5 cohorts analysed individually and in a meta-analysis (Table 3). These associations remain significant in subgroup meta-analyses when stratified by thrombosis type (DVT or PE), sex, or cause (provoked/unprovoked) (Supplementary data 1 [Tab_9, Table A–C]). In subgroup analyses in the individual cohorts, the association of CFHR5 with VTE did not reach significance in females in VEBIOS Coagulation and FARIVE and in males in DFW-VTE. Furthermore, the association with provoked VTE in RETROVE and with unprovoked VTE in FARIVE were not significant. The results were consistent when further adjusting for BMI and/or CRP when this information was available (Supplementary data 1 [Tab_10, Table A–D]).

## CFHR5 and risk of recurrent VTE

We measured plasma CFHR5 concentration in a sample of 669 VTE patients from the MARseille Thrombosis Association Study (MARTHA) study that have been followed for VTE recurrence, among which 124 experienced a recurrent event (Supplementary data 2 [Tab_5])[47]. After adjusting for sex, familial history of VTE, provoked or unprovoked status of the first VTE, age at first VTE, and BMI, the Hazard Ratio (HR) associated of 1 SD increase in CFHR5 levels was HR = 1.13 [0.96-1.32], $p$ = 0.134. The association was consistent between females (HR = 1.1 [0.90 −1.38], $p$ = 0.320) and males (HR = 1.14 [0.91-1.44], $p$ = 0.260) and between patients with DVT (HR = 1.18 [0.98-1.42], $p$ = 0.080) or PE as first event (HR = 1.13 [0.80-1.61], $p$ = 0.489). This trend for association was strongest in the subgroup of patients with unprovoked first VTE (HR = 1.32 [0.99–1.77], $p$ = 0.056), as no association was observed when the first event was provoked (HR = 1.01 [0.83–1.23], $p$ = 0.90). However, the test for heterogeneity between these two HRs did not reach 0.05 significance ($p$ = 0.23).

## Genome wide association study on CFHR5 plasma levels

To explore if CFHR5 concentration in plasma was influenced by genetic variants, we first performed a meta-analysis of GWAS for dual binder assay based CFHR5 concentrations in individuals from the FARIVE ($n$ = 1033), RETROVE ($n$ = 668) and MARTHA ($n$ = 1266) studies. The results from the association results are summarised in Supplementary information Fig. S3a. Of 7,135,343 SNPs tested in a total sample of 2967 individuals, one genome-wide significant ($p$ < 5E-08) signal was observed on chr1q31.3. The lead SNP at this locus was rs10737681, mapping to *CFHR1/CFHR4* (Supplementary information Fig. S3b), and the G allele was associated with a one SD increase in CFHR5 levels of $\beta$ = +0.25 ± 0.03 ($p$ = 6.49E-21). In the latest GWAS for VTE risk built on ~72 K cases and >1 M controls of European ancestry[17], the

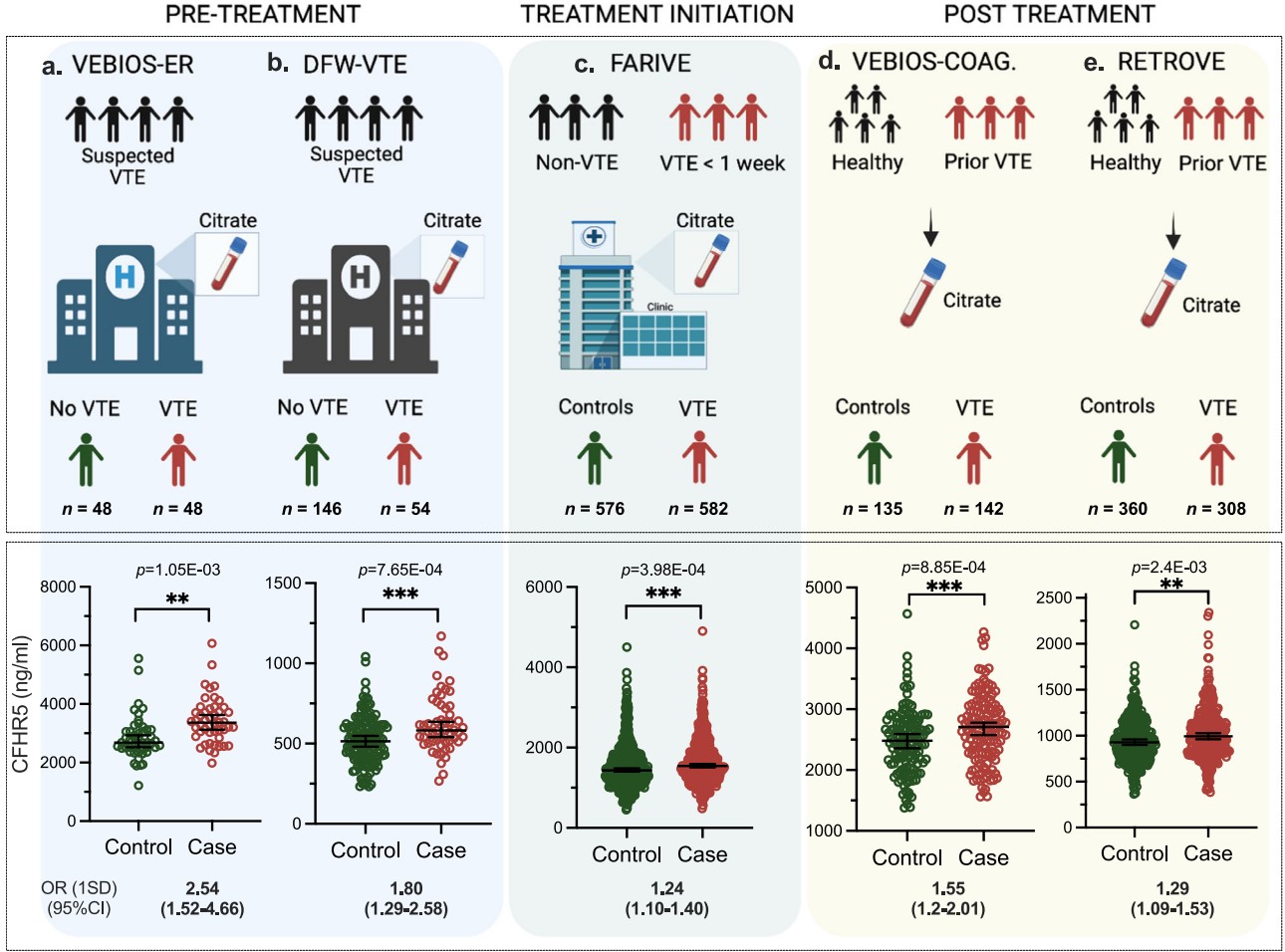

**Fig. 3 | CFHR5 concentration is associated with VTE in 5 independent studies.** Plasma samples were generated as part of: (**a**) the Swedish VEBIOS ER or (**b**) the Swedish DFW-VTE study, both of which recruited patients presenting with suspected VTE. Samples were drawn pre- treatment, and cases and controls were identified based on confirmed or ruled out diagnosis. **c** The French FARIVE study recruited patients with confirmed acute VTE, with controls recruited from hospital patients treated for non-VTE causes. Samples were drawn within 1 week from diagnosis, during initiation of treatment (**d**) The Swedish VEBIOS Coagulation or (**e**), Spanish RETROVE study recruited cases from patients who had a prior first time VTE, sampled post-treatment (6–12 months anticoagulants), with healthy controls recruited from the general population. CFHR5 concentration was measured in the respective samples using a dual binder assay. Case and control groups was compared using a linear model adjusting for age and sex (3**a**–**e**). ***$p < 0.001$, ****$p < 0.0001$. OR (1 SD) = Odds ratio for 1 standard deviation elevation. CI confidence interval. All dot plots are represented as median value with 95% CI. Source data are provided as a Source Data file.

rs10737681–G allele was associated with a marginal ($p = 0.016$) decreased risk of VTE (Supplementary data 1 [Tab_11]). This pattern of association is not consistent with the relationship between increased CFHR5 plasma levels and increased VTE risk observed in the present studies, since the rs10737681-G *CFHR5* increasing allele would have been expected to be associated with increased VTE risk (Supplementary data 1 [Tab_11]). A second round of meta-analysis, integrating GWAS summary statistics from 3 additional proteogenomic resources where *CFHR5* was measured with different assays [see methods], totalling ~50,200 individuals, confirmed the association of this locus with CFHR5 plasma levels. Interestingly, the rs10737681 was also identified as the lead SNP at this locus in the extended meta-analysis ($\beta = 0.28 \pm 0.01$, $p = 2.94E\text{-}396$) (Supplementary data 1 [Tab_12]). Strong linkage disequilibrium holds at the locus covering >10 Mb and extending from *CFHR1* to *CFHR5* (Fig. 4b). The extended meta-analysis identified 5 additional independent loci associated with CFHR5 levels: *HNF1A* (rs2393776, $p = 1.48E\text{-}21$) on 12q24.31, *JMJD1C* (rs7916868, $p = 4.61E\text{-}12$) on 10q21.3, *TRIB1* (rs28601761, $p = 4.39E\text{-}09$) on 8q24.13, *DNAH10* (rs7133378, $p = 2.43E\text{-}08$) also on 12q24.31 and *HNF4A* (rs1800961, $p = 4.97E\text{-}08$) on 20q13.12 (Fig. 4a and Supplementary data 1 [Tab_12]). All of the lead SNPs at these loci, except *HNF1A*

rs2393776 ($p = 0.17$), demonstrated marginal ($p < 0.05$) association with VTE risk (Supplementary data 1 [Tab_11]). However, only two, *JMJD1C*_rs7916868 and *DNAH10*_rs7133378, showed patterns of association with VTE that are compatible with the association of increased CFHR5 plasma levels with VTE risk. This explains why Mendelian Randomization (MR) analyses are not supportive for a causal association between increased CFHR5 levels and VTE (Supplementary data 1 [Tab_13]).

Of note, 1230 MARTHA participants with CFHR5 plasma levels have also been genotyped with an Illumina exome12 v1.2 DNA array[48] dedicated to the genotyping of coding polymorphisms, mainly of low frequency. Capitalizing on this additional genetic resource, we investigated whether low-frequency coding variants at the *CFHR5* locus (including the nearby *CFHR1/CFHR2/CFHR3/CFHR4* genes) could contribute to the inter-individual variability of CFHR5 plasma levels. Twelve rare variants were found polymorphic at this locus in MARTHA participants (Supplementary data 1 [Tab_14]). Three of these variants showed evidence for association with CFHR5 in plasma (in bold). These were three rare non-synonymous *CFHR5* variants: rs139017763 (G278S) $p = 4.75E\text{-}05$, rs41299613 (C208R) $p = 1.6E\text{-}03$ and rs35662416 (R356H) $p = 7.1E\text{-}03$, where rare minor alleles were associated with decreased

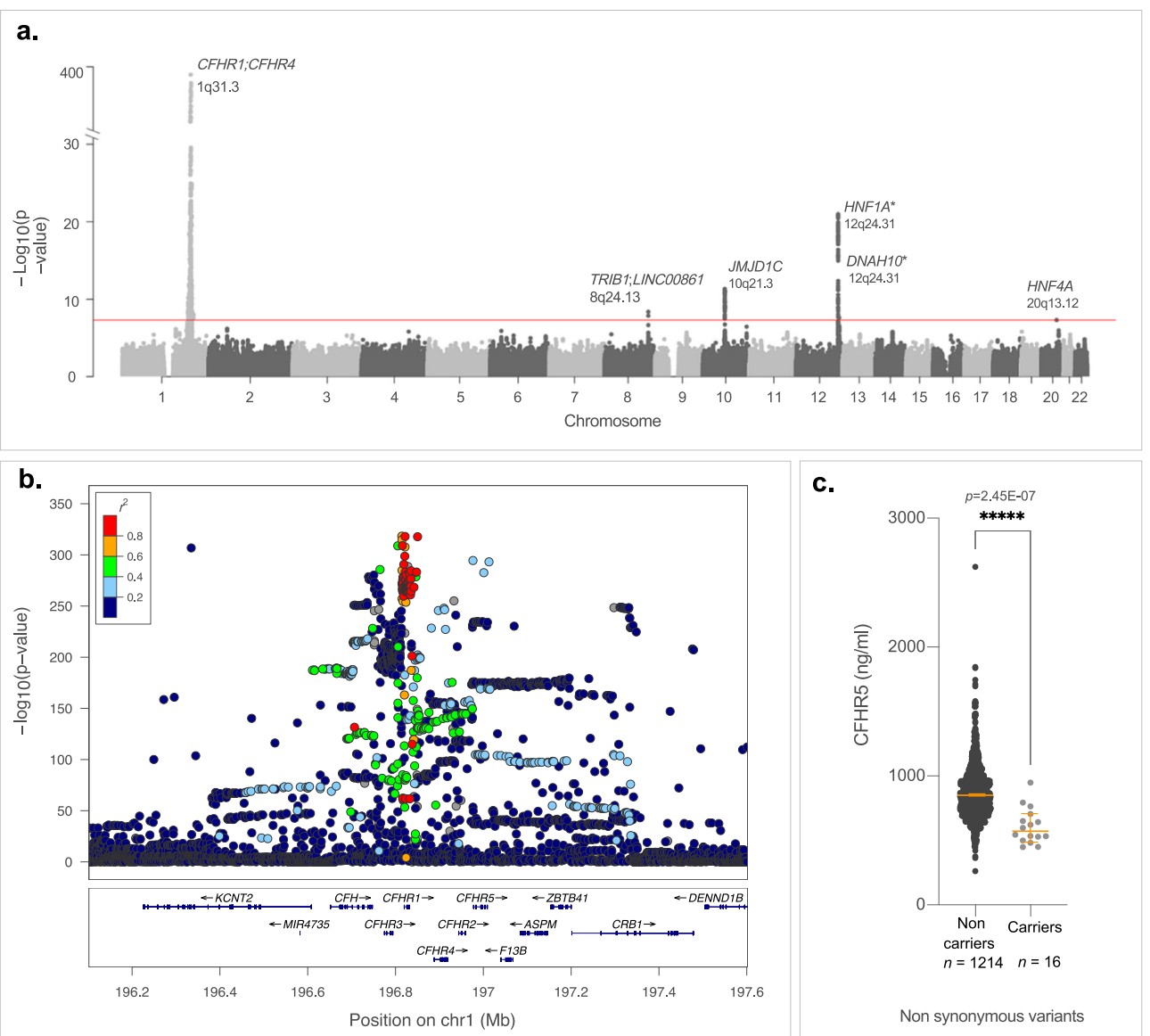

**Fig. 4 | GWAS analysis identifies a CFHR5 pQTL on Chromosome 1 q31.3.**
**a** Manhattan plot of the meta-analysis on INVENT-MVP consortium resources [17] showing six loci associated with CHFR5 plasma levels and VTE risk: *CFHR1, CFHR4* (rs10737681, *p* = 2.94E-396), *HNF1A* (rs2393776, *p* = 1.48E-21), *JMJD1C* (rs7916868, *p* = 4.61E-12), *TRIB1* (rs28601761, *p* = 4.39E-09), *DNAH10* (rs7133378, *p* = 2.43E-08) and *HNF4A* (rs1800961, *p* = 4.97E-08). Lead SNPs at *HNF1A* and *DNAH10* are rs2393776 and rs7133378, respectively. They are ~3 Mb apart and do not show any linkage disequilibrium (pairwise *r²* = 0). **b** Regional association plot[127] at the CFHR5 plasma levels. It is important to emphasize that these 3 *CFHR5* non-synonymous variants were carried by 16 distinct individuals. Figure 4c illustrates the difference in CFHR5 plasma levels between the

Chromosome 1 locus covering >10 Mb from *CFHR1* to *CFHR5* around the lead SNP associated with CFHR5 plasma levels. **c** CFHR5 plasma levels for 16 patients who are carriers of rare non-synonymous *CFHR5*-associated variants (rs139017763, rs41299613 or rs35662416) and non-carriers (*n* = 1214). *t*-test, two sided. *****p* < 0.00001. Dot plot (**c**) is represented as median value with 95% CI. See also Supplementary data 1 [Tabs_11–13]. Source data are provided as a Source Data file (**c**).

16 carriers of rare CFHR5-associated variants and non-carriers. This difference remained significant (*p* = 2.45E-07) after adjusting for the common rs10737681 variant identified in the GWAS analysis.

## CFHR5 is associated with thrombin generation potential in patients with previous VTE

As thrombin generation has been associated with increased risk of VTE[49], we tested the association between CFHR5 plasma concentration and thrombin generation as measured by thrombinoscope in MARTHA (*n* = 774 VTE cases, see Supplementary data 2 [Tab_5] for details) with replication in RETROVE (308 cases/360 controls, see Supplementary

data 2 [Tab_4] for details). In both MARTHA and RETROVE cases, we find significant association between CFHR5 and lag time (ρ = 0.181, *p* = *p* < 0.0001 and ρ = 0.176, *p* < 0.0001, respectively), Endogenous Thrombin Potential (ETP) (ρ = 0.105, *p* = 0.0036, and ρ = 0.130, *p* < 0.0001, respectively), peak (ρ = 0.117, *p* = 0.0012, and ρ = 0.132, *p* < 0.0001), and ttPeak (ρ = 0.116, *p* = 0.0012, and ρ = 0.086, *p* = 0.0274) (see Supplementary data 1 [Tab_15]).

## CFHR5 enhances platelet activation and degranulation in plasma

C3a, generated by cleavage of C3, can increase platelet activation[50–52]. As CFHR5 has a regulatory role upstream of C3/C3a activation, we investigated the effect of recombinant CFHR5 (rCFHR5) on platelet activation in vitro. Functionality of the rCFHR5 was validated by its

capacity to form a homodimer complex and its ability to bind known interaction partners C-reactive protein (CRP)[53] and properdin[54] (see methods and Supplementary information Fig. S2b and Fig. S4). Platelet rich plasma (PRP) was pre-incubated with 6 μg/ml recombinant CFHR5, a concentration corresponding to the upper range of that detected in the plasma of the VTE case group in VEBIOS ER, and in agreement with what has been reported in previous literature[53]. Platelet activation was measured by surface expression of P-selectin, activated GP IIb/IIIa or CD63 (Figs. 5a–c, respectively) in response to adenosine diphosphate (ADP), convulxin or TRAP6 (Fig. 5a–c, top, middle and bottom panels, respectively). Following stimulation with ADP, a higher percentage of platelets pre-incubated with CFHR5 expressed P-selectin (Fig. 5a, top) ($p = 0.0056$), activated GP IIb/IIIa (Fig. 5b, top) ($p = 0.031$) and CD63 (Fig. 5c, top) ($p = 0.009$), compared to the control. Pre-incubation with CHFR5 also potentiated platelet activation in response to convulxin or TRAP6 stimulation (Fig. 5a–c, middle and bottom panels, respectively), although the effect appeared to be more strongly linked to stimulus concentration, than that observed for ADP. Although CFHR5 potentiated the expression of platelet activation markers in response to ADP, it did not modify ADP-induced platelet aggregation (Fig. 5d). Washed platelet response to ADP, convulxin or TRAP6 was not modified by preincubation with CFHR5 (Supplementary information Fig. S5) (ANOVA all $p > 0.05$), indicating that additional components in plasma were required for the observed response, and that they are not a direct effect of CFHR5 on platelets. In PRP anticoagulated with hirudin, baseline detection of activated GP IIb/IIIa (PAC1 + ) on unstimulated and ADP stimulated platelets was significantly lower compared to when citrate was used as anticoagulant (mean % expression ± std dev). [unstimulated: citrate $9.6 ± 4.8$ vs. hirudin $1.6 ± 1.2$] and [ADP-stimulated: citrate $30 ± 15.0$ vs. hirudin $7.5 ± 3.2$]. We did not observe any effect of recombinant CFHR5 on activated GP IIb/IIIa (PAC1+) on platelets from hirudin anticoagulated blood following ADP stimulation (Supplementary information Fig. S5d, e).

A proposed function of CHFR5 is that it augments complement activation by antagonising complement factor H (CFH), the main negative regulator of alternative pathway (AP) activation in plasma. CFH inhibits C3 convertase, preventing formation of C3a[55], which has been suggested to have role in platelet activation and subsequent thrombosis formation[50]. To determine if CFHR5-induced augmentation of platelet activation was dependent on C3 cleavage and generation of C3a, we pre-treated platelets with an inhibitor of C3 cleavage and activation of C3a (compstatin), or an anti-C3a antibody, prior to CFHR5 pre-incubation and subsequent ADP stimulation. The potentiation effect of CFHR5 on baseline and ADP-induced activated GP IIb/IIIa expression (Fig. 5e, f) was abolished following compstatin (Fig. 5e) or anti-C3a antibody (Fig. 5f) pre-treatment; data consistent with a complement dependent effect of CFHR5 on platelet activation.

### Complement fragment 3c concentration correlates with CFHR5 in VEBIOS ER subset

As a marker for C3 cleavage and activation in plasma, we measured complement fragment 3c (C3c) in a subset of plasma samples from VEBIOS ER, selected based on a low ( < 2500 ng/ml, 10 samples) or high ( > 3800 ng/ml, 10 samples) plasma CFHR5 concentration. Mean C3c concentration was greatest in the high CFHR5 group (C3c (ng/ml) ± std dev: CFHR5 low: $0.91 ± 0.2$ vs. CFHR5 high: $1.08 ± 0.2$), although this difference failed to reach statistical significance ($p < 0.086$) (Supplementary information Fig. S6a). However, C3c and CFHR5 concentrations were positively correlated across this sample set ($ρ = 0.51$, $p < 0.02$) (Supplementary information Fig. S6b).

### CFHR5 does not induce a procoagulant response to inflammation in endothelial cells

To investigate a potential effect of CFHR5 on endothelial cells, we treated primary human umbilical vein endothelial cells (HUVECs) with

recombinant CFHR5 under unstimulated and TNF stimulated conditions and assessed the effect on coagulation and inflammation, using thrombin generation assay (TGA) and measurement of mRNA expression of several markers (F3, IL8, vWF, THBD, TFPI, PLAT, ICAM1). No differences were observed in the presence of rCFHR5 compared with buffer (PBS), under any of the conditions (Supplementary information Fig. S7).

## Discussion

Here, we aimed to identify biomarkers associated with acute VTE that are linked to disease pathogenesis and risk. Using a nested case-control study, derived from a cohort of patients presenting to the ER with suspected acute VTE, and from a case-control study with patients that had suffered a previous first VTE, we identify CFHR5, a regulator of the alternative complement activation pathway, as such a biomarker. The association of CFHR5 with current or previous VTE was replicated in three additional cohorts or case-control studies, and we also found a trend for association with risk for recurrence of unprovoked VTE. We identify 6 independent loci with CFHR5 levels including the CFHR1-5 gene cluster loci. We further provide evidence of a direct role of CFHR5 in the induction of a pro-thrombotic phenotype, through its effect on platelet activation. Our findings indicate CFHR5 has potential application as a clinical biomarker for VTE diagnosis and risk prediction, providing further support to the idea that complement regulation is a key element of VTE pathogenesis.

Currently, D-dimer is the only plasma biomarker used in VTE diagnostic workup, but its clinical utility is limited to ruling out VTE in low-risk patients. Several studies have attempted to identify novel biomarkers with potential clinically usefulness for the confirmation of VTE diagnosis, and although a number have been identified[7], none have yet been implemented in clinical practice. For many, like D-dimer, elevated levels are a consequence of thrombosis formation, e.g. biomarkers of fibrinolysis, clot re-modelling or resolution (e.g. MMPs), inflammation secondary to local vascular and tissue injury (e.g., CRP, IL-6, IL-10, fibrinogen), or of endothelial and/or platelet activation (e.g., vWF, P-selectin)[7,56,57]. We found no correlation or association between D-dimer and CFHR5 in the acute VTE setting, supportive of that increased CFHR5 concentration at diagnosis is not secondary to thrombus formation. In contrast, we found a strong correlation between D-dimer and CFHR5 levels in patients followed up after ending treatment for a first VTE, but not in controls. D-dimer has been associated with increased risk of first and recurrent VTE[8–14,58] and thus, our results are consistent with a link between CFHR5 and subclinical coagulability in these patients at follow-up, possibly due to persistent risk factors.

The CFHR5 locus maps to chromosome 1q31.3 at one end, a gene cluster that spans ~350 kb including (in order from CFHR5) the CFHR2, CFHR4, CFHR1, CFHR3, and CFH loci. The rs10737681 we identified with genome wide association with plasma CFHR5 level maps between the CFHR4 and CFHR1 genes. Of note, the CFHR2 locus has just been identified as a novel susceptibility locus for VTE in the recently published international effort on VTE genetics[17]. The lead SNP at this locus is rs143410348, which is in moderate LD with the rs10737681 ($r^2$ ~ 0.40 in European population[59]), which here we found associated with CFHR5 plasma levels. In a combined meta-analysis of 37,770 individuals from 3 cohorts (EPIC/FARIVE/Omicscience) where it was imputed, rs143410348 was less strongly associated with CFHR5 levels than the lead rs10737681 ($β = −0.15$, $p = 1.9E-32$ vs. $β = −0.27$, $p = 2.1E-95$). Altogether, the observations from the GWAS analyses emphasize the need for a deeper investigation of the genetic architecture of the CFHR3/CFHR1/CFHR4/CFHR2/CFHR5 locus with respect to CFHR5 levels and VTE risk. Five additional candidate loci were identified as participating to the genetic regulation of CFHR5 plasma levels: DNAH10, HNF1A, HNF4A, JMJD1C and TRIB1. All 5 loci have been reported to associate with various lipids traits (see GWAS catalogue[60]) and 3

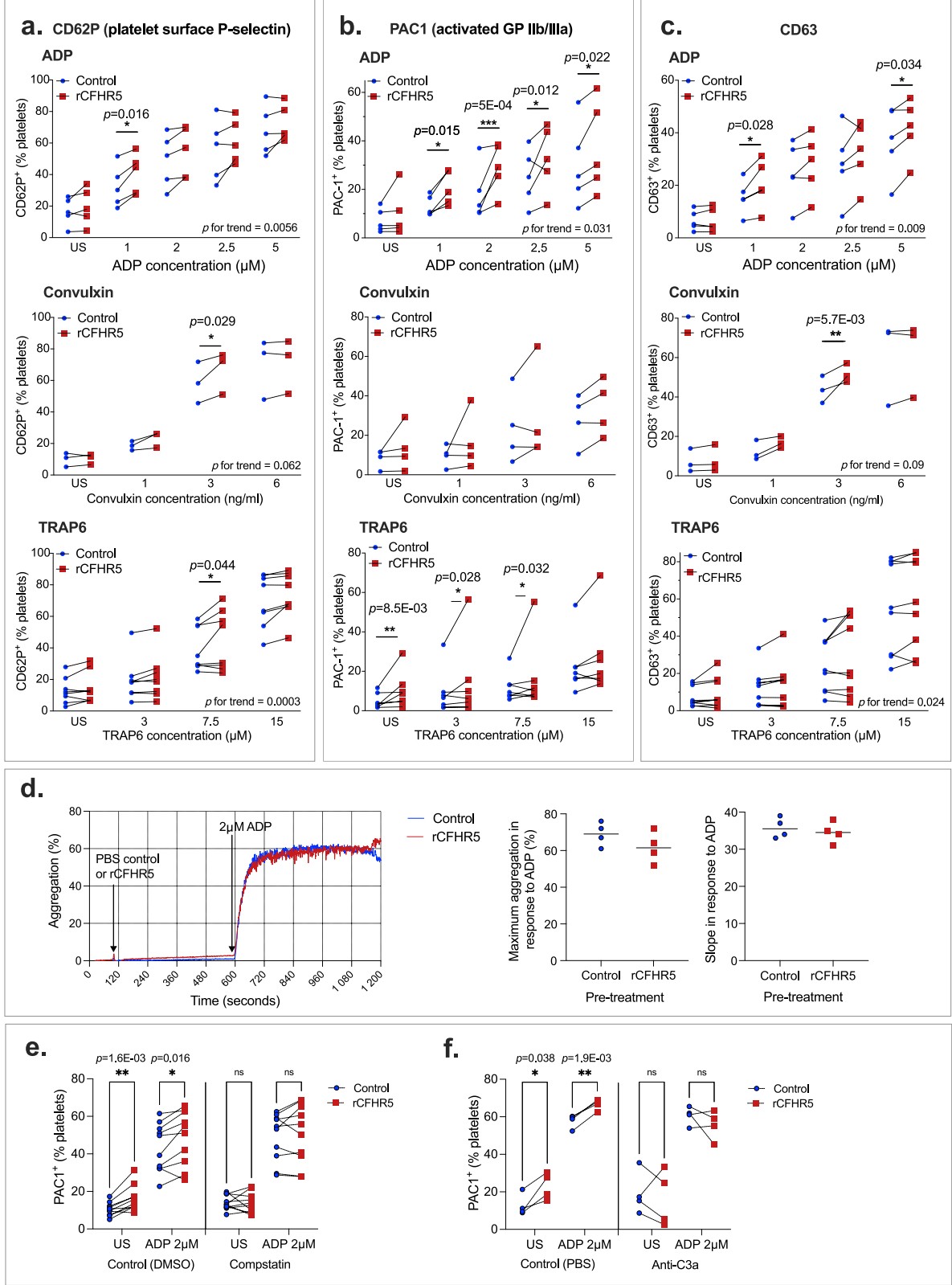

**Fig. 5 | Recombinant CFHR5 enhances platelet activation in platelet rich plasma.** Platelet activation was measured by surface expression of (**a**) P-selectin, (**b**) activated GP IIb/IIIa (PAC1$^+$) or (**c**) CD63, following treatment of platelet rich plasma with different concentrations of adenosine diphosphate (ADP), convulxin or TRAP6, [**a**–**c**: top, middle and bottom panels, respectively] following pre-incubation with recombinant CFHR5, or PBS control. **d** Platelet aggregation was measured in response to ADP (2 µm) following pre-incubation with recombinant CFHR5, or PBS control: representative aggregation curve [left], maximum aggregation [middle] and slope [right], of four independent experiments. ADP-induced platelet activated GP IIb/IIIa (PAC1$^+$) was measured following preincubation with: (**e**) DMSO (control) or compstatin, or (**f**) PBS (control) or an anti-C3a antibody, followed by treatment with PBS or rCFHR5. US: unstimulated (PBS control). Each experiment is represented by an individual point and paired experiments connected by a dotted line. Anova (**a**–**c**) and $t$-test (**d**–**f**) were performed (two-tailed). *$p < 0.05$ **$p < 0.01$ ***$p < 0.001$ ($p$ for trend bottom right). Source data are provided as a Source Data file (Fig. 5).

(*HNF1A, HNF4A* and *TRIB1*) have been reported to also associate with liver enzymes. *DNAH10* is also known to be a locus involved in white & red blood cell biology[61] while the *JMJD1C* is a locus linked to platelet biology[62,63]. Most of these traits are well known risk factors for VTE, and this may suggest that the association of CFHR5 levels with VTE risk implies many additional biological players with pleiotropic effects. This may explain why the MR analyses did not provide causal evidence for a link between CFHR5 plasma levels and VTE risk.

The complement and haemostatic systems interact at several points during initiation, propagation, and regulation of complement activation and coagulation[64]. Studies have indicated a role of complement in VTE pathogenesis[43,51], but underlying mechanisms are not well understood.

CFHR5 shares protein sequence and structural homology with Complement Factor H (CFH)[65], the main negative regulator of alternative pathway (AP) activation in plasma[55]. Under normal conditions, the AP is constitutively active through spontaneous hydrolysis of the thioester bond in C3 and the formation of the initial fluid phase C3 convertase, C3($H_2O$)Bb, which cleaves C3 into C3a and C3b[55]. CFH promotes decay of the alternative and classical pathway convertases and is a cofactor in the cleavage of C3b, hereby regulating excess activation of C3[66]. CFH inhibits C3 convertase, preventing formation of C3a. CFHR5 antagonizes CFH function, through competitive binding to C3b and its fragment C3d[67], thus deregulating AP activation. CFHR5 also promotes complement activation by interfering with CFH binding to CRP, pentraxin 3 (PTX3), and extracellular matrix (ECM)[68].

Elevated plasma C3 in baseline samples has been shown to be associated with increased risk of future VTE[43]. Consistent with these findings, C3 was associated with prior VTE in the VEBIOS coagulation study, but not with acute VTE in the VEBIOS ER study. In both cases, and in the previous study by Nordgaard et al.[43] total C3 level, rather than the active form (C3a) is measured; it is possible that in acute VTE, regulation of C3 convertase (by CFHR5) is important, rather than absolute C3. Consistent with this, we observe a trend for higher plasma levels of complement C3c fragment, a marker of C3 activation, in samples with higher CFHR5 concentrations at VTE diagnosis. It could be speculated that the association of C3 with VTE in individuals sampled pre-VTE[43] or following treatment for a prior VTE, reflects co-regulation of CFHR5 and C3 under basal conditions, which would be consistent with our finding that in VEBIOS coagulation, the association with VTE for CFHR5 was weaker when adjusting for levels of C3, and vice versa.

The mechanisms underlying venous and arterial thrombosis development differ; venous thrombi contain an abundance of red blood cells trapped in a fibrin clot together with platelets, a structure quite distinct from the vast platelet aggregates found in arterial thrombi[69]. Thus, arterial thrombosis is treated with therapies that target platelet activation and/or aggregation while VTE is traditionally treated with drugs targeting the coagulation system. Historically, platelet function has attracted attention primarily in arterial thrombosis, however more recently the role of platelets in VTE has been recognised[70]. Elevated levels of markers of platelet activation, such as P-selectin, are associated with acute VTE[7]; a protein we also identified as one of four candidates associated with VTE in the discovery screen of VEBIOS ER. Furthermore, anti-platelet therapy with acetylic salicylic acid had a protective effect against VTE[71], and reduced the size of venous thrombus linked to inhibition of platelet activation in mice[72]. Our results indicates that CFHR5 has a possible role in platelet activation, which could provide a mechanistic link to the observed association between CFHR5 plasma levels and VTE. Subramaniam et al. showed that C3 and C5 affected platelet activation and tissue factor procoagulant activity by different mechanisms, independent of formation of the terminal complement C5b-C9 complex[52]. C3, but not C5, deficient mice had reduced platelet activation ex vivo, reduced platelet deposition in vivo, and reduced thrombosis incidence ( < 30% *vs.* 80%

in wild type). These data indicate that in VTE C3 has an important role in initial haemostasis, independent of downstream complement proteins[51]. C3a, acting through platelet receptor C3aR, is suggested to have role in the activation of the glycoprotein IIb/IIIa fibrinogen receptor via intraplatelet signalling, and subsequent thrombosis formation[50]. The presence of a C3a receptor on human platelets has been controversial, with several contradictory studies[73,74]. However, recent studies have confirmed the presence of C3aR on human platelets using several independent techniques[50]. Similar to the study of Subramniam et al.[52] Sauter et al. showed C3 deficient mice had prolonged bleeding time, that could be reversed by intravenous administration of C3a peptide[50]. The C3a-C3aR induced intracellular signaling was mediated through the Rap1b activation, where co-stimulation of platelets with C3a-ADP resulted in increased Rap1b activity on top of the platelet stimulation by only ADP. In our study, we observe a similar co-stimulatory effect of CFHR5 on ADP- (and convulxin- or TRAP6-) induced platelet activation. This effect was observed on platelets in plasma, but not on those that were pre-washed, consistent with the effect of CFHR5 on platelet activation being due to its interaction with other complement factors (i.e.,C3) in plasma. Furthermore, in the presence of compstatin, an inhibitor of C3 cleavage and formation of C3a, or anti-C3a antibody, the co-stimulatory effect of CFHR5 was not observed. On basis of these recently published mechanistic findings, our in vitro results indicate that CFHR5 regulation of the alternative pathway of complement activation has a role in C3a mediated platelet activation in thrombosis, providing a potential functional link to its association with acute VTE. As the anti-C3a monoclonal antibody used detects an epitope also present on intact C3, the observed effect could be due to the antibody blocking of C3 activation, rather than neutralisation of C3a, thus alternative mechanisms are possible. In PRP anticoagulated with hirudin the co-stimulatory effect of CFHR5 observed in PRP anticoagulated with citrate does not occur, in line with studies describing an inhibitory effect of hirudin on platelet activation[75]. Hirudin acts through an irreversible strong specific inhibition of thrombin, which is also is an extremely potent platelet activation agonist. Citrate, by chelating calcium ions, reduce the activity of several enzymes in the coagulation system and parts of the complement system (e.g., classical pathway), however not as efficiently as EDTA, and residual thrombin generation can occur, as demonstrated by the difference in background platelet activation observed between hirudin PRP and citrate PRP. One could speculate that CFHR5 promotes the low degree of thrombin generation that occurs when platelets are activated by agonists which would explain why no effect is observed in the presence of hirudin, or in washed platelets where there is no source of thrombin.

The co-stimulatory effect of CFHR5 on platelet activation did not translate into an effect on ADP-induced platelet aggregation. Activated platelets express and secrete proinflammatory and procoagulant factors that could directly drive VTE, independent of platelet aggregation[76]. In the RETROVE study (where we found CFHR5 is associated with VTE and increased thrombin generation) previous studies found no association between VTE and platelet aggregation in response to ADP or epinephrine[46]. Thus, one could speculate that CFHR5 has a role in VTE-linked platelet activation that is independent of platelet aggregation, however the lack of an observable potentiating effect of CFHR5 on platelet aggregation could have methodological explanations (e.g., reflecting the limited sensitivity of light aggregometry), since we found CFHR5 to enhance activation of GPIIb/IIIa, the primary receptor mediating platelet aggregation.

Our study has various strengths and limitations; VEBIOS ER, the discovery cohort, was derived from a single centre, where blood sampling for plasma biobanking was performed in parallel to that for routine tests after initial evaluation (before diagnostic imaging or anticoagulant treatment), thus avoiding bias in inclusion or biobanking. Samples were handled according to standard clinical chemistry lab

routine, thus variations in needle-to-spin-to-freeze time were equivalent between case and control samples. As biobanking was based on the routine sample flow, this increases the feasibility that identified biomarker candidates are suitable for clinical translation into a routine setting. Importantly, we demonstrate an association of CFHR5 with VTE in several independent studies, that include patients in the acute setting, at follow up, and prior to recurrence. One limitation of our study is that we have not analysed a cohort of individuals that were sampled prior to VTE event. Our proteomics and GWAS analyses were mainly conducted in European ancestry populations and should be further investigated in populations of other ancestry origin.

Our screening panel included many candidate proteins (e.g., selected based on GWAS and/or transcriptomics studies) that are poorly understood or uncharacterized, and thus not included in larger commercial panels, such as those available on Olink and Somascan screening platforms. However, using a relatively small custom panel for screening meant we likely failed to comprehensively identify all plasma proteins with currently unknown links to VTE. As our aim was to identify biomarkers associated with acute VTE that were potentially linked to the underlying disease pathogenesis and risk, we prioritised the antibody target HPA059937 (raised against SULF1) for further work on the basis that higher plasma concentrations observed in individuals with a documented increased risk of VTE (e.g., previous VTE in VEBIOS Coagulation) indicated that it could also represent a constitutive and/ or persistent risk factor. It is possible that any of the 3 other candidates that were associated with diagnosis of acute VTE (Fig. 1d, e) could be more informative value in a clinical diagnostic tool for acute VTE than CFHR5. Further studies are needed to investigate this. Some established procoagulant VTE associated proteins included in the screening panel did not pass the significance threshold as VTE associated in VEBIOS ER (Supplementary data 1 [Tab_1]). The control group in this cohort were patients seeking acute medical care with symptoms that initially prompted a diagnostic workup for VTE, and both cases and controls had elevated CRP levels, with no significant difference between them (Table 2), indicating an inflammatory status in both. As F8 and vWF are acute phase reactants, the plasma levels of which increase during inflammation, the lack of association of these proteins with VTE was likely due to elevated levels in both cases and controls. Indeed, using the same assay, we previously reported that both F8 and vWF had a strong association with VTE in VEBIOS Coagulation[23], a study where healthy population-based controls were used.

From a technological perspective, our study demonstrates the need for orthogonal verification of any potential biomarker identified using antibody-based proteomics screening[30,31]. The same caution should be extended to findings generated using other high throughput affinity proteomics technologies vulnerable to non-specific protein binding, such as aptamer-based[77], where missense single nucleotide polymorphisms can affect binding in a manner where a genetic difference drive associations, rather than protein levels (Supplementary information Fig. S1f)[78,79]. Studies comparing different affinity proteomics technologies have found correlations of proteins assayed with two or more platforms to range from highly concordant (Spearman´s $\rho = 0.95$) to inversely correlated ($\rho = -0.48$)[78], highlighting further the need for orthogonal validation of any potential biomarker identified.

The next steps towards the translation of our findings into a clinical setting is to develop and establish standardised methods for quantification, to establish reference intervals and define cut off values with respect to specificity and sensitivity. Current clinical decision rule (CDR) in diagnostic workup of suspected acute VTE is based on age adjusted D-dimer and Wells score. In VEBIOS ER we found that adding CFHR5 to D-dimer increased diagnostic accuracy of acute VTE in the VTE-likely group (Wells score ≥2 for DVT and ≥4 in PE). This group represents the major diagnostic challenge, as an elevated D-dimer is common in several of the conditions associated with increased risk for VTE, e.g., cancer and surgery, both of which are included in Wells score. Therefore, according to current CDR, patients with high clinical probability based on Wells score proceed to diagnostic imaging without prior D-dimer testing[80,81]. Thus, adding CFHR5 concentration to D-dimer in the diagnostic workup could potentially reduce number of negative imaging procedures, to the benefit of patients and health care system. It remains to be established if the incorporation of CFHR5 measurements into clinical decision rules or other scores can improve predictive power. The inclusion of CFHR5 measurements as a diagnostic and/or risk predictive marker in randomized clinical trials of acute VTE and VTE recurrence would be particularly informative, as these are two areas of high clinical relevance in need of improved tools for clinical decision making. Furthermore, while our study indicates that CFHR5 has a functional role in VTE development, further studies are needed to understand the mechanism.

## Methods
The research in this study complies with all relevant ethical regulations. The study protocols have been approved by the regional research ethics committee in Stockholm, Sweden (KI 2010/636-31/4, DNR 2013-2143-31-2, 2015/1294-31/2), the Paris Broussais-HEGP ethics committee in Paris (2002-034), the Institutional Review Board of the Hospital de la Santa Creu i Sant Pau, Spain, the Department of Health and Science, France (2008-880 & 09.576), and the Human Ethics Committee of the Medical University of Vienna (EK237/2004).

### Patients and samples
**Discovery study.** Venous thromboembolism biomarker study (VEBIOS)

VEBIOS is part of a collaboration between Karolinska University Hospital, Karolinska Institute and Royal Institute of Technology (KTH) designed to identify new plasma biomarkers for VTE[23]. VEBIOS comprises two different studies: (i) VEBIOS ER study is a prospective cohort study carried out at the Emergency Room (ER) at the Karolinska University Hospital in Solna, Sweden, between December 2010 and September 2013. All patients admitted with the suspicion of deep vein thrombosis (DVT) in the lower limbs and/or pulmonary embolism (PE), over 18 years old were eligible for the study. Exclusion criteria were patients with on-going anticoagulant treatment, pregnancy, active cancer, short life expectancy or lack of capacity to leave approved consent. A case was defined if (a) VTE was confirmed by diagnostic imaging - compression venous ultrasonography (CUS) in patients with suspected DVT in the lower limbs, and computed tomography pulmonary angiography (CTPA) in patients with suspected PE, and (b) anticoagulant treatment was initiated based on the VTE diagnosis. Patients with no evidence of an acute VTE, (neither by diagnostic imaging nor by Wells clinical criteria) that had a normal D-dimer test[5], were referred as controls in the study. All participants were sampled before any anticoagulant treatment. Whole blood was collected at the same timepoint in citrate or EDTA anticoagulant at the ER and sent within 30 min to the Karolinska University Laboratory. After centrifugation at 2000 x $g$ for 15 min, plasma aliquots were snap frozen and stored at −80 °C. Data collection: For each patient, doctors filled in a questionnaire detailing (1) any provoking factors within 1 month preceding the visit to the ER (2) current health situation, alcohol consumption and smoking habits; (3) family history of VTE (4) ongoing antithrombotic (antiplatelet) treatment and (5) estrogen containing contraceptives and hormone replacement therapy (women only). Information from the ER visit on patient sex, weight and height (when available) along with results from routine laboratory tests e.g., blood count, D-dimer, C-reactive protein (CRP), creatinine, international normalized ratio (INR) and activated partial thromboplastin time (aPTT) were extracted from the medical records. In total, 158 patients were included (52 cases). For the present study, 48 cases were available for analysis and 48 controls were matched by sex, and as closely as

possible by age (mean age difference [years] cases *vs.* controls, women: 0.95, men: 3.65). Clinical characteristics of the sample set is given in Table 1 and 2. (ii) VEBIOS Coagulation study is an on-going case-control study established January 2011 of patients sampled at an outpatient coagulation clinic sampled 1–6 months after discontinuation of 6–12 months anticoagulant treatment after a verified first VTE (DVT to the lower limbs and/or PE), sex and age matched with healthy controls from the population. Patients were between 18 and 70 years of age, free from cancer, severe thrombophilia and pregnancy at inclusion[23]. In the present study, we analysed an extended sample set of VEBIOS Coagulation comprising all available samples; 144 cases and 140 controls (Supplementary data 2 [Tab_1]). Approval for VEBIOS was granted by the regional research ethics committee in Stockholm, Sweden (KI 2010/636-31/4) and all participants gave informed written consent, in accordance with the Declaration of Helsinki.

**Replication cohorts.** The Swedish Karolinska Age Adjusted D-dimer study (DFW-VTE study) includes patients with clinically suspected acute VTE, prospectively recruited from the ER of Karolinska University Hospital in Huddinge, Stockholm, as previously described[44]. The patients were out-patients with low-to-high probability of acute PE or DVT in a lower limb. The study was approved by the regional ethics review board in Stockholm (DNR 2013-2143-31-2), and all participants gave informed written consent, in accordance with the Declaration of Helsinki. For the current study, biobanked plasma aliquots collected at the ER visit were available for a subset of subjects comprising 15 patients with PE, 39 with DVT, and 146 controls where VTE was excluded. Controls were identified based on negative diagnostic imaging, or a low Wells score together with negative D-dimer. Clinical characteristics are described in Supplementary data 2 [Tab_2].

The FARIVE study is a French multicentre case-control study carried out between 2003–2009, as previously described[45]. The study consists of patients with first confirmed VTE (DVT to the lower limbs and/or PE) from 18 years of age, matched to hospital controls with no previous thrombotic event. All patients were free of known or recently discovered cancer at the time of VTE diagnosis. Patients treated for cancer >5 years before the episode without recurrence could be included. The study was approved by the Paris Broussais-HEGP ethics committee in Paris (2002-034) and all participants gave informed written consent, in accordance with the Declaration of Helsinki. In the current study we used a subset of FARIVE samples ($n = 1158$), as previously described[23,82]. From most cases, blood was collected in the first week after diagnosis and during anticoagulant treatment initiation. Clinical characteristics are described in Supplementary data 2 [Tab_3]. Information of sex was obtained from medical records and population registries.

The Riesgo de Enfermedad TROmboembólica VEnosa (RETROVE) study is a prospective case–control study of 400 consecutive patients with VTE (cancer associated thrombosis excluded) and 400 healthy control volunteers. Individuals were recruited at the Hospital de la Santa Creu i Sant Pau of Barcelona (Spain) between 2012 and 2016. Controls were selected according to the age and sex distribution of the Spanish population (2001 census)[83]. All individuals were ≥18 years. All procedures were approved by the Institutional Review Board of the Hospital de la Santa Creu i Sant Pau, and all participants gave informed written consent, in accordance with the Declaration of Helsinki. In the current study, samples from 308 cases and 360 controls were used. Clinical characteristics are described in Supplementary data 2 [Tab_4].

The Marseille Thrombosis Association study (MARTHA) is a population based single centre study, as previously described[82]. Recruitment in MARTHA started in 1994 at Timone Hospital in Marseille (France) and is still ongoing. The cohort from 1994 and 2008, includes a total of 1542 VTE-cases (66% women) that donated blood for further analysis. All patients had a history of a first VTE documented by venography, Doppler ultrasound, angiography and/or ventilation/perfusion lung scan[47]. Ethical approval was granted from the Department of Health and Science, France (2008-880 & 09.576) and all participants gave informed written consent, in accordance with the Declaration of Helsinki. In the current study, proteomics data generated for 1322 sampled MARTHA cases was used. For 669 of the MARTHA cases, follow up data up to 12 years post-event was available and used to analyse risk of recurrent VTE. For a subset of 774 cases data for thrombin generation potential (TGP) was available for the same samples used for CFHR5 measurement[84], which was used to analyse the association between CFHR5 and blood coagulability. Clinical characteristics are described in Supplementary data 2 [Tab_5].

## Analysis of plasma by targeted affinity proteomics
The candidate target selection was based on the following, as previously described:[23] (1) proteins with previous support, or hypothesis of association with VTE and/or intermediate traits, and, (2) availability of corresponding antibodies assessed for target specificity in the Human Protein Atlas (HPA) antibody resource. Based on type of prior support and/or rationale for inclusion, selected candidate targets were grouped into four categories, ranging from ´known/probable´ (A) to ´plausible/hypothetical´ (D):

- Category A: Proteins with an established VTE association, including support from functional analysis, e.g., von Willebrand factor (vWF)[85,86].
- Category B: Targets with: (a) a reported genetic association with VTE, such as single nucleotide polymorphism (SNP) in the gene/locus e.g. Adhesion G protein-coupled receptor B3 (ADGRB3)[87], or (b) an associated with cardiovascular events and/or arterial thrombosis, based on genetic and/or functional data e.g., class IA phosphoinositide 3-kinase β (PI3Kβ)[88].
- Category C: (a) protein encoded by genes we previously identified as having body-wide endothelial cell enriched expression e.g., Myc target 1 (MYCT1)[28], or (b) proteins involved in intermediate traits related to thrombosis, e.g., protein disulphide isomerase A4 (PDIA4)[89], or (c) plasma proteins we previously identified as associated with myocardial infarction or stroke[90].
- Category D: proteins with functions in pathways of relevance to thrombosis or intermediate traits, in the absence of evidence for a direct role e.g., integrin alpha 4 subunit (ITGA4)[91].

Following assessment of available target specific antibodies in the HPA resource, a final panel of 408 target proteins were selected for the discovery screen (from 586 proposed candidates). Target categories for each candidate are given in Supplementary data 1 [Tab_1].

Plasma proteomic profiles in VEBIOS ER were generated using multiplexed suspension bead arrays (SBA) with 756 individual HPA antibodies targeting the 408 proteins (Supplementary data 1 [Tab_1]), using identical design, procedures and methods as in the previous screen of VEBIOS Coagulation[23]. Briefly, paired samples were randomly distributed within the same 96-well area. Two suspension bead arrays composed of 380 antibodies and 4 controls were used to sequentially generate profiles of the 96 samples in parallel. Median fluorescent intensity (MFI) values were obtained from the suspension bead array assay by detecting at least 32 beads per ID and sample with the Flex-Map 3D instrument (xPONENT 4.3, Luminex® Corp). Proteomics profiling was performed in both EDTA and Citrate plasma. The selected target proteins and categories are given in Supplementary data 1 [Tab_1]. A significance threshold of $p < 0.01$ in both EDTA and Citrate plasma was used as selection criteria.

## Immunocapture mass spectrometry (IC-MS)
IC-MS was performed in triplicate of pooled plasma, as previously described[23] using the HPA059937 antibody (Atlas Antibodies) or MAB3845 (R&D systems, clone:390513) and rabbit or mouse immunoglobulin G (rIgG, AB-105-C [R&D] and PMP01X [Biorad], respectively) as respective negative controls. In brief, samples were treated in

10 mM dithiothreitol followed by 50 mM chloroacetamide. Overnight sample digestion at 37 °C using Trypsin, was quenched with 0.5% (v/v) trifluoroacetic acid. Digested samples were analyzed using an Ultimate 3000 RSLC nanosystem (Dionex) coupled to a Q-Exactive HF (Thermo). Resulting raw files were searched using the engine Sequest and Proteome Discoverer platform (PD, v1.4.0.339), Thermo Scientific and Uniprot whole human proteome [20180131, for HPA058337], or MaxQuant[92] (v. 2.1.4.0) against whole human proteome (UniProt,20210811, for MAB3845) using default settings and label-free quantification. An internal database containing the most common proteins detected by IC-MS in plasma was used to calculate Z-scores[32]. A z-score of ≥3, corresponding to a p-value < 0.01 [Confidence level 99%], was used as cut-off.

## Mass spectrometry analysis

**Sample preparation.** Blood plasma was diluted 10 times with 1x PBS, 1% sodium deoxycholate and processed as described in[93] and above in the IC-MS section. The digested samples were desalted using in-house prepared StageTips packed with Empore C18 Bonded Silica matrix (CDS Analytical, CN: 98-0604-0217-3) as described in[94]. Briefly, three layers of octadecyl membrane were placed in 200 μl pipette tips, activated by addition of 100% acetonitrile (ACN) and subsequently equilibrated with 0.1% TFA. Approximately 15 μg of peptides was added to the StageTip membrane and washed twice with 0.1% TFA. The peptides were eluted in two-step elution with 30 μl of solvent containing 80% ACN, 0.1% formic acid (FA). Each desalting step required an in between centrifugation for 2 min at 1000 x g. Desalted peptides were vacuum-dried and stored at −20 °C. Prior to LC-MS/MS analysis samples were dissolved in Solvent A (3% ACN, 0.1% FA) and amount corresponding to ~3 μg of raw plasma subjected to LC-MS/MS analysis.

**DIA-MS analysis.** The LC-MS/MS analysis was performed using an online system of Ultimate 3000 LC (Thermo Scientific) connected to Q Exactive HF (Thermo Scientific) mass spectrometer. First, the amount corresponding to 3 μg of raw plasma was loaded onto a trap column (CN:164535, Thermo Scientific) and washed for 3 min at 7 μl/min with solvent A. Peptides were separated by a 25 cm analytical column (CN:ES802A, Thermo Scientific) following a linear 40-min gradient ranging from 1 to 32% Solvent B (95% ACN, 0.1% FA) at 0.7 μl/min The washout of analytical column was performed with 99% B for 1 min followed by two seesaw gradients from 1 to 99% Solvent B over 4 min. Column was then equilibrated for 9 min with 99% Solvent A. The MS was operated in DIA mode with each cycle comprising of one full MS scan performed at 30,000 resolution (AGC target 3e6, mass range 300–1200 $m/z$ and injection time 105 ms) followed by 30 DIA MS/MS scans with 10 $m/z$ windows with 1 $m/z$ margin ranging 350-1000 $m/z$ at 30,000 resolution (AGC target 1e6, NCE 26, isolation window 12 $m/z$, injection time 55 ms).

## Data processing

Resulting raw files were converted to mzML format using peak picking filter within ProteoWizard provided software tool msConvert (v.3.0.20321)[95]. Resulting mzML files were searched using EncyclopeDIA (v.1.12.31)[96] against a spectral library generated with a deep learning network Prosit, which is integrated into ProteomicsDB (v.1.1)[97]. A whole human proteome (*Homo Sapiens* UniProt ID: #UP000009606, 20,205 entries, accessed 20170918) was used as a background proteome. Finally, the quantification reports were saved, and the protein quantities calculated using top3 method from the peptide intensities[98].

## Western blotting

Recombinant CFHR5 (rCFHR5, 100 ng, R&D, 3845-F5), normal human plasma (NP, 1 μl, George king, pooled from 59 individuals, 31 males, 28 females) and normal human plasma depleted of the 14 most abundant proteins by depletion spin column (Thermo Scientific) (DP, 10 μl) were loaded on SDS PAGE 4-12% (Invitrogen) in non-reducing (without dithiothreitol [DTT], NR) or reducing conditions (with DTT, R). After electrophoresis and transfer onto PVDF membrane, protein was detected using antibodies HPA059937 (original target SULF1), HPA072446 (Atlas Antibodies) and MAB3845 (R&D systems), both targeting CFHR5 (all diluted 1:250). After incubation with horseradish peroxidase (HRP)-coupled goat anti-rabbit (P0448) or anti-mouse (P0447) antibodies (1:2000, both from Dako), bands were detected using chemiluminescence (ECL, Biorad). Molecular weight attributed using PageRuler™ prestained protein ladder (Thermo scientific). WB analysis verified that HPA072446 and MAB3845 bind monomeric and homodimeric form of recombinant CFHR5, and that HPA072446 detects a band in plasma corresponding to CFHR5 (Supplementary information Fig. S2b).

## In-house developed bead based dual binder immunoassays

A Suspension Bead Array (SBA) was built with the capture antibodies raised against human extracellular sulfatase 1-SULF1 (rabbit polyclonal HPA059937) and human CFHR5 (rabbit polyclonal HPA072446 and HPA073894, Atlas Antibodies). 1.76 μg of each antibody was covalently coupled to half million color-coded magnetic beads[99,100]. Bead-coupled rabbit IgG and mouse-IgG and bare beads were included as negative controls. Anti-human SULF1 antibodies, mouse polyclonal ABIN525031 (Abnova), rabbit polyclonals ab172404 (Abcam), PA5-113112 (Thermofisher) HPA054728 and HPA051204 (Atlas antibodies), and mouse monoclonal anti-human CFHR5 (R&D systems, MAB3845, clone:390513) antibody were labelled with biotin and used as detection antibodies in combination with their respective capture antibodies. Citrate plasma samples were thawed on ice and centrifuged for 1 min at 900 x g and diluted in buffer polyvinyl casein 10% rIgG (PVXcas 10% rIgG; polyvinyl alcohol, Sigma Aldrich P8136; polyvinylpyrrolidone, Sigma Aldrich PVP360; Blocker Casein, Thermo 37528), heated at 56 °C for 30 min and incubated with the SBA overnight. The detection antibody was used at 1 μg/mL for 90 min, and streptavidin- R-phycoerythrin (R-PE) conjugate (Life Technologies; SA10044) was used for the fluorescence read out in Luminex platform. The dual binder assays based on HPA059937, HPA072446 or HPA073894 as capture antibodies together with the monoclonal anti-human CFHR5 were used to measure samples in the VEBIOS ER cohort. Median fluorescent intensity (MFI) values were acquired by FlexMap 3D instrument (xPonent 4.3, Luminex® Corp). The polyclonal anti-human CFHR5 HPA072446 capture antibody and the anti-human CFHR5 MAB3845 detection antibody were selected for development of a quantitative assay together with human recombinant CFHR5 (R&D systems; 3845-F5).

## Absolute quantification of CFHR5, C3, C3c and D-dimer in plasma

For CFHR5 quantification, rabbit polyclonal anti-human CFHR5 HPA072446 (Atlas Antibodies) and mouse monoclonal anti-human CFHR5 (R&D systems; MAB3845) antibodies were used in a dual binder assay. Human recombinant CFHR5 (R&D systems; 3845-F5) spiked into chicken plasma (Sigma Aldrich, P3266) was used as a standard. All samples were diluted 1:300 in PVXcas 10% rIgG. For C3 quantification, mouse anti-human C3 and mouse monoclonal anti-human C3 antibodies (Bsi0263, Bsi0190, respectively, Biosystems International) were used in a dual binder assay. Human recombinant C3 (Sigma Aldrich, C2910) spiked into C3 depleted serum (Merck, 234403) was used as a standard. All samples were diluted 1:5000 and analysed as described above. Concentration of complement fragment 3c (C3c) in 20 VEBIOS ER cohort samples (female = 11, male = 9), selected based on low (< 2500 ng/ml) (female = 6, male = 4) or high (3800 ng/ml) (5 of both sex) plasma concentrations of CFHR5 were measured using a commercial sandwich C3c ELISA kit (Nordic Biosite, EKX-JD9XBE-96).

Samples were measured in duplicate and C3c concentration calculated using a standard curve.

D-dimer was quantified by ELISA (D-Di 96 test, product #00947, Asserachrom) following the manufacturer´s instructions. In the DFW-VTE study, D-dimer values were analysed at the Karolinska University Hospital Laboratory in fresh samples sent for routine clinical chemistry analysis, as part of the workup in the ER.

## Statistical analysis

**Plasma protein profiling and quantification.** Median fluorescent intensity (MFI) values were obtained from the suspension bead array assay or dual binder assay were processed as follow: (1) probabilistic quotient normalization as accounting for any potential sample dilution effects[101], and (2) multidimensional MA (M = log ratio; A = mean average, scales) normalization to minimize the difference amount the subgroups of the samples generated by experimental factor as multiple batches[102]. Log-transformation was applied to reduce any skewness in the proteomic data distribution. Quantitative plasma levels for CFHR5 and C3 resulted from extrapolating processed MFI values to standard curves generated from 4 or 5 parametric logistic models[103]. Association of target proteins with VTE was tested using linear regression analysis while adjusting for age and sex, unless stated otherwise. BMI data was lacking for 33 of the 96 patients (7 cases and 26 controls), so we did not adjust for BMI in the discovery analysis, however adjustment with BMI, CRP or the combination of both were applied to CFHR5 quantitative data (Supplementary data 1 [Tab_10]) when data was available. Correlation estimations among protein and/or clinical variables were calculated by Spearman´s rank method. Odds ratio analyses were performed scaling the data to one standard deviation, and significance was obtained from applying generalized linear models. All tests were two-tailed. Analyses were performed using the R statistical computing software (versions R3.2.0-R4.0.5)[104] and data visualizations were created using GraphPad Prism (version 9.1.2).

Association of CFHR5 levels with VTE recurrence in the MARTHA cohort was assessed using a Cox survival model with left truncature at age at sampling. Analysis was adjusted for sex, familial history of VTE, provoked or unprovoked status of the first VTE, age at first VTE, and BMI, and were conducted using the Survival R package. The heterogeneity of the association between CFHR5 and VTE (recurrence) according to specific subgroups was assessed using the Cochran-Mantel-Haenszel statistical test[105].

## Diagnostic Prediction model

Discriminatory accuracy of plasma concentrations of CFHR5 and of D-dimer categorized as positive or negative using age adjusted D-dimer cutoff[35] in the different models was assessed using logistic regression analysis and presented as Area Under the Receiver Operating Characteristics curve (AUC). Statistical analyses were performed using R version 4.0.3. ROC curves for the different biomarker-based risk models based on plasma concentration CFHR5, dichotomized data on D-dimer (positive or negative) and Wells score (VTE likely ( ≥ 2 for DVT and ≥4 for PE) or VTE unlikely) were compared using the function roc.test (Delonge's test) in the RStudio attachment. All tests were two-tailed.

## CFHR5 mRNA expression across human organs

As part of the Human Protein Atlas (HPA, www.proteinatlas.org/), the average TPM value of all individual samples for each human tissue in both the HPA and Genotype-Tissue Expression Project (GTEx) transcriptomics datasets were used to estimate the respective gene expression levels. To be able to combine the datasets into consensus transcript expression levels, a pipeline was set up to normalize the data for all samples. In brief, all TPM values per sample were scaled to a sum of 1 million TPM (denoted pTPM) to compensate for the non-coding transcripts that had been previously removed. Next, all TPM values of

all samples within each data source were normalized separately using Trimmed mean of M values to allow for between-sample comparisons. The resulting normalized transcript expression values (nTPM) were calculated for each gene in each sample. For further details see www.proteinatlas.org/about/assays+annotation – normalization_rna. Analysis of liver single cell transcriptomes and visualisation was performed as part of the HPA single-cell transcriptomics map[38] from data generated in[106].

## CFHR5 mRNA co-expression analysis

Liver bulk RNAseq data analysed in this study was part of the Genotype-Tissue Expression (GTEx) Project (gtexportal.org)[107] (dbGaP Accession phs000424.v7.p2) (n = 226). Pearson correlation coefficients were calculated between *CFHR5* expression values and those for all other mapped protein coding genes across the sample set. Weighted correlation network (WGCNA) analysis: The R package WGCNA was used to perform co-expression network analysis for gene clustering, on log2 expression values. The analysis was performed according to recommendations in the WGCNA manual. Genes with too many missing values were excluded using the goodSamplesGenes() function. The remaining genes were used to cluster the samples, and obvious outlier samples were excluded. Using these genes and samples a soft-thresholding power was selected and the networks were constructed using a minimum module size of 15 and merging threshold of 0.05. Eigengenes were calculates from the resulting clusters and eigengene dendrograms were constructed using the plotEigengeneNetworks() function. Gene ontology and reactome analysis was performed using (http://geneontology.org/docs/go-enrichment-analysis/)[39,40], PFDR values for the top six over represented terms in each category are provided in Supplementary data 1 [Tab_6].

## Genome wide genotyping methods

RETROVE samples were typed with the Illumina Infinium Global Screening Array v2.0 (GSAv2.0) array at the Spanish National Cancer Research Centre in the Human Genotyping lab, a member of CeGen. After genotyping, all monomorfic and unannotated variants were removed as well as polymorphisms with call rate <95% and those whose genotype distributions deviate from Hardy-Weinberg equilibrium at $p < 0.000001$. Remaining polymorphisms were then imputed using the TOPMed r2 reference panel using Eagle v2.4. FARIVE participants were genotyped using the Illumina Infinium Global Screening Array v3.0 (GSAv3.0) at the Centre National de Recherche en Génomique Humaine (CNRGH). A control quality has been performed on individuals and genetic variants using Plink v1.9[108] and the R software v3.6.2[104]. Individuals with at least one of the following criteria were excluded: discordant sex information (n = 20), relatedness individuals (n = 9) identified by pairwise clustering of identity by state distances (IBS), genotyping call rate <99% (n = 5), heterozygosity rate higher/lower than the average rate +/− 3 standard deviation (n = 34). These criteria led to a final sample composed of 1266 individuals. Among the 730,059 genotyped variants, we excluded 145,238 variants with incorrect annotation, 656 variants with deviation from Hardy-Weinberg equilibrium (HWE) in controls using the statistical threshold of $p < 10E-06$, 47,286 variants with a Minor Allele Count (MAC) <20, 1774 variants with a call rate <95%. Finally, 535,105 markers passed the control quality and were used for the imputation. The imputation was performed with Minimac4 using the 1000 Genomes phase 3 version 5 reference panel[109]. MARTHA participants were genotyped with the Illumina bead arrays[110]. Quality control procedures were as previously described[111–113]. Briefly, SNPs showing genotyping call rate <99%, significant ($p < 1E-05$) deviation from Hardy-Weinberg Equilibrium (HWE), with minor allele frequency (MAF) <1% in were filtered out. Individuals were excluded based on the following criteria: (i) genotyping success rates <95%, (ii) close relatedness as detected by pairwise clustering of identity by state distances (IBS) and multi-dimensional scaling (MDS)

using PLINK, (iii) genetic outliers using principal components approach as calculated by EIGENSTRAT. After application of quality control filters, 1525 participants remained for association testing with CFHR5 plasma levels. We imputed genotypes by using MaCH (v.1.0.18.c) and haplotypes from the 1000 Genomes Total European Ancestry (EUR) population (August 2010 release).

## Genome wide association study on CFHR5 plasma levels

All SNPs with imputation quality criterion ($r^2$/info score) >0.30 and minor allele frequency (MAF) >0.01 in each participating cohorts (FARIVE, MARTHA, RETROVE) were tested for association with CFHR5 plasma levels. Associations were assessed using a linear regression model adjusted for age, sex and study-specific principal components derived from genome-wide genotype data. Results obtained in the different contributing cohorts were then meta-analysed through a fixed effect model as implemented in the GWAMA software (v.2.2.2)[114].

A second round of meta-analysis was performed by integrating GWAS summary statistics from 3 additional proteogenomic resources where CFHR5 have been measured. These include 35,559 Icelander participants of the Decode project[115] and 10,708 participants from the Fenland study with CFHR5 plasma levels measured using the Somalogic platform[116] and an additional independent sample of 1178 EPIC participants with CFHR5 measured using the Olink platform[117]. Of note, in a sample of 485 Fenland participants with both measurements CFHR5, the correlation between Somalogic and Olink-derived CFHR5 levels was 0.35 [Supplementary Data Set 2 in ref. 78]. To meta-analyse GWAS results for CFHR5 plasma levels measured with three different techniques (dual binder assay, Somalogic and Olink), we used the Z-score fixed-effect model implemented in the METAL software[118]. In order to conduct the other genetic analyses, normalized regression coefficient and their standard error were derived from Z-scores using the method described in Chauhan et al.[119].

## Shared genetics between CFHR5 plasma levels and VTE risk

Using summary statistics of the latest GWAS on VTE risk[17], we deployed two complementary approaches to assess whether genetics could support a causal role for CFHR5 levels on VTE risk. First, we performed a colocalisation analysis[120] at each locus presenting with SNPs significantly ($p < 5E$-08) associated CFHR5 levels to estimate the posterior probability (PP4) of a shared causal variant between CFHR5 and VTE risk. For this, all SNPs located at +/− 100 kb from a lead variant were investigated through the use of the coloc R package. Second, we conducted a Mendelian Randomization (MR) analysis using, as instrumental variables, the lead SNPs at each locus found genome-wide significantly associated with CFHR5 plasma levels. MR estimates were computed with fixed effect Inverse Variance Weighted method as implemented in the MendelianRandomization R library. Additional MR methodologies expected to be more robust to the presence of pleiotropy including the Weighted Median[121] and Egger[122] methods, were also used.

## Measurement of Thrombin generation potential

Thrombin generation potential (TGP) was measured in fresh frozen platelet poor plasma (PPP) using the Calibrated Automated Thrombogram (CAT®) method according to the manufacturer's instructions. Analyses in MARTHA and RETROVE are as previously described[84,123]. Output parameters recorded were Lagtime (min), the time to the initial generation of thrombin after induction; Endogenous Thrombin Potential (ETP)(nmol/min), equal to the area under the Thrombogram curve; Peak (nmol/L), the maximum amount of thrombin produced after induction by 5pM tissue factor, time to peak (ttPeak, min), time from initiation of assay to peak thrombin generation.

## Validation of structural functionality of recombinant CFHR5

In order to confirm the functionality of the commercial rCFHR5 reagent used, we evaluated its binding capacity to its known partners C-reactive protein (CRP)[53] and properdin[54], using in-house designed ELISA. Flat-bottom microtiter plates (Nunc) were coated with recombinant CRP (1 µg, AG723-M Sigma), properdin (5 µg, Sigma, 341283) or BSA (5 µg, negative control). After blocking with TBS-T containing 3% BSA, serial dilutions of rCFHR5 (3845-F5, R&D systems) or heat denatured rCFHR5 (mock) were added in triplicate and incubated for 1 h at 20 °C. rCFHR5 binding to CRP or properdin was detected with monoclonal mouse anti-human CFHR-5 (MAB3845, R&D systems) followed by HRP-labeled rabbit anti-mouse Ig (1:2000, Dako, P0447). Binding was visualized using o-Phenylenediamine dihydrochloride (OPD) (Sigma, P9187) and absorbance measured at 492 nm following the addition of 3 M HCl as stop solution in microplate photometer (Multiskan FC, Thermo-scientific). rCFHR5 bound CRP and properdin in a dose-dependent manner (Supplementary information Fig. S4a and S4b, respectively). Binding properties were impaired following rCFHR5 heat-denaturation, indicating structure-dependent function (Supplementary information Fig. S4a and S4b). Immunostaining of rCFHR5 by Western blot in non-reducing conditions revealed two bands (Supplementary information Fig. S2b, lane 1 and 7), showing intact capacity for dimerization[124].

## Effect of recombinant CFHR5 on platelet activation

Blood was drawn from healthy volunteers free from any anti-platelet therapy for at least 10 days and anticoagulated with sodium citrate or hirudin. All donors signed informed consent, in accordance with approval of the Human Ethics Committee of the Medical University of Vienna (EK237/2004) and the Declaration of Helsinki. Whole blood was centrifuged (120 x $g$, 20 min, room temperature (RT)) and platelet-rich plasma (PRP) harvested. To obtain isolated platelets, PRP was diluted with PBS and treated with $PGI_2$ (100 ng/ml), centrifuged for 90 sec at 3000 x $g$ and platelets were resuspended in PBS. This step was repeated twice. Platelet-rich plasma (PRP) or isolated platelets were incubated with recombinant CFHR5 (rCFHR5) in PBS (6 µg/ml, 3845-F5, R&D systems) or PBS alone for 10 min before treatment with varying concentrations of ADP (1-5 µM), TRAP-6 (3-15 µM) or convluxin (1-6 ng/ml) for 15 min. Platelets were subsequently incubated with mouse primary antibodies: anti-human CD62P-AF647 (clone:AK4, 304918), anti-human CD63-PE (clone:H5C6,353004) or anti-human CD41/CD61-FITC (clone:PAC-1, 362804) (all Biolegend) for 20 min, washed (PBS then 500 x $g$ for 10 min), then fixed with 1% paraformaldehyde and incubated with Alexa Fluor 647-streptavidin (Jackson Immuno Research, cat:016-600-084) for 20 min. Samples were analysed by flow cytometry (Cytoflex, Beckman Coulter GmbH, Krefeld, Germany) and data processed using Cytexpert v.2.5 (Beckman Coulter GmbH, Krefeld, Germany). In some experiments PRP was incubated for 20 min with 0.25% DMSO, 100 µM compstatin (R&D systems, 2585), PBS, 10 µg/ml anti-C3a/C3a (desArg) (Sigma, clone:K13/16, MABF1978), prior to assay as described above.

## Effect of recombinant CFHR5 on platelet aggregation

PRP was prepared from citrated blood of healthy volunteers by centrifugation (120 x $g$, 20 min, RT). Platelet aggregation was determined by light transmission aggregometry using a Platelet Aggregation Profiler PAP-8 (möLab), defining 0% aggregation as naïve PRP as and 100% aggregation as platelet-poor plasma (PPP), which was obtained by centrifuging PRP in the presence of 0.1 µg/ml PGI2 (1000 x $g$, 90 s, RT). To determine the effect of CFHR5 on platelet aggregation, PRP was monitored in the aggregometer for 1 min before addition of 6 µg/ml (CFHR5 or PBS). After 10 min, PRP was stimulated with 7 µM ADP and aggregation monitored for further 10 min.

## Effect of CFHR5 on HUVECs

Human umbilical vein endothelial cells (HUVEC) were isolated from 2 anonymised umbilical cords collected from Karolinska University Hospital. Ethical approval for HUVEC isolation and subsequent experimentation was granted by Regional ethics committee in

Stockholm (DNR 2015/1294-31/2). HUVEC were cultivated in M199 (M199, Gibco) supplemented with 20% foetal bovine serum (FBS), 100 U/ml penicillin, 0.1 mg/ml streptomycin, 1 µg/ml Hydrocortisone, 1 ng/ml Human Epidermal Growth Factor (all Sigma), and 1.25 µg/ml Amphotericin B (Invitrogen) and seeded on flat-bottom 96 well plates (Falcon, C-treated culture plate). HUVEC were stimulated with rCFHR5 (6 µg/ml), the inflammatory cytokine tumour necrosis factor alpha (TNF, Sigma) (10 ng/ml), both or neither (PBS controls), for 24 h before measurement of thrombin generation or analysis of selected gene expression. HUVEC were blocked with 3% BSA and thrombin formation was initiated in 120 µL reaction mixtures containing human citrated plasma (George King), 4 µM phospholipids, 16.6 mM $Ca^{2+}$ and 2.5 mM fluorogenic substrate (Z-Gly-Gly-Arg-AMC) (Thrombinoscope BV, Diagnostica Stago). All real time thrombin formation assays were run in duplicate. Thrombin generation was quantified using the Thrombinoscope software package (Version 3.0.0.29). For relative gene expression, cDNA was prepared using TaqMan Gene Expression Cells-to-Ct Kit (Ambion), and qPCR performed using Taqman Fast Universal PCR Master Mix and 18 s rRNA reference primer (4319413E), with target primers for *ICAM1* (Hs00164932), *CXCL8* (Hs00174103), *F3* (Hs01076029), *vWF* (Hs01109446), *THBD* (Hs00264920), *TFPI* (Hs00196731) and *PLAT* (Hs00263492) using a StepOnePlus Real-Time PCR System (all Applied Biosystems).

### Availability of materials
Availability of human plasma samples in respective study are subject to limitations in local ethical permits and discretion of respective study PI.

### Reporting summary
Further information on research design is available in the Nature Portfolio Reporting Summary linked to this article.

## Data availability
Source Data are provided with this paper.

The affinity proteomics data for VEBIOS ER generated in this study has been deposited in Figshare, https://doi.org/10.17044/scilifelab.22225942[125]. The mass spectrometry proteomics data generated in this study has been deposited in the ProteomeXchange Consortium via the PRIDE[126] partner repository with the dataset identifier PXD040913. The summary statistic of GWAS data generated in this study is available through GWAS catalogue (GCP ID: GCST90244658).For legal reasons and to minimize the possibility of unintentionally sharing information that can be used to re-identify private information, participant-level datasets containing full information (e.g., including sex, age, BMI, clinical data) cannot be openly shared. A subset of the data that support the findings of this study are available from the corresponding authors upon reasonable request (e.g., for validation). By contacting the corresponding authors (J.O., D.-A.T.), procedures for sharing data, analytic methods, and study materials for reproducing the results or replicating the procedure can be arranged. When submitting an access request, please indicate: [name of PI and host organisation/contact details (including your name and email)/scientific purpose of data access request/commitment to inform when the data has been used in a publication/commitment not to host or share the data outside the requesting organisation/statement of non-commercial use of data].External databases used: Genotype-Tissue Expression (GTEx) Project (dbGaP Accession phs000424.v7.p2), www.gtexportal.org. Human Protein Atlas, human tissue expression data (v.18-20), www.proteinatlas.org. *Homo Sapiens* UniProt ID: #UP000009606; www.uniprot.org/proteomes/UP000009606. TOPMed r2 reference panel using Eagle v.2.4 (https://topmedimpute.readthedocs.io/en/latest/getting-started.html). 1000 Genomes phase 3 version reference panel (http://csg.sph.umich.edu/abecasis/mach/download/1000G.Phase3.v5.html). 1000 Genomes Total European Ancestry (EUR) population (August 2010 release: http://csg.sph.umich.edu/abecasis/mach/download/1000G-2010-08.html; 1000 G.EUR.20100804.tgz). Source data are provided with this paper.

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

## Acknowledgements

We thank research nurses Anna Fahlén and Doris Näslin for assistance with inclusion in the VEBIOS Coagulation study. We are grateful to all the participants of the EPIC-Norfolk study who have been part of the project and to the many members of the study teams at the University of Cambridge who have enabled this research. Data usage: We used data from the Genotype-Tissue Expression (GTEx) Project (gtexportal.org)[37]. The GTEx project was supported by the Office of the Director of the National Institutes of Health, and by NCI, NHGRI, NHLBI, NIDA, NIMH, and NINDS. Figures: Illustrations in Figs. 1a, e, f and 3a–e and Supplementary information Fig. S1, Fig. S8 were created with BioRender.com. The study was supported by grants from Stockholm County Council (Region Stockholm) to J.O. (FoUI-948982, FoUI-949280, FoUI-950776, FoUI-952641, FoUI-952912, FoUI-954024, FoUI-955547), from Familjen Erling Personssons Stiftelse to M.U., Knut och Alice Wallenberg foundation to J.O. (2020.0182, 2020.0241), from Swedish Heart Lung Foundation to L.M.B. (20170759, 20170537), Swedish Research Council (VR) to L.M.B. (2019-01493), HelseNord to JO (HNF1544-20). The Human Protein Atlas (HPA) is funded by The Knut and Alice Wallenberg Foundation. P.-E.M., D.-A.T., J.E., J.-F.D., and D.M.S. acknowledge funding for MARTHA and FARIVE related genetics research programs by the GENMED Laboratory of Excellence on Medical Genomics [ANR-10-LABX-0013], a research program managed by the National Research Agency (ANR) as part of the French Investment for the Future and supported by the French INvestigation Network on Venous Thrombo-Embolism (INNOVTE). MARTHA and FARIVE genetic data analyses benefit from the technical support of the CBiB computing centre of the University of Bordeaux. G.M and D.-A.T. are supported by the EPIDEMIOM-VT Senior Chair from the University of Bordeaux initiative of excellence IdEX. G.M. benefited from the EUR DPH, a PhD program supported within the framework of the PIA3 (Investment for the future). Project reference 17-EURE-0019. The RETROVE study was supported by grants PI12/00612 and PI15/0026. Genotyping of the RETROVE samples was supported by grant PT17/0019, of the PE I + D + i 2013-2016, funded by ISCIII and ERDF. TR acknowledges German Research Foundation (DFG) grants: 25440785 - SFB877, P6 - KFO306, 80750187 - SFB841. The EPIC-Norfolk study (DOI 10.22025/2019.10.105.00004) has received funding from the Medical Research Council (MR/N003284/1 MC-UU_12015/1 and MC_UU_00006/1) and Cancer Research UK (C864/A14136). The genetics work in the EPIC-Norfolk study was funded by the Medical Research Council (MC_PC_13048). Proteomics measurements in EPIC-Norfolk work was supported in part by MRC Rapid Call (MC_PC_21036) and HDRUK Multi-Omics (G107794) grants and the UKRI/NIHR Strategic Priorities Award in Multimorbidity Research for the Multimorbidity Mechanism and Therapeutics Research Collaborative (MR/V033867/1). Proteomics measurements were also supported by a collaboration agreement between the University of Cambridge and Olink. C.L., M.P. are funded by the Medical Research Council (MC_UU_00006/1). MK is supported by Gates Cambridge Trust. SMD is supported by IK2-CX001780. This publication does not represent the views of the Department of Veterans Affairs or the United States Government. S.M.D. receives research support from Renalytix AI and Novo Nordisk, all outside the scope of the current research. D.K. was supported by funding from the Department of Veterans Affairs Office of Research and Development, Million Veteran Program Grant MVP000; Department of Veterans and IK2BX005759-01. The views expressed in this manuscript are those of the authors and do not necessarily represent the views of the National Heart, Lung, and Blood Institute; the National Institutes of Health; or the US

Department of Health and Human Services. M.F. receives research support from Martin Rinds stiftelse and Venforskningsstiftelsen.

## Author contributions

Conceptualisation: J.O., D.-A.T., L.M.B., P.-E.M.. Supervision: J.O., D.-A.T., L.M.B., P.-E.M.. Proteomics data and analyses: M.J.I., L.S.-R., M.G.H., J.M.S., P.M.S., D.K., F.E., D.-A.T., G.M., L.M.B. Experimental data and analyses: M.J.I., L.S.-R., C.N., L.M.B., P.D., W.S., A.A., J.B.K.-P., M.I.-K., P.M.S.. GWAS data and analyses: D.-A.T., G.M., M.G., L.G., F.T., A.B., J.-F.D., M.K., M.P., C.L., S.M.D., A.D.J., N.L.S., D.M.K. Resources and cohorts: M.F., J.A., A.A., M.U., T.R., L.G., M.I.-K., M.M., M.H., D.M.S., A.S., A.M.-P., J.E., J.-F.D., J.M.S.F., J.C.S.A., L.M.B., D.-A.T., P.-E.M., J.O.. Writing—original draft: L.M.B., J.O., L.S.-R., M.J.I., D.-A.T.. Writing—review and editing: All. Visualisation: L.M.B., M.J.I., D.-A.T., G.M., C.N., J.B.K.-P., J.O.. Funding acquisition: J.O., D.-A.T., L.M.B., P.-E.M., M.U., T.R., J.-F.D., J.E., D.M.S., J.M.S.F., S.M.D., M.F., M.K., M.P., C.L.

## Funding

## Competing interests

Dr. Klarin is a scientific advisor and reports consulting fees from Bitterroot Bio, Inc unrelated to the present work. All other authors declare no competing interests.

## Additional information

Maria Jesus Iglesias[1,2,3,37], Laura Sanchez-Rivera[1,37], Manal Ibrahim-Kosta[4], Clément Naudin [1,3], Gaëlle Munsch [5], Louisa Goumidi [4], Maria Farm[6,7], Philip M. Smith[8,9], Florian Thibord[10,11], Julia Barbara Kral-Pointner[12], Mun-Gwan Hong [1], Pierre Suchon[4], Marine Germain[5,13], Waltraud Schrottmaier [12], Philip Dusart[1,3], Anne Boland[14,15], David Kotol[1], Fredrik Edfors[1], Mine Koprulu[16], Maik Pietzner[16,17,18], Claudia Langenberg[16,17,18], Scott M. Damrauer [19,20], Andrew D. Johnson[10,11], Derek M. Klarin [21,22], Nicholas L. Smith[23,24,25], David M. Smadja[26,27], Margareta Holmström[28], Maria Magnusson[6,28,29], Angela Silveira [8], Mathias Uhlén [1], Thomas Renné[30,31,32], Angel Martinez-Perez [33], Joseph Emmerich[34], Jean-Francois Deleuze [14,15,35], Jovan Antovic[6,7], Jose Manuel Soria Fernandez[33], Alice Assinger [12], Jochen M. Schwenk[1], Joan Carles Souto Andres [36], Pierre-Emmanuel Morange [4,38], Lynn Marie Butler [1,3,6,7,38], David-Alexandre Trégouët [5,13,38] ✉ & Jacob Odeberg [1,2,3,8,28,38] ✉

¹Science for Life Laboratory, Department of Protein Science, CBH, KTH Royal Institute of Technology, SE-171 21, Stockholm, Sweden. ²Division of Internal Medicine, University Hospital of North Norway (UNN), PB100, 9038 Tromsø, Norway. ³Translational Vascular Research, Department of Clinical Medicine, UiT The Arctic University of Norway, 9019 Tromsø, Norway. ⁴Aix-Marseille Univ, INSERM, INRAE, C2VN, Laboratory of Haematology, CRB Assistance Publique—Hôpitaux de Marseille, HemoVasc (CRB AP-HM HemoVasc), Marseille, France. ⁵University of Bordeaux, INSERM, Bordeaux Population Health Research Center, UMR 1219ELEANOR, Bordeaux, France. ⁶Department of Molecular Medicine and Surgery, Karolinska Institute, Stockholm, Sweden. ⁷Department of Clinical Chemistry, Karolinska University Hospital, Stockholm, Sweden. ⁸Department of Medicine Solna, Karolinska Institute and Karolinska University Hospital, Stockholm, Sweden. ⁹Theme of Emergency and Reparative Medicine, Karolinska University Hospital, Stockholm, Sweden. ¹⁰Population Sciences Branch, Division of Intramural Research, National Heart, Lung and Blood Institute, Framingham, MA, USA. ¹¹The Framingham Heart Study, Boston University, Framingham, MA, USA. ¹²Center for Physiology and Pharmacology, Institute of Vascular Biology and Thrombosis Research, Medical University of Vienna, Vienna, Austria. ¹³Laboratory of Excellence GENMED (Medical Genomics), Bordeaux, France. ¹⁴Université Paris-Saclay, CEA, Centre National de Recherche en Génomique Humaine (CNRGH), 91057 Evry, France. ¹⁵Laboratory of Excellence GENMED (Medical Genomics), Evry, France. ¹⁶MRC Epidemiology Unit, University of Cambridge School of Clinical Medicine, Institute of Metabolic Science, Cambridge CB2 0QQ, UK. ¹⁷Computational Medicine, Berlin Institute of Health at Charité-Universitätsmedizin Berlin, 10117 Berlin, Germany. ¹⁸Precision Healthcare University Research Institute, Queen Mary University of London, London, UK. ¹⁹Corporal Michael Crescenz VA Medical Center, Philadelphia, PA, USA. ²⁰Department of Surgery and Department of Genetics, Perelman School of Medicine, University of Pennsylvania, Philadelphia, PA, USA. ²¹VA Palo Alto Healthcare System, Palo Alto, CA, USA. ²²Department of Vascular Surgery, Stanford University School of Medicine, Palo Alto, CA, USA. ²³Department of Epidemiology, University of Washington, Seattle, WA, USA. ²⁴Kaiser Permanente Washington Health Research Institute, Seattle, WA, USA. ²⁵Seattle Epidemiologic Research and Information Center, Department of Veterans Affairs Office of Research and Development, Seattle, WA, USA. ²⁶Hematology Department and Biosurgical Research Lab (Carpentier Foundation), European Georges Pompidou Hospital, Assistance Publique Hôpitaux de Paris, 20 rue Leblanc, Paris 75015, France. ²⁷Innovative Therapies in Haemostasis, INSERM, Université de

Paris, 4 avenue de l'Observatoire, Paris 75270, France. [28]Coagulation Unit, Department of Haematology, Karolinska University Hospital, SE-171 76, Stockholm, Sweden. [29]Department of Clinical Science, Intervention and Technology, Karolinska Institute, 171 77, Stockholm, Sweden. [30]Institute for Clinical Chemistry and Laboratory Medicine, University Medical Centre Hamburg-Eppendorf, D-20246 Hamburg, Germany. [31]Center for Thrombosis and Hemostasis (CTH), Johannes Gutenberg University Medical Center, D-, 55131 Mainz, Germany. [32]Irish Centre for Vascular Biology, School of Pharmacy and Biomolecular Sciences, Royal College of Surgeons in Ireland, Dublin 2 D02 YN77, Ireland. [33]Genomics of Complex Diseases Group, Research Institute Hospital de la Santa Creu i Sant Pau. IIB Sant Pau, Barcelona, Spain. [34]Department of vascular medicine, Paris Saint-Joseph Hospital Group, INSERM 1153-CRESS, University of Paris Cité, 185 rue Raymond Losserand, Paris 75674, France. [35]Centre D'Etude du Polymorphisme Humain, Fondation Jean Dausset, Paris, France. [36]Unitat d'Hemostàsia i Trombosi. Hospital de la Santa Creu i Sant Pau and IIB-Sant Pau, Barcelona, Spain. [37]These authors contributed equally: Maria Jesus Iglesias, Laura Sanchez-Rivera. [38]These authors jointly supervised this work: Pierre-Emmanuel Morange, Lynn Marie Butler, David-Alexandre Trégouët, Jacob Odeberg. ✉e-mail: david-alexandre.tregouet@u-bordeaux.fr; jacob1@kth.se

