## [Peer Review File · Nature Communications]

Elevated plasma Complement Factor H Related 5 Protein is associated with venous thromboembolismREVIEWER COMMENTS

Reviewer #1 (Remarks to the Author):

This is an interesting and well written manuscript that attempts to identify plasma biomarkers of VTE risk. It leverages independent human VTE studies from European investigators to identify and support the plasma protein CFHR5 as a candidate biomarker for VTE risk. This is a novel finding and seems to be associated with a modest increased risk for VTE. Although other studies (that are referenced by the authors) have used similar approaches, this study is better powered and demonstrates more validation and replication making it a higher impact study. I have a couple questions for clarity and suggestions that may increase to impact of the paper.

1. One potential limitation of the paper is the strategy of looking at "only" 408 plasma proteins in VTE cases and controls instead of a more broad survey of the plasma proteome. In the discovery phase, this may have lead to false negatives. The authors should further justify their selection of target plasma proteins.

2. Included in the list of targeted plasma proteins are known procoagulant VTE risk proteins such as VWF and FVIII. It would be helpful to understand how these proteins performed in the study and discuss why there were not significant differences in VTE cases and controls which is contrary to previous literature and perhaps unexpected.

3. In the GWAS for plasma CFHR5, the authors state that the top SNPs explain ~2% of the variance of plasma CFHR5 but were not associated with VTE when looking in two independent VTE cohorts (FARIVE and MARTHA?). I think a colocalization study in a much larger VTE cohort like the full INVENT or MVP cohorts (UK Biobank?) would have a higher probability of finding association and would strengthen the findings in the manuscript. I realize that the authors are not attempting to support causality by using MR here but I do think a better powered colocalization study would increase the impact.

Reviewer #2 (Remarks to the Author):

I found the paper by Sanchez-Rivera et al. quite interesting. The relationship between complement dysregulation and thrombosis is not very well understood despite genetic and clinical evidence implying complement as an important factor in thrombotic disease. In this regard, the Sanchez-Rivera paper provides an important addition to this ongoing scientific debate.

The authors describe the discovery of CHFR5 as a novel biomarker for venous thrombosis (VTE). The authors used a targeted multiplexed approach in their discovery cohorts and subsequently validated CFHR5 in independent cohorts. The main strength of the paper is the thorough validation of their approaches to detect CFHR5, and their subsequent replication of CHFR5 as a VTE biomarker in several independent cohorts.

On the other hand, the evidence that CFHR5 is a causal risk factor for VTE are suggestive at best and their functional studies seem quite preliminary. The authors also include results from a cohort of hospitalized Covid-19 patients. These studies do not seem connected to the rest of the paper, and I'm not sure if they add any value to the manuscript.

Specific points:

1. The authors state in the abstract and discussion that CFHR5 is a potential diagnostic plasma biomarker for VTE and also discuss the limitations of D-dimer as the only VTE biomarker in routine clinical use. It would therefore be interesting to see how CFHR5 performs as a biomarker for VTE compared to D-dimer. Currently this comparison is only shown as a correlation analysis, but the authors should also perform a ROC curve analysis to show the sensitivity and specificity of CFHR5 for VTE and how it compares to D-dimer. Moreover, does a combined assessment of D-dimer and

CFHR5 improve VTE prediction?

2. In figure 4, the authors report a genetic association study that identifies a novel CFHR5 pQTL. The identification of a genetic marker for CFHR5 plasma levels offers a possibility to assess a causal role of CFHR5 in the outcome studied. The lead SNP is highly significant for CFHR5 plasma levels, yet does not show any association with VTE. Although the genetics of the CFHR1-5 locus is quite complex, this result seems at odds with the author's conclusion that CFHR5 is a causal factor for VTE and should be discussed in more detail. Moreover, does the CFHR5 pQTL associate with other parameters such as D-dimer levels and other biomarkers?

3. Figure 5 demonstrates a moderate potentiating effect of exogenously added CHFR5 on platelet activation in platelet-rich plasma but not washed platelets. These results are interesting but feel somewhat preliminary. The mechanism is unclear, and the calcium chelator citrate is used as anticoagulant, which might interfere with complement activation under the conditions studied.

a. The results are based on flow cytometric measurements of platelet activation markers. These results should be complemented by a functional assessment of platelet activation, e.g., aggregometry.

b. The authors should provide additional experimental results to be able to say something about the mechanism by which CFHR5 potentiates platelet activation. A proposed function of CHFR5 is that it augments complement activation by competing with factor H for binding to host surfaces. Hence, is the effect of CFHR5 on platelet activation complement dependent? Does CFHR5 still augment platelet activation in the presence of complement inhibitors, such as a C3 inhibitor?

c. With regards to the potential mechanism, the authors propose that increased generation of the anaphylatoxin C3a would lead to platelet activation. It's important to note that the expression of anaphylatoxin receptors on platelets is controversial. Other studies (see e.g. PMID: 3283237) have found no evidence of C3a receptor expression in human platelets, which should be acknowledged in the manuscript.

d. The anticoagulant citrate chelates calcium and magnesium ions that are required for complement activation. Can the authors use an alternative anticoagulant for the platelet studies, such as hirudin, that does not interfere with complement?

e. CHFR5 was obtained from a commercial source. How was CFHR5 characterized, and is it functional? Moreover, platelets are delicate to handle and quite prone to unspecific activation. A mock recombinant protein from the same source should be used as control to exclude any unspecific effects in the CFHR5 stimulation experiments.

f. If the authors believe that CHFR5 exerts prothrombotic effect via increased platelet activation this should be discussed in relation to the association of CHFR with venous thrombosis. Venous thrombosis has historically been more associated with plasma coagulation than platelets. In contrast to arterial thrombosis, VTE is treated with anticoagulants and not anti-platelet drugs. This should be discussed in more detail.

4. The last section of the manuscript and figure 6 report on CHFR5 plasma levels in hospitalized Covid-19 patients. Even though the rest of the manuscript focuses on VTE, nothing is mentioned about thrombotic events or coagulation status in these patients. Currently it is unclear to the reviewer what these results add to the manuscript. At least, the authors should provide information on thrombotic events and coagulation biomarkers in these patients and their relationship to CFHR5 levels.

Reviewer #3 (Remarks to the Author):

Review of Elevated plasma Complement Factor H Regulating Protein 5 is associated with venous thromboembolism and COVID-19 severity

Reviewer: Brent Richards

This is an impressive paper. The authors have done a formidable job provided orthogonal lines of evidence indicating that CFHR5 is a risk factor for VTE. The use of replication cohorts is greatly appreciated. The manuscript is clear and well-written. The figures are generally excellent. All in all, this is a solid contribution to the field.

I will attempt to limit my comments within the realm of genetic epidemiology, since I lack expertise in VTE, protein assays or platelet activation. I'd like to also state that this paper was reviewed by both me and one of my PhD students independently. Many of our comments were overlapping (which either suggests groupthink or potential problems to address).

Major Comments:

Genetics

-CHFR5 is measured in both Olink Explore and Somascan assays. There are cis-pQTLs identified for this protein. In the deCODE pQTL paper (Ferkingstad et al, NG, 2021), there are cis-pQTLs for CFHR1, 2, 3, 4 and 5. There are 14 cis pQTLs for CFHR5 in this paper. This represents an opportunity to better understand if the ci-pQTLs for CFHR5 are unique to CFHR5, or whether they are pleiotropic.

-The authors should undertake Mendelian randomization studies of CFHR5 levels, given that they have a cis-pQTL. They should also assay if their associated lead SNP is associate with other CFHR protein levels. This can be done in the deCODE paper data, I listed above. The benefit of the MR study is that it would allow for a tremendous reduction in potential confounding and they could assess the CFHR5 effect on VTE using the largest GWAS meta-analysis of VTE, providing a much larger sample size than the FARIVE and RETROVE studies.

-They should be particularly weary of potential horizontal pleiotropy in their MR study as I suspect that the lead SNP may also be driving association with other CFHR proteins.

-In the GWAS, which is really helpful, the authors identify a hit near CFHR5. This is compelling and strongly suggests that their assay is measuring a CFHR protein. However, the region is gene rich and seems to have several genes that are also CFHR genes. Are these gene duplications? Is it possible that their assay is measuring other CFHR proteins?

-In the Biobanque Quebecois COVID-19 cohort (bqc19.ca), we have generated SomaScan v4 data on hundreds of individuals with detailed COVID-19 outcomes. Many samples are longitudinal. The authors can access this data through its data application process and might be helpful, as it will allow for testing of other CFHR proteins. They can also apply for access to the samples if helpful.

-The authors state that they retained SNPs with an imputation quality criterion greater than 0.3. Do they mean the info score? If so, please state this.

-How did the authors control for ancestry and relatedness in their GWAS? Why was a random effects model used in GWAMA? Personally, I would used fixed effects.

-The authors should clarify if their GWAS results are available for sharing. This is a community standard and all summary statistics should be shared through the GWAS Catalog. Could they confirm this will be done upon publication?

- Given that ancestry is a known risk factor for VTE, it would be worth motioning the ancestral background of five cohorts (and samples used for GWASs).

Epidemiology

-Table 2 presents the Odds of VTE (not the risk). More importantly, it might be relevant to consider meta-analysis of these findings across cohorts. Further, in the legend to this table I suggest to provide the covariates that were used in the analysis

-The narrative of the story could be improved. The authors start by outlining how diagnosis and prediction of VTE would be helpful, but then provide little data to demonstrate that their findings solve this problem. (No AUROC, no AUPRC, no estimates of precision etc...) To this end, such metrics should be reported.

-There is no reference for the "Respiratory Index". While this score seems reasonable to me at face value, is this a score accepted by the community? If not, why not use another scoring system like the WHO COVID severity score?

-While I lack expertise in measurement of proteins, is it not somewhat concerning that SULF1 is detected by the discovery antibodies? Would it not be possible to assess publicly available data from different assays to see if the results are concordant? For example, if the SomaScan assay also found an association between CFHR5 and VTE, but a lack of association of SULF1 with VTE, that might be helpful

-In general, the epidemiology work done requires further thought. The covariates used to calculate the ORs should be listed in the text and table, comparing CFHR5 levels between cases and controls. If no covariates were used, why aren't the authors worried about confounding? For example, if CFHR5 levels are associated with D-dimer levels in cases, could this not influence its association with VTE via confounding?

-In another example, on the MARTHA cohort, adjustments were made for family history of VTE. Why? Why adjust for provoked, or unprovoked VTE?

-In another example: In the "ANALYSIS OF PLASMA BY TARGETED PROTEOMICS" section, the authors did not adjust for BMI (lines 258–259). However, in other sections, such as "Analysis of CFHR5 in COVID-19 patients in the COMMUNITY study" (line 312) and "CFHR5 and risk of recurrent VTE" (line 581), they adjusted for BMI. I recommend that they consistently adjust for BMI and other covariables. If it was not possible for some analyses due to lack of data, they should clearly state that.

- In the "CFHR5 is associated with VTE independent of C3" section (starting from line 534), they used 0.05 as an absolute cut-off to declare CFHR5's dependent/independent association with C3, but I find this misleading. The p-value differences is minor (0.032 vs 0.0645), and $p = 0.0645$ does not support independence. It just means that it was not statistically significant. So I suggest wording this section more carefully, especially given the relatively small sample size. Besides, since the authors indicated that CFHR5 is associated with C3 in other sections (e.g., Fig2. protein-protein interaction analysis, line 605–606), the author should elaborate on these seemingly contradicting claims.

Minor:

-This is a very awkward phrase to read in English: we found CFHR5 levels at baseline were associated with short-time prognosis of disease severity. What does "short-time prognosis of disease severity" mean? I think you mean maximum level of respiratory support. If so, why not just say that?

-Figures. I suggest to have less dense figures. For example, Fig 1H iii presents data that is very small. Why not have more figures and let the reader be able to see them?

-Please provide QC procedures for genotyping MARTHA, rather than a reference for this. Why were patients with SLE removed?

Minors:

-Line 163–164 (and other sections mentioning Well's criteria): For Well's criteria, the authors should clarify the cut-off they used.

-Line 176–177: Given matching cases and controls can greatly affect the results, especially when the sample size is relatively small, I recommend more elaboration on how the authors matched cases and controls. For TableS1 Tab 1–S5, statistical testing is needed to evaluate baseline differences between cases and controls (adding a p-value column). It would also be helpful if the authors provided an additional table listing all five cohorts' characteristics, comparing their differences.

- Line 201: "All participants were free from cancer." Is this correct? Or does this mean that they were not in the active treatment of cancer?
- Line 312, 331 (and other sections mentioning computational tools): I recommend the authors include version information.
- In the Methods section, the authors should clearly state the cut-off for each of the statistical testing performed, including the one for FDR.
- Line 472–474: I suggest the authors include these data (quantitative signal) in Supplementary materials.
- Line 518: Since FDR stands for false discovery rate (e.g., 5%), "FDR =2.4E-16" may be inappropriate. It should instead be written as FDR-adjusted P-value (PFDR) or q-value.
- Line 582–584: P-values for subgroup analyses should be shown next to HR and its confidence intervals. As none of them reached a conventional statistical threshold of 0.05, the authors should be careful in interpreting these associations. E.g., I found the term "this association is driven by" too strong since the primary analysis was not statistically significant.
- Line 587: Typo. Testhomogeneity -> test of homogeneity
- Line 634: Was CFHR5 associated with any clinical measurements/features/symptoms that suggest thrombosis formation/VTE events in these COVID-19 patients? I understand sample size can be an issue, and if there are not sufficient data available, the authors do not have to address this.
- Line 748: The reference is not properly formatted.

Reviewer #4 (Remarks to the Author):

VTE is an important disorder with high morbidities and mortalities. Although several risk factors and mediators have been established its pathogenesis is still not completely understood. In the present study the authors provide several lines of evidence supporting an important role of CFHR5 in this disorder. The finding was first detected in a discovery proteomic study and thereafter confirmed in five replication cohorts including a total of 1137 patients and 1272 controls. A role of CFHR5 was also supported by GWAS analysis of 2967 individual. They also showed that recombinant CFHR5 promote platelet activation in vitro. Their findings are novel and the use of several replication cohorts and in vitro studies clearly strengthen their findings. Their study have also some important limitations.

1. Although mentioned is some of the study studies, the author should in all studies the relationship of CFHR5 to provoked and spontaneous VTE and to DVT and PE, separately.
2. In some sub-studies they perform adjustment for other risk factor (e.g., BMI, aged, gender, CRP, co-morbidities), but this should be consistently be preformed in all sub-analyses.
3. In the statistical approach they could use multivariable Cox proportional hazards models to calculate HR. In fact, I cannot find a thorough description of the statistical methods.
4. As mentioned by the authors, CFHR5 will inhibit FH and thereby enhance activity of the alternative complement pathway. Have the authors measured the activity in the alternative and terminal pathway for example by measuring C3bc, detecting the C3b, iC3b and C3c fragments, the C3 convertase C3bBbP and C5b-9/TCC at least in a subgroup of patients.
5. The relevance of platelet activation for VTE is questionable. It would have strengthened their findings if they could show a pro-coagulant effect of recombinant FHR5 on endothelial cells.
6. The relevance of their COVID-19 findings is questionable. Several studies have shown that a large number of factors are associated with the degree of respiratory failure including TCC and in the present study the authors present unadjusted analyses. They discuss the importance of VTE in the pathogenesis of severe respiratory failure, but they present no data on either PA or VTE in the included patients. This part of the study should either be deleted or markedly improved.

Reviewer #1 (Remarks to the Author):

This is an interesting and well written manuscript that attempts to identify plasma biomarkers of VTE risk. It leverages independent human VTE studies from European investigators to identify and support the plasma protein CFHR5 as a candidate biomarker for VTE risk. This is a novel finding and seems to be associated with a modest increased risk for VTE. Although other studies (that are referenced by the authors) have used similar approaches, this study is better powered and demonstrates more validation and replication making it a higher impact study. I have a couple questions for clarity and suggestions that may increase to impact of the paper.

1. One potential limitation of the paper is the strategy of looking at "only" 408 plasma proteins in VTE cases and controls instead of a more broad survey of the plasma proteome. In the discovery phase, this may have lead to false negatives. The authors should further justify their selection of target plasma proteins.

The reviewer highlights an important conceptual limitation of our study; in that we screen only 408 proteins out of a total of ~17,700 in the human proteome. These proteins were selected based on a known, assumed, hypothesized or potential role in thrombosis, cardiovascular disease and/or relevant intermediate traits, using various established sources or novel criteria for inclusion (e.g., genetic associations, or proteins that we previously identified as having enriched expression in cell types with potential disease relevance, such as endothelial cells). As the flexibility of the suspension bead array technology and access to the Human Protein Atlas antibody resource allowed us to design a totally custom panel, the final candidate list contained many poorly understood or uncharacterized proteins, as well as those with low abundance in plasma. Whilst the latest versions of Olink and SomaScan platforms now include assays for surveying 3000-7000 proteins, they rely on the measurement of pre-defined panels of proteins with known functions in biological pathways and pathophysiological processes and are thus biased against the inclusion of such proteins. Thus, each approach has advantages and limitations.

We have added a more detailed description of selection criteria in the methods [lines **722-743**]:

'The candidate target selection was based on the following, as previously described (Bruzelius et al., 2016a): (1) proteins with previous support or hypothesis of association with VTE and/or intermediate traits and (2) availability of corresponding antibodies assessed for target specificity in the Human Protein Atlas (HPA) antibody resource. Based on type of prior support and/or rationale for inclusion, selected candidate targets were grouped into four categories, ranging from 'known/probable' (A) to 'plausible/hypothetical' (D):

- *Category A: Proteins with an established VTE association, including support from functional analysis, e.g., von Willebrand factor (VWF) (Smith et al., 2010; Tsai et al., 2002).*
- *Category B: Targets with: (a) a reported genetic association with VTE, such as single nucleotide polymorphism (SNP) in the gene/locus e.g. Adhesion G protein-coupled receptor B3 (ADGRB3) (Antoni et al., 2010), or (b) an associated with cardiovascular events and/or arterial thrombosis, based on genetic and/or functional data e.g., class IA phosphoinositide 3-kinase β (PI3K β) (Laurent et al., 2015).*
- *Category C: (a) protein encoded by genes we previously identified as having body-wide endothelial cell enriched expression e.g., Myc target 1 (MYCT1) (Butler et al., 2016), or (b) proteins involved in intermediate traits related to thrombosis, e.g., protein disulphide isomerase A4 (PDIA4) (Cho, 2013), or (c) plasma proteins we previously identified as associated with myocardial infarction or stroke (Matic et al., 2018).*
- *Category D: proteins with functions in pathways of relevance to thrombosis or intermediate traits, in the absence of evidence for a direct role e.g., integrin alpha 4 subunit (ITGA4) (Nalls et al., 2011).*

Following assessment of available target specific antibodies in the HPA resource, a final panel of 408 target proteins were selected for the discovery screen (from 586 proposed candidates). Target categories for each candidate are given in Table S1, Tab 1'

As stated in the text above, we have also added the candidate classifications in Table S1, Tab 1.

We have acknowledged the limitations associated with using a comparatively narrow candidate screening panel in the discussion [lines **584-588**]:

Our screening panel included many novel candidate proteins (e.g., selected based on GWAS and/or transcriptomics studies) that are poorly understood or uncharacterized, and thus not included in larger commercial panels, such as those available on Olink and Somascan screening platforms. However, using a relatively small custom panel for screening meant we likely failed to comprehensively identify all plasma proteins with currently unknown links to VTE'

2. Included in the list of targeted plasma proteins are known procoagulant VTE risk proteins such as vWF and FVIII. It would be helpful to understand how these proteins performed in the study and discuss why there were not significant differences in VTE cases and controls which is contrary to previous literature and perhaps unexpected.

The reviewer raises an interesting and relevant point. In the current study, the association of vWF with acute VTE fell just below statistical significance in citrate plasma ($p=0.078$) and just above in EDTA plasma ($p=4.77E-04$). F8 was not associated with acute VTE in either citrate or EDTA plasma ($p=0.582$ and $p=0.914$, respectively). In contrast to many existing studies, the control population in the current study was composed of patients presenting to the emergency room with symptoms that prompted a diagnostic work-up for suspected VTE (which was subsequently excluded). Therefore, this population could not be considered healthy - indeed, both cases and controls had elevated CRP levels (with no significant difference between them [$p=0.1$, Table 1]), indicating a baseline level of inflammation in both. F8 and vWF are acute phase reactants, which can also be elevated in response to (non-VTE associated) inflammation or cardiovascular risk factors, such as obesity (Conlan et al., 1993; Terraube et al., 2010). Thus, in VEBIOS ER the elevated levels of these proteins in the control group could underlie the modest/lack of association between vWF/F8 and VTE. Indeed, these data illustrate the limitations of such biomarkers for VTE diagnosis in the acute setting.

In contrast, in our previously published study (Bruzelius et al., 2016b), where we performed a proteomics analysis of plasma collected after ending anticoagulation treatment following a first time VTE ('VEBIOS coagulation') vs. healthy population-based controls, both vWF and F8 were significantly associated with disease ($p<0.001$). Importantly, the protocols and antibody reagents used in both studies were identical, providing further support that the control population to which cases are compared is of key importance.

To address these points, we have added the following text to the discussion [lines **596-605**]:

Some established procoagulant VTE associated proteins included in the screening panel did not pass the significance threshold as VTE associated in VEBIOS ER (Table S1, Tab 1). The control group in this cohort were patients seeking acute medical care with symptoms that initially prompted a diagnostic workup for VTE, and both cases and controls had elevated CRP levels, with no significant difference between them (Table 1), indicating an inflammatory status in both. As F8 and vWF are acute phase reactants, the plasma levels of which increase during inflammation, the lack of association of these proteins with VTE was likely due to elevated levels both cases and controls. Indeed, using the same assay, we previously reported that both F8 and vWF had a strong association with VTE in VEBIOS Coagulation (Bruzelius et al., 2016a), a study where healthy population-based controls were used'

In addition, we have included more detailed data for all proteins screened in both citrate and EDTA plasma, including b values, fold changes and association significances (Table S1, Tab 1), so the reader can access this information for any specific protein of interest in the panel.

3. In the GWAS for plasma CFHR5, the authors state that the top SNPs explain ~2% of the variance of plasma CFHR5 but were not associated with VTE when looking in two independent VTE cohorts (FARIVE and MARTHA?). I think a colocalization study in a much larger VTE cohort like the full INVENT or MVP cohorts (UK Biobank?) would have a higher probability of finding association and would strengthen the findings in the manuscript. I realize that the authors are not attempting to support causality by using MR here but I do think a better powered colocalization study would increase the impact.

We thank the reviewer for their comment. This prompted us to interrogate recently released proteogenomics database where GWAS on CFHR5 levels have been conducted, and to perform a meta-analysis of our own CFHR5 GWAS, based on ~3000 individuals, together with GWAS results obtained in the Omicscience (Pietzner et al., 2021a), EPIC (Koprulu, 2022) and Decode (Feringstad et al., 2021) genomic initiatives. This increased the sample size of our GWAS for CFHR5 levels to include ~52,000 individuals.

We identified 6 loci that were significantly associated with CFHR5 levels with genome-wide significance ($5E-08$) in the extended meta GWAS analysis (see Figure 1 for review). One of these loci was the *CFHR1-CFHR5* gene cluster. The lead SNP at this locus was the same as that we initially reported in the previous version of this manuscript, where the GWAS was conducted only in ~3000 samples and CFHR5 was measured using our dual binder assay. Strong linkage disequilibrium holds at this locus covering more than 10Mb and extending from CFHR1 to CFHR5 (see Figure 2 for review). Importantly, in the newly analysed cohorts included in the extended meta-analysis, plasma CFHR5 was measured using two independent techniques (namely, Somalogic and Olink). This consistency between analysis platforms, together with the homogeneity of the effects detected in the extended meta-analysis [Table S1, Tab 12], provide additional support for our original results and conclusions. Of note, to address the fact that different techniques were used to measure CFHR5, the meta-analysis of GWAS results was conducted using the Z-score fixed-effect model as recommended in Chauhan et al. (Chauhan et al., 2015b).

Figure 1 for review [new Figure 4A in revised manuscript]. Manhattan plot of the meta-analysis on INVENT-MVP consortium resources [17] identifying 6 loci associated with CHFR5 plasma levels and VTE risk.

Figure 2 for review [new Figure 4B in revised manuscript]. Regional association plot around the lead SNP at CFHR1-CFHR5 locus.

We then went on to assess the impact on VTE risk of the lead polymorphisms that were detected in the extended GWAS on CFHR5 and to conduct Colocalization and Mendelian Randomization (MR) analyses using the latest GWAS findings on VTE obtained by the INVENT-MVP consortium (Thibord et al., 2022a). Five of the six lead SNPs at the identified loci with VTE risk, showed marginal association ($p=4E-04$ to $p=0.026$) with VTE. However, for only two of them (*JMJD1C* rs7916868 and *DNAH10* rs7133378), the pattern of association with CFHR5 and with VTE risk was consistent. Of note, these two polymorphisms were those showing the stronger (but still moderate) probability of colocalization (PP4 ~40%) [Table S1, Tab 11]. As a consequence, the MR analysis does not provide strong support for causality between CFHR5 levels and VTE risk [Table S1, Tab 13]. The MR analysis was performed using genetic factors for VTE as a whole without distinguishing pulmonary embolism and deep vein thrombosis, provoked and unprovoked events. Such heterogeneity may have impacted our MR analysis.

Data has been added to Table S1, Tab 11, 12 and 13.

We have added the following text to the results section [lines 374-389]:

*A second round of meta-analysis, integrating GWAS summary statistics from 3 additional proteogenomic resources where CFHR5 was measured with different assays [see methods], totalling ~50,200 individuals, confirmed the association of this locus with CFHR5 levels. Interestingly, the rs10737681 was also identified as the lead SNP at this locus in the extended meta-analysis ($\beta = 0.28 \pm 0.01$, $p=2.94E-396$) (Table S1, Tab 12). Strong linkage disequilibrium holds at the locus covering more than 10Mb and extending from CFHR1 to CFHR5 (Figure 4B). The extended meta-analysis identified 5 additional independent loci associated with CFHR5 levels: *HNF1A* (rs2393776, $p=1.48E-21$) on 12q24.31, *JMJD1C* (rs7916868, $p=4.61E-12$) on 10q21.3, *TRIB1* (rs28601761, $p=4.39E-09$) on 8q24.13, *DNAH10* (rs7133378, $p=2.43E-08$) also on 12q24.31 and *HNF4A* (rs1800961, $p=4.97E-08$) on 20q13.12 (Figure 4A and Table S1 Tab 12). All of the lead SNPs at these loci, except *HNF1A* rs2393776 ($p=0.17$), demonstrated marginal ($p<0.05$) association with VTE risk (Table S1, Tab 11). However, only two, *JMJD1C*_rs7916868 and *DNAH10*_rs7133378, showed patterns of association with VTE that*

are compatible with the association of increased CFHR5 levels with VTE risk. This explains why MR analyses are not supportive for a causal association between increased CFHR5 levels and VTE (Table S1, Tab 13)

We have added the following text to the discussion section [lines **483-503**]:

The CFHR5 locus maps to chromosome 1q31.3 at one end, a gene cluster that spans approximately 350 kb including (in order from CFHR5) the CFHR2, CFHR4, CFHR1, CFHR3, and CFH loci. The rs10737681 we identified with genome wide association with plasma CFHR5 level maps between the CFHR4 and CFHR1 genes. Of note, the CFHR2 locus has just been identified as a novel susceptibility locus for VTE in the recently published international effort on VTE genetics (Thibord et al., 2022b). The lead SNP at this locus is rs143410348, which is in moderate LD with the rs10737681 ($r^2 \sim 0.40$ in European population (Machiela and Chanock, 2015)), which here we found associated with CFHR5 plasma levels. In a combined meta-analysis of 37,770 individuals from 3 cohorts (EPIC/FARIVE/Omicscience) where it was imputed, rs143410348 was less strongly associated with CFHR5 levels than the lead rs10737681 ($\beta = -0.15$, $p = 1.9E-32$ vs $\beta = -0.27$, $p = 2.1E-95$). Altogether, the observations from the GWAS analyses emphasize the need for a deeper investigation of the genetic architecture of the CFHR1/CFHR4/CFHR2/CFHR5 locus with respect to CFHR5 levels and VTE risk. Five additional candidate loci were identified as participating to the genetic regulation of CFHR5 plasma levels: DNAH10, HNF1A, HNF4A, JMJD1C and TRIB1. All 5 loci have been reported to associate with various lipids traits (see GWAS catalogue (Sollis et al., 2022)) and 3 (HNF1A, HNF4A and TRIB1) have been reported to also associate with liver enzymes. DNAH10 is also known to be a locus involved in white & red blood cell biology (Chen et al., 2020) while the JMJD1C is a locus linked to platelet biology (Eicher et al., 2016; Johnson et al., 2010). Most of these traits are well known risk factors for VTE, and this may suggest that the association of CFHR5 levels with VTE risk implies many additional biological players with pleiotropic effects. This may explain why the MR analyses did not provide causal evidence for a link between CFHR5 levels and VTE risk

We have added the following text to the methods section [lines **954-965**]:

A second round of meta-analysis was performed by integrating GWAS summary statistics from 3 additional proteogenomic resources where CFHR5 have been measured. These include 35,559 Icelander participants of the Decode project (Feringstad et al., 2021) and 10,708 participants from the Fenland study with CFHR5 plasma levels measured using the Somalogic platform (Pietzner et al., 2021a) and an additional independent sample of 1,178 EPIC participants with CFHR5 measured using the Olink platform (Koprulu, 2022). Of note, in a sample of 485 Fenland participants with both measurements CFHR5, the correlation between Somalogic and Olink-derived CFHR5 levels was 0.35 [Supplementary Data Set 2 in (Pietzner et al., 2021b)]. To meta-analyse GWAS results for CFHR5 plasma levels measured with three different techniques, (dual binder assay, Somalogic and Olink), we used the Z-score fixed-effect model implemented in the METAL software (Willer et al., 2010). In order to conduct the other genetic analyses, normalized regression coefficient and their standard error were derived from Z-scores using the method described in Chauhan et al (Chauhan et al., 2015a)

Reviewer #2 (Remarks to the Author):

I found the paper by Sanchez-Rivera et al. quite interesting. The relationship between complement dysregulation and thrombosis is not very well understood despite genetic and clinical evidence implying complement as an important factor in thrombotic disease. In this regard, the Sanchez-Rivera paper provides an important addition to this ongoing scientific debate. The authors describe the discovery of CFHR5 as a novel biomarker for venous thrombosis (VTE). The authors used a targeted multiplexed approach in their discovery cohorts and subsequently validated CFHR5 in independent cohorts. The main strength of the paper is the thorough validation of their approaches to detect CFHR5, and their subsequent replication of CFHR5 as a VTE biomarker in several independent cohorts.

On the other hand, the evidence that CFHR5 is a causal risk factor for VTE are suggestive at best and their functional studies seem quite preliminary. The authors also include results from a cohort of hospitalized Covid-19 patients. These studies do not seem connected to the rest of the paper, and I'm not sure if they add any value to the manuscript.

Specific points:

1. The authors state in the abstract and discussion that CFHR5 is a potential diagnostic plasma biomarker for VTE and also discuss the limitations of D-dimer as the only VTE biomarker in routine clinical use. It would therefore be interesting to see how CFHR5 performs as a biomarker for VTE compared to D-dimer. Currently this comparison is only shown as a correlation analysis, but the authors should also perform a ROC curve analysis to show the sensitivity and specificity of CFHR5 for VTE and how it compares to D-dimer. Moreover, does a combined assessment of D-dimer and CFHR5 improve VTE prediction?

This is indeed an important point the reviewer has raised. We therefore performed an analysis adding absolute concentrations levels of CFHR5 into a model based on the clinical decision rule (CDR) used in most ER settings today. In the current CDR algorithm, an assessment of clinical probability is first performed using a Wells score, with different score sets used for DVT and PE. In patients with Wells score <2 for DVT and <4 for PE, a VTE diagnosis is considered to be of low probability i.e., 'VTE unlikely'; in these patients a D-dimer test is performed. If this is negative (below age adjusted cut off, i.e., $0.5 \text{ mg/L} + (\text{age}-50)/100$), it is considered safe to rule out VTE without the need for further diagnostic imaging (high negative predictive value), but D-dimer is positive, imaging is used to confirm or exclude a VTE diagnosis. In patients with high probability according to Wells score (≥ 2 for DVT and ≥ 4 for PE), current guidelines dictate that diagnostic imaging should be performed without prior D-dimer testing, as several factors incorporated in the Wells score are associated with increased D-dimer. International studies have shown that less than 10-20% of computed tomography pulmonary angiogram (CTPA) performed on suspicion of PE confirms such a diagnosis (Mittadodla et al., 2013) and thus, particularly in the group 'VTE likely' there is a high overutilisation of imaging within current CDR guidelines.

We performed an analysis to determine the potential application of CRHR5 as a diagnostic biomarker in: (i) the full VEBIOS ER group, and (ii) where patients were stratified according to 'VTE likely' or 'VTE unlikely'. Adding CFHR5 to a base model of D-dimer alone (dichotomized according to age adjusted cutoff) in the full VEBIOS ER group resulted in a non-significant improvement in AUC (0.88 versus 0.82, $p=0.110$). A model based on D-dimer + CFHR5 concentration + Wells score did not perform better than D-dimer alone (AUC 0.86 versus 0.82, $p=0.197$). In the sub analysis, in the 'VTE unlikely' group ($n=43$), adding CFHR5 to D-dimer alone resulted in a non-significant increased accuracy compared to the base model (AUC 0.84 versus 0.81, $p=0.61$). However, in the group representing the main diagnostic challenge in the acute setting, the 'VTE likely' group, the combination of CFHR5 and D-dimer performed significantly better than D-dimer alone (AUC 0.92 vs 0.83; $p=0.035$).

All data from this analysis has been added to a new tab in Table S1 (Tab 5 UAC).

In VEBIOS ER we have not defined cut-off values for CFHR5 concentration to calculate sensitivity and specificity, as additional studies are needed to establish reference range and cut-off value in patients with suspected acute VTE.

We have added the following text to the results section [lines 253-269]:

‘CFHR5 measurement can increase diagnostic accuracy in patients with likely VTE

To explore the potential usefulness of CFHR5 as a biomarker to be included in the diagnostic workup of suspected acute VTE, we assessed the discriminatory power of CFHR5 in VEBIOS ER using logistic regression in different models together, with D-dimer dichotomised using current Clinical Decision Rules (CDR) as ‘positive’ or ‘negative’ (below age adjusted cut-off (Douma et al., 2010)) and Wells score (VTE likely (≥ 2 for DVT and ≥ 4 for PE) or unlikely) (Table S1, Tab 5). In VEBIOS ER, D-dimer had negative predictive value (NPV) of 100% (0 false negatives) for VTE, while the specificity and positive predictive value (PPV) was only 62.8% and 74% respectively, with 16 false positive cases. Adding CFHR5 to the base model of D-dimer alone resulted in a non-significant improvement in AUC (0.88 versus 0.82, $p=0.110$), as did adding Wells score to the base model (AUC 0.85, $p=0.33$) (Table S1, Tab 5). D-dimer alone performed better than CFHR5 alone (AUC 0.73 versus 0.82, $p=0.128$). When stratifying patients based on Wells score, in the group where VTE was considered unlikely based on Wells score ($n=43$), adding CFHR5 to the base model resulted in a non-significant increased accuracy compared to the base model (AUC 0.84 versus 0.81, $p=0.61$). However, in the group where VTE was considered likely ($n=41$), the addition of CFHR5 to the base model resulted in a significantly increased accuracy compared to D-dimer alone (AUC 0.92 vs 0.83; $p=0.035$) (Table S1, Tab 5)‘

We have added the following text to the discussion section [lines 618-627]:

‘Current clinical decision rule (CDR) in diagnostic workup of suspected acute VTE is based on age adjusted D-dimer and Wells score. In VEBIOS ER we found that adding CFHR5 to D-dimer increased diagnostic accuracy of acute VTE in the VTE-likely group (Wells score ≥ 2 for DVT and PE). This group represents the major diagnostic challenge, as an elevated D-dimer is common in several of the conditions associated with increased risk for VTE, e.g., cancer and surgery, both of which are included in Wells score. Therefore, according to current CDR, patients with high clinical probability based on Wells score proceed to diagnostic imaging without prior D-dimer testing (Kline, 2020; Zarabi et al., 2021). Thus, adding CFHR5 concentration to D-dimer in the diagnostic work-up could potentially reduce number of negative imaging procedures, to the benefit of patients and health care system‘

We have added the following text to the methods section [lines 879-886]:

‘Discriminatory accuracy of plasma concentrations of CFHR5 and of D-dimer categorized as ‘positive’ or ‘negative’ using age adjusted D-dimer cutoff (Douma et al., 2010) in the different models was assessed using logistic regression analysis and presented as Area Under the Receiver Operating Characteristics curve (AUC). Statistical analyses were performed using R version 4.0.3. ROC curves for the different biomarker-based risk models based on plasma concentration CFHR5, dichotomized data on D-dimer (positive or negative) and Wells score (‘VTE likely’ (≥ 2 for DVT and ≥ 4 for PE) or ‘VTE unlikely’) were compared using the function roc.test (Delonge’s test) in the RStudio attachment. All tests were two-sided.‘

2. In figure 4, the authors report a genetic association study that identifies a novel CFHR5 pQTL. The identification of a genetic marker for CFHR5 plasma levels offers a possibility to assess a causal role of CFHR5 in the outcome studied. The lead SNP is highly significant for CHFR5 plasma levels, yet does not show any association with VTE. Although the genetics of the CFHR1-5 locus is quite complex, this result seems at odds with the author's conclusion that CFHR5 is a causal factor for VTE and should be discussed in more detail. Moreover, does the CFHR5 pQTL associate with other parameters such as D-dimer levels and other biomarkers?

The response below was also provided to reviewer 1 (point 3):

We interrogated recently released proteogenomics databases where GWAS on CFHR5 levels have been conducted, and to perform a meta-analysis of our own CFHR5 GWAS, based on ~3000 individuals, together with GWAS results obtained in the Omicscience (Pietzner et al., 2021a), EPIC (Koprulu, 2022) and Decode (Feringstad et al., 2021) genomic initiatives. This increased the sample size of our GWAS for CFHR5 levels to include ~52,000 individuals.

We identified 6 loci that were significantly associated with CFHR5 levels with genome-wide significance ($5E-08$) in the extended meta GWAS analysis (see Figure 1 for review). One of these loci was the *CFHR1-CFHR5* gene cluster. The lead SNP at this locus was the same as that we initially reported in the previous version of this manuscript, where the GWAS was conducted only in ~3000 samples and CFHR5 was measured using our dual binder assay. Strong linkage disequilibrium holds at this locus covering more than 10Mb and extending from CFHR1 to CFHR5 (see Figure 2 for review). Importantly, in the newly analysed cohorts included in the extended meta-analysis, plasma CFHR5 was measured using two independent techniques (namely, Somalogic and Olink). This consistency between analysis platforms, together with the homogeneity of the effects detected in the extended meta-analysis [Table S1, Tab 12], provide additional support for our original results and conclusions. Of note, to address the fact that different techniques were used to measure CFHR5, the meta-analysis of GWAS results was conducted using the Z-score fixed-effect model as recommended in Chauhan et al. (Chauhan et al., 2015b).

Figure 1 for review [new Figure 4A in revised manuscript]. Manhattan plot of the meta-analysis on INVENT-MVP consortium resources identifying 6 loci associated with CHFR5 plasma levels and VTE risk.

Figure 2 for review [new Figure 4B in revised manuscript]. Regional association plot around the lead SNP at CFHR1-CFHR5 locus.

We then went on to assess the impact on VTE risk of the lead polymorphisms that were detected in the extended GWAS on CFHR5 and to conduct Colocalization and Mendelian Randomization (MR) analyses using the latest GWAS findings on VTE obtained by the INVENT-MVP consortium (Thibord et al., 2022a). Five of the six lead SNPs at the identified loci with VTE showed marginal association ($p=4E-04$ to $p=0.026$) with VTE. However, for only two of them (*JMJD1C* rs7916868 and *DNAH10* rs7133378), the pattern of association with CFHR5 and with VTE risk was consistent. Of note, these two polymorphisms were those showing the stronger (but still moderate) probability of colocalization (PP4 ~40%) [Table S1, Tab 11]. As a consequence, the MR analysis does not provide strong support for causality between CFHR5 levels and VTE risk [Table S1, Tab 13]. The MR analysis was performed using genetic factors for VTE as a whole without distinguishing pulmonary embolism and deep vein thrombosis, provoked and unprovoked events. Such heterogeneity may have impacted our MR analysis.

Data has been added to Table S1, Tab 11, 12 and 13.

According to the GWAS catalog, these pQTL variants did not show evidence for association with D-dimer but with liver enzymes (*HNF1A*, *HNF4A* and *TRIB1*), with lipids (*DNAH10*, *HNF1A*, *HNF4A*, *JMJD1C* and *TRIB1*) and some other biological traits (platelets, red/white blood cells) involved the physiopathology underlying VTE. This point has been mentioned on page 22

We have added the following text to the results section [lines 374-389]:

A second round of meta-analysis, integrating GWAS summary statistics from 3 additional proteogenomic resources where CFHR5 was measured with different assays [see methods], totalling ~50,200 individuals, confirmed the association of this locus with CFHR5 levels. Interestingly, the rs10737681 was also identified as the lead SNP at this locus in the extended meta-analysis ($\beta = 0.28 \pm 0.01$, $p=2.94E-396$) (Table S1, Tab 12). Strong linkage disequilibrium holds at the locus covering more than 10Mb and extending from CFHR1 to CFHR5 (Figure 4B). The extended meta-analysis identified 5 additional independent loci associated with CFHR5

levels: HNF1A (rs2393776, $p=1.48E-21$) on 12q24.31, JMJD1C (rs7916868, $p=4.61E-12$) on 10q21.3, TRIB1 (rs28601761, $p=4.39E-09$) on 8q24.13, DNAH10 (rs7133378, $p=2.43E-08$) also on 12q24.31 and HNF4A (rs1800961, $p=4.97E-08$) on 20q13.12 (Figure 4A and Table S1 Tab 12). All of the lead SNPs at these loci, except HNF1A rs2393776 ($p=0.17$), demonstrated marginal ($p<0.05$) association with VTE risk (Table S1, Tab 11). However, only two, JMJD1C_rs7916868 and DNAH10_rs7133378, showed patterns of association with VTE that are compatible with the association of increased CFHR5 levels with VTE risk. This explains why MR analyses are not supportive for a causal association between increased CFHR5 levels and VTE (Table S1, Tab 13)

We have added the following text to the discussion section [lines 483-503]:

The CFHR5 locus maps to chromosome 1q31.3 at one end, a gene cluster that spans approximately 350 kb including (in order from CFHR5) the CFHR2, CFHR4, CFHR1, CFHR3, and CFH loci. The rs10737681 we identified with genome wide association with plasma CFHR5 level maps between the CFHR4 and CFHR1 genes. Of note, the CFHR2 locus has just been identified as a novel susceptibility locus for VTE in the recently published international effort on VTE genetics (Thibord et al., 2022b). The lead SNP at this locus is rs143410348, which is in moderate LD with the rs10737681 ($r^2\sim 0.40$ in European population (Machiela and Chanock, 2015)), which here we found associated with CFHR5 plasma levels. In a combined meta-analysis of 37,770 individuals from 3 cohorts (EPIC/FARIVE/Omicscience) where it was imputed, rs143410348 was less strongly associated with CFHR5 levels than the lead rs10737681 ($\beta = -0.15$, $p = 1.9E-32$ vs $\beta = -0.27$, $p = 2.1E-95$). Altogether, the observations from the GWAS analyses emphasize the need for a deeper investigation of the genetic architecture of the CFHR1/CFHR4/CFHR2/CFHR5 locus with respect to CFHR5 levels and VTE risk. Five additional candidate loci were identified as participating to the genetic regulation of CFHR5 plasma levels: DNAH10, HNF1A, HNF4A, JMJD1C and TRIB1. All 5 loci have been reported to associate with various lipids traits (see GWAS catalogue (Sollis et al., 2022)) and 3 (HNF1A, HNF4A and TRIB1) have been reported to also associate with liver enzymes. DNAH10 is also known to be a locus involved in white & red blood cell biology (Chen et al., 2020) while the JMJD1C is a locus linked to platelet biology (Eicher et al., 2016; Johnson et al., 2010). Most of these traits are well known risk factors for VTE, and this may suggest that the association of CFHR5 levels with VTE risk implies many additional biological players with pleiotropic effects. This may explain why the MR analyses did not provide causal evidence for a link between CFHR5 levels and VTE risk

We have added the following text to the methods section [lines 954-965]:

A second round of meta-analysis was performed by integrating GWAS summary statistics from 3 additional proteogenomic resources where CFHR5 have been measured. These include 35,559 Icelander participants of the Decode project (Feringstad et al., 2021) and 10,708 participants from the Fenland study with CFHR5 plasma levels measured using the Somalogic platform (Pietzner et al., 2021a) and an additional independent sample of 1,178 EPIC participants with CFHR5 measured using the Olink platform (Koprulu, 2022). Of note, in a sample of 485 Fenland participants with both measurements CFHR5, the correlation between Somalogic and Olink-derived CFHR5 levels was 0.35 [Supplementary Data Set 2 in (Pietzner et al., 2021b)]. To meta-analyse GWAS results for CFHR5 plasma levels measured with three different techniques, (dual binder assay, Somalogic and Olink), we used the Z-score fixed-effect model implemented in the METAL software (Willer et al., 2010). In order to conduct the other genetic analyses, normalized regression coefficient and their standard error were derived from Z-scores using the method described in Chauhan et al (Chauhan et al., 2015a)

3. Figure 5 demonstrates a moderate potentiating effect of exogenously added CHFR5 on platelet activation in platelet-rich plasma but not washed platelets. These results are interesting but feels somewhat preliminary. The mechanism is unclear, and the calcium chelator citrate is used as anticoagulant, which might interfere with complement activation under the conditions studied.

a. The results are based on flow cytometric measurements of platelet activation markers. These results should be complemented by a functional assessment of platelet activation, e.g., aggregometry.

We agree that such a functional readout would be a useful addition to the manuscript. Therefore, as suggested, we performed additional experiments to determine if platelet aggregation in response to stimulation was potentiated by recombinant CFHR5. We used ADP as the agonist to initiate platelet aggregation, as the potentiation of activation marker expression induced by CFHR5 was most marked when this was used as the platelet agonist (see original Figure 5). However, we did not observe any effect of recombinant CFHR5 on platelet aggregation response to ADP, compared to the control (PBS alone) (see Figure 3 for review).

Figure 3 for review [new Figure 5 D.i-iii in revised manuscript]. Recombinant CFHR5 does not potentiate platelet aggregation in response to ADP stimulation. (A) Representative plot showing platelet aggregation in response to ADP stimulation in the presence or absence of rCFHR5. (B) Data from four independent biological replicates showing: (i) maximum aggregation or (ii) slope.

Activated platelets express and secrete proinflammatory and procoagulant factors that can drive VTE independent of platelet aggregation (Heemskerk et al., 2013). While increased plasma levels of platelet activation markers have been linked to acute VTE in several studies, e.g., (Montoro-Garcia et al., 2016), reports are inconsistent regarding the association between increased platelet aggregation (*ex vivo*, in response to platelet activation agonists) and VTE (Llobet et al., 2021; Panova-Noeva et al., 2020; Puurunen et al., 2017; Weber et al., 2002). Previously, in the RETROVE study (the cohort in which we found CFHR5 associated with both VTE and thrombin generation) an analysis of platelet aggregation in response to ADP and epinephrine, using platelet rich plasma from 400 patients and 400 controls, found no association between VTE and hyperaggregability (Llobet et al., 2021). Thus, our results could indicate a role for CFHR5 in VTE-linked platelet activation, which is independent of platelet aggregation.

We have added the following text to the results section [lines 430-431]:

Although CFHR5 potentiated the expression of platelet activation markers in response to ADP, it did not modify ADP-induced platelet aggregation (Figure 5D.i-iii)

We have added the following text to the discussion section [lines 564-571]:

‘The co-stimulatory effect of CFHR5 on platelet activation did not translate into an effect on ADP-induced platelet aggregation. Activated platelets express and secrete proinflammatory and procoagulant factors that can directly drive VTE, independent of platelet aggregation (Heemskerk et al., 2013). In the RETROVE study (where we found CFHR5 is associated with VTE and increased thrombin generation) previous studies found no association between VTE and platelet aggregation in response to ADP or epinephrine (Llobet et al., 2021). Thus, one could speculate that CFHR5 has a role in VTE-linked platelet activation that is independent of platelet aggregation’

We have added the following text to the methods section [lines **1024-1031**]:

‘PRP was prepared from citrated blood of healthy volunteers by centrifugation (120g, 20min, RT). Platelet aggregation was determined by light transmission aggregometry using a Platelet Aggregation Profiler PAP-8 (möLab), defining 0% aggregation as naïve PRP as and 100% aggregation as platelet-poor plasma (PPP), which was obtained by centrifuging PRP in the presence of 0.1µg/ml PGI2 (1000g, 90sec, RT). To determine the effect of CFHR5 on platelet aggregation, PRP was monitored in the aggregometer for 1 minute before addition of 6µg/ml CFHR5 or PBS). After 10 minutes, PRP was stimulated with 7µM ADP and aggregation monitored for further 10 minutes’

b. The authors should provide additional experimental results to be able to say something about the mechanism by which CFHR5 potentiates platelet activation. A proposed function of CHFR5 is that it augments complement activation by competing with factor H for binding to host surfaces. Hence, is the effect of CFHR5 on platelet activation complement dependent? Does CFHR5 still augment platelet activation in the presence of complement inhibitors, such as a C3 inhibitor?

We performed additional experiments to address this point. As in the response to point 3a above, for these additional experiments we focused on the platelet response to ADP, selecting activated GP IIb/IIIa (PAC1⁺) expression as the end point readout, as in the original dataset the expression of this activation marker was the most consistently augmented by CFHR5 (see original Figure 5). To determine if the mechanism was potentially linked to complement activation, prior to treatment we pre-incubated platelets with either compstatin, a C3-targeted complement inhibitor (Mastellos et al., 2015) or an anti-C3a function blocking antibody. CFHR5 pretreatment increased the expression of platelet activated GP IIb/IIIa (PAC1⁺) in the controls (DMSO or PBS, for compstatin and anti-C3a antibody, respectively) and that expressed in response to ADP stimulation (Figure 4 for review A.i and B.i). When complement was inhibited by either compstatin or anti-C3a antibody, the increased activated GP IIb/IIIa (PAC1⁺) detection potentiation by CFHR5 was no longer observed (Figure 4 for review A.ii and B.ii). Thus, these data indicate a role for complement activation in the CFHR5-induced potentiation in activated GP IIb/IIIa (PAC1⁺).

or **(B)** (i) PBS or (ii) an anti-C3a antibody, followed by treatment with PBS or recombinant CFHR5 (indicated by blue or red data points, respectively). US: unstimulated (PBS control).

We have added the following text to the results section [lines **435-445**]:

‘A proposed function of CHFR5 is that it augments complement activation by antagonising complement factor H (CFH), the main negative regulator of alternative pathway (AP) activation in plasma. CFH inhibits C3 convertase, preventing formation of C3a (Cserhalmi et al., 2019), which has been suggested to have role in platelet activation and subsequent thrombosis formation (Sauter et al., 2018). To determine if CFHR5-induced augmentation of platelet activation was dependent on C3 cleavage and generation of C3a, we pre-treated platelets with an inhibitor of C3 cleavage and activation of C3a (compstatin), or an anti-C3a antibody, prior to CHFR5 pre-incubation and subsequent ADP stimulation. The potentiation effect of CHFR5 on baseline and ADP-induced activated GP IIb/IIIa expression (Figure 5E.i and 5F.i) was abolished following compstatin (Figure 5E.ii) or anti-C3a antibody (Figure 5F.ii) pre-treatment; data consistent with a complement dependent effect of CFHR5 on platelet activation’

We have added the following text to the methods section [lines **1020-1022**]:

‘In some experiments PRP was incubated for 20 minutes with 0.25% DMSO, 100µM compstatin, PBS, 10µg/ml anti-C3a/C3a (desArg)(clone K13/16), prior to assay as described above’

We have added the following text to the discussion [lines **560-561**]:

‘In our study, we observe a similar co-stimulatory effect of CFHR5 on ADP- (and convulxin- or TRAP6-) induced platelet activation. This effect was observed on platelets in plasma, but not on those that were pre-washed, consistent with the effect of CFHR5 on platelet activation being due to its interaction with other complement factors (i.e., C3) in plasma. Furthermore, in the presence of compstatin, an inhibitor of C3 cleavage and formation of C3a, or anti-C3a antibody, the co-stimulatory effect of CFHR5 was not observed’

c. With regards to the potential mechanism, the authors propose that increased generation of the anaphylatoxin C3a would lead to platelet activation. It's important to note that the expression of anaphylatoxin receptors on platelets is controversial. Other studies (see e.g. Fukuoka et al, J Immunol;1988,140(10):3496-501) have found no evidence of C3a receptor expression in human platelets, which should be acknowledged in the manuscript.

As the reviewer mentions, the expression of C3a receptor on platelets was controversial for several decades; with some studies providing evidence for the expression of C3a receptors on the platelet surface e.g., (Polley and Nachman, 1983), whilst other refuted such findings e.g., (Fukuoka and Hugli, 1988). However, more recent studies (Sauter et al., 2018) have demonstrated the presence of C3aR on human platelets using several different techniques (flow cytometry, immunostaining, platelet activation assays).

As suggested, we have acknowledged these points in the discussion [lines **547-551**]:

‘C3a, acting through platelet receptor C3aR, is suggested to have role in the activation of the glycoprotein IIb/IIIa fibrinogen receptor via intraplatelet signaling, and subsequent thrombosis formation (Sauter et al., 2018). The presence of a C3a receptor on human platelets has been controversial, with several contradictory studies (Fukuoka and Hugli, 1988; Polley and Nachman, 1983). However, recent studies have confirmed the presence of C3aR on human platelets using several independent techniques (Sauter et al., 2018)’

d. The anticoagulant citrate chelates calcium and magnesium ions that are required for complement activation. Can the authors use an alternative anticoagulant for the platelet studies, such as hirudin, that does not interfere with complement?

As the reviewer suggested, we performed additional experiments to compare results obtained using platelet rich plasma from blood anticoagulated with citrate or hirudin. In citrated blood, addition of recombinant CFHR5 potentiated platelet activated GP IIb/IIIa (PAC1⁺) detection (Figure 5 for review only, A.i and B.i). Baseline detection of activated GP IIb/IIIa (PAC1⁺) on unstimulated and ADP stimulated platelets was significantly lower when hirudin was used as the anticoagulant, compared to citrate (mean % expression ± std dev. [unstimulated: citrate 9.6±4.8 vs. hirudin 1.6±1.2] and [ADP stimulated: citrate 30±15.0 vs. hirudin 7.5±3.2]). We did not observe any effect of recombinant CFHR5 on activated GP IIb/IIIa (PAC1⁺) detection in either condition on platelets from hirudin anticoagulated blood (Figure 5 for review only, A.i and ii).

The P2Y12 receptor is the predominant receptor involved in the ADP-stimulated activation of the glycoprotein IIb/IIIa receptor, inducing enhanced platelet degranulation, thromboxane production and platelet aggregation. In line with our results, previous studies have described an inhibition of activation on platelets isolated from hirudin, e.g., (Janse van Rensburg and van der Merwe, 2017), compared to citrate anticoagulated blood. One study, designed to compare different platelet function tests for measuring the effect of P2Y12 receptor antagonists (e.g. clopidogrel) on platelet reactivity, revealed a significantly higher magnitude of platelet aggregation in response to ADP induced stimulation of P2Y12 when measured in citrate, compared to hirudin (Pittens et al., 2009). Thus, one could speculate that a lack of general responsiveness of platelets isolated from hirudin could contribute to the lack of potentiation of response by rCFHR5. Due to manuscript space restrictions and the rather speculative nature of any interpretation of these results, we have not included this new data in the revised version of the manuscript.

Figure 5 for review only. Baseline and CFHR5 potentiated ADP-induced expression of activated GP IIb/IIIa (PAC1⁺) is inhibited on platelets isolated from hirudin vs. citrate anticoagulated blood. (A) Baseline or ADP-induced platelet activated GP IIb/IIIa (PAC1⁺) expression (%) was measured on platelets in plasma isolated from (i) citrate- or (ii) hirudin- anticoagulated blood, following preincubation with either PBS control or recombinant CFHR5 (indicated by blue or red data points, respectively). The same data, normalized to each respective ADP-stimulated control, is presented in (B). US: unstimulated control (PBS).

e. CHFR5 was obtained from a commercial source. How was CFHR5 characterized, and is it functional?

In order to confirm functionality of the commercial recombinant CFHR5 protein (rCFHR5) used in the manuscript, we evaluated its capacity to bind its known partners C-reactive protein (CRP) (McRae et al., 2005) and properdin (Chen et al., 2016). Using ELISA-based assays we found that the rCFHR5

binds in a dose-dependent manner to both CRP and properdin (see Figure 6 for review). Moreover, binding properties are impaired after heat-denaturation, generating a signal comparable to the bovine serum albumin (BSA) control, demonstrating structure-dependent binding function.

Figure 6 for review [new Figure S4 in revised manuscript]. CFHR5 binding to C-reactive protein and properdin. Either (A) C-reactive protein (CRP) or (B) properdin, were immobilised on microtiter plates, before serial dilutions of rCFHR5 were added and binding detected using a mouse anti-human CFHR5 (MAB3845) antibody, followed by HRP-labeled rabbit anti-mouse Ig. Binding was visualized using *o*-Phenylenediamine dihydrochloride (OPD). Data is displayed as mean signal for serial dilutions or (ii) individual replicates and controls (mock; heat-denaturation rCFHR5, BSA; bovine serum albumin).

Further, we performed immunostaining of rCFHR5 by Western blot (see Figure 7 for review). In relation to the structure-function of the protein, in non-reducing conditions two bands were detected, demonstrating that the capacity of dimerization was intact (Goicoechea de Jorge et al., 2013).

Figure 7 for review [image extracted from Figure S2B in revised manuscript]. Immunodetection of recombinant CFHR5 (rCFHR5, 100 ng) by Western blot. rCFHR5 in non-reducing (without dithiothreitol [DTT]) or reducing conditions (with DTT) was loaded on SDS PAGE. After electrophoresis and transfer on PVDF membrane, protein was assayed using detection antibody targeting CFHR5 (MAB3845). The arrow shows the band corresponding to CFHR5 protein.

We have added the following text to the results section [lines **417-419**]:

‘Functionality of the rCFHR5 was validated by its capacity to form a homodimer complex and its ability to bind known interaction partners C-reactive protein (CRP) (McRae et al., 2005) and properdin (Chen et al., 2016) (see methods and Figure S2B and S4)’

We have added the following text to the methods section [lines **988-1003**]:

‘Validation of structural functionality of recombinant CFHR5

In order to confirm the functionality of the commercial rCFHR5 reagent used, we evaluated its binding capacity to its known partners C-reactive protein (CRP) (McRae et al., 2005) and properdin (Chen et al., 2016), using in-house designed ELISA. Flat-bottom microtiter plates (Nunc) were coated with recombinant CRP (1 µg, AG723-M Sigma), properdin (5 µg, 341283 - Sigma) or BSA (5 µg, negative control). After blocking with TBS-T containing 3% BSA, serial dilutions of rCFHR5 (3845-F5, RnD systems) or heat denatured rCFHR5 (mock) were added in triplicate and incubated for 1 hour at 20°C. rCFHR5 binding to CRP or properdin was detected with monoclonal mouse anti-human CFHR-5 (MAB3845, RnD) followed by HRP-labeled rabbit anti-mouse Ig (Dako). Binding was visualized using o-Phenylenediamine dihydrochloride (OPD) (P9187 Sigma) and absorbance measured at 492 nm following the addition of 3 M HCl as stop solution in microplate photometer (Multiskan FC, ThermoScientific). rCFHR5 bound CRP and properdin in a dose-dependent manner (Figure S4A.i and S4B.i, respectively). Binding properties were impaired following rCFHR5 heat-denaturation, indicating structure-dependent function (Figure S4A.ii-B.ii). Immunostaining of rCFHR5 by Western blot in non-reducing conditions revealed two bands (Figure S2B, lane 1 and 7), showing intact capacity for dimerization (Goicoechea de Jorge et al., 2013)’

Moreover, platelets are delicate to handle and quite prone to unspecific activation. A mock recombinant protein from the same source should be used as control to exclude any unspecific effects in the CFHR5 stimulation experiments.

It was difficult to select a recombinant protein that we could be confident would have no effect on platelets, so we attempted to use heat denatured rCFHR5 as a control. Unfortunately, the results were

inconsistent and inconclusive; indeed it is possible that heat treatment alone is not sufficient to consistently abolish rCFHR5 function (see in Figure S4 for review A.ii and B.ii - some binding capacity remains even after heat treatment [‘mock’]). However, in addressing other parts of the reviewers comments we performed additional platelet functional experiments (as described above), and when this new data was taken together with the original data, a number of observations strongly indicate that platelet response to rCFHR5 is not due to a non-specific activation:

- rCFHR5 augmented activated GP IIb/IIIa was abolished in the presence of two different types of complement inhibitors, which have independent mechanisms of action.
- rCFHR5 did not induce platelet aggregation.
- rCFHR5 augmented activation of platelet GP IIb/IIIa in response to ADP stimulation was greater and more consistent than augmentation of other activation markers, e.g.-selectin or CD63, or that observed in response to other platelet stimulants, e.g., convulxin or TRAP6.
- There was no rCFHR5 augmented platelet activation when washed platelets were assayed.

f. If the authors believe that CHFR5 exerts prothrombotic effect via increased platelet activation this should be discussed in relation to the association of CHFR with venous thrombosis. Venous thrombosis has historically been more associated with plasma coagulation than platelets. In contrast to arterial thrombosis, VTE is treated with anticoagulants and not anti-platelet drugs. This should be discussed in more detail.

We have added the following text to the discussion [lines **529-539**]

‘The mechanisms underlying venous and arterial thrombosis development differ; venous thrombi contain an abundance of red blood cells trapped in a fibrin clot together with platelets, a structure quite distinct from the vast platelet aggregates found in arterial thrombi (Koupenova et al., 2017). Thus, arterial thrombosis is treated with therapies that target platelet activation and/or aggregation while VTE is traditionally treated with drugs targeting the coagulation system. Historically, platelet function has attracted attention primarily in arterial thrombosis, however more recently the role of platelets in VTE has been recognised (Montoro-Garcia et al., 2016). Elevated levels of markers of platelet activation, such as P-selectin, are associated with acute VTE (Jacobs et al., 2016); a protein we also identified as one of four candidates associated with VTE in the discovery screen of VEBIOS ER. Furthermore, anti-platelet therapy with acetylic salicylic acid had a protective effect against VTE (Simes et al., 2014), and reduced the size of venous thrombus linked to inhibition of platelet activation in mice (Tarantino et al., 2016)’

4. The last section of the manuscript and figure 6 report on CHFR5 plasma levels in hospitalized Covid-19 patients. Even though the rest of the manuscript focuses on VTE, nothing is mentioned about thrombotic events or coagulation status in these patients. Currently it is unclear to the reviewer what these results add to the manuscript. At least, the authors should provide information on thrombotic events and coagulation biomarkers in these patients and their relationship to CFHR5 levels.

We agree with the general consensus between the reviewers that this section does not significantly add to the manuscript in its current form, and we have therefore removed it.

Reviewer #3 (Remarks to the Author):

Review of Elevated plasma Complement Factor H Regulating Protein 5 is associated with venous thromboembolism and COVID-19 severity

Reviewer: Brent Richards

This is an impressive paper. The authors have done a formidable job provided orthogonal lines of evidence indicating that CFHR5 is a risk factor for VTE. The use of replication cohorts is greatly appreciated. The manuscript is clear and well-written. The figures are generally excellent. All in all, this is a solid contribution to the field.

I will attempt to limit my comments within the realm of genetic epidemiology, since I lack expertise in VTE, protein assays or platelet activation. I'd like to also state that this paper was reviewed by both me and one of my PhD students independently. Many of our comments were overlapping (which either suggests groupthink or potential problems to address).

Major Comments:**Genetics**

1. CHFR5 is measured in both Olink Explore and Somascan assays. There are cis-pQTLs identified for this protein. In the deCODE pQTL paper (Feringstad et al, NG, 2021), there are cis-pQTLs for CFHR1, 2, 3, 4 and 5. There are 14 cis pQTLs for CFHR5 in this paper. This represents an opportunity to better understand if the ci-pQTLs for CFHR5 are unique to CFHR5, or whether they are pleiotropic.

We thank Brent Richards (and his PhD student) for this comment, which is similar to those also made by other reviewers. Our response below was also provided to Reviewer #1 (point 3).

We interrogated recently released proteogenomics databases where GWAS on CFHR5 levels have been conducted, and to perform a meta-analysis of our own CFHR5 GWAS, based on ~3000 individuals, together with GWAS results obtained in the Omicscience (Pietzner et al., 2021a), EPIC (Koprulu, 2022) and Decode (Feringstad et al., 2021) genomic initiatives. This increased the sample size of our GWAS for CFHR5 levels to include ~52,000 individuals.

We identified 6 loci that were significantly associated with CFHR5 levels with genome-wide significance ($5E-08$) in the extended meta GWAS analysis (see Figure 1 for review). One of these loci was the *CFHR1-CFHR5* gene cluster. The lead SNP at this locus was the same as that we initially reported in the previous version of this manuscript, where the GWAS was conducted only in ~3000 samples and CFHR5 was measured using our dual binder assay. Strong linkage disequilibrium holds at this locus covering more than 10Mb and extending from CFHR1 to CFHR5 (see Figure 2 for review). Importantly, in the newly analysed cohorts included in the extended meta-analysis, plasma CFHR5 was measured using two independent techniques (namely, Somalogic and Olink). This consistency between analysis platforms, together with the homogeneity of the effects detected in the extended meta-analysis [Table S1, Tab 12], provide additional support for our original results and conclusions. Of note, to address the fact that different techniques were used to measure CFHR5, the meta-analysis of GWAS results was conducted using the Z-score fixed-effect model as recommended in Chauhan et al. (Chauhan et al., 2015b).

Figure 1 for review [new Figure 4A in revised manuscript]. Manhattan plot of the meta-analysis on INVENT-MVP consortium resources identifying 6 loci associated with CHFR5 plasma levels and VTE risk.

Figure 2 for review [new Figure 4B in revised manuscript]. Regional association plot around the lead SNP at CFHR1-CFHR5 locus.

We then went on to assess the impact on VTE risk of the lead polymorphisms that were detected in the extended GWAS on CFHR5 and to conduct Colocalization and Mendelian Randomization (MR) analyses using the latest GWAS findings on VTE obtained by the INVENT-MVP consortium (Thibord et al., 2022a). Five of the six lead SNPs at the identified loci with VTE risk, showed marginal association ($p=4E-04$ to $p=0.026$) with VTE. However, for only two of them (*JMD1C* rs7916868 and *DNAH10* rs7133378), the pattern of association with CFHR5 and with VTE risk was consistent. Of note, these two polymorphisms were those showing the stronger (but still moderate) probability of colocalization ($PP4 \sim 40\%$) [Table S1, Tab 11]. As a consequence, the MR analysis does not provide strong support for causality between CFHR5 levels and VTE risk [Table S1, Tab 13]. The MR analysis was performed using genetic factors for VTE as a whole without distinguishing pulmonary embolism

and deep vein thrombosis, provoked and unprovoked events. Such heterogeneity may have impacted our MR analysis.

We have added the following text to the results section [lines **374-380**]:

‘A second round of meta-analysis, integrating GWAS summary statistics from 3 additional proteogenomic resources where CFHR5 was measured with different assays [see methods], totalling ~50,200 individuals, confirmed the association of this locus with CFHR5 levels. Interestingly, the rs10737681 was also identified as the lead SNP at this locus in the extended meta-analysis ($\beta = 0.28 \pm 0.01$, $p = 2.94E-396$) (Table S1, Tab 12). Strong linkage disequilibrium holds at the locus covering more than 10Mb and extending from CFHR1 to CFHR5 (Figure 4B)’

We have added the following text to the methods section [lines **954-965**]:

‘A second round of meta-analysis was performed by integrating GWAS summary statistics from 3 additional proteogenomic resources where CFHR5 have been measured. These include 35,559 Icelander participants of the Decode project (Feringstad et al., 2021) and 10,708 participants from the Fenland study with CFHR5 plasma levels measured using the Somalogic platform (Pietzner et al., 2021a) and an additional independent sample of 1,178 EPIC participants with CFHR5 measured using the Olink platform (Koprulu, 2022). Of note, in a sample of 485 Fenland participants with both measurements CFHR5, the correlation between Somalogic and Olink-derived CFHR5 levels was 0.35 [Supplementary Data Set 2 in (Pietzner et al., 2021b)]. To meta-analyse GWAS results for CFHR5 plasma levels measured with three different techniques, (dual binder assay, Somalogic and Olink), we used the Z-score fixed-effect model implemented in the METAL software (Willer et al., 2010). In order to conduct the other genetic analyses, normalized regression coefficient and their standard error were derived from Z-scores using the method described in Chauhan et al (Chauhan et al., 2015a)’

While of great interest, disentangling the exact genetic architecture at these loci that influence CFHR5 plasma levels is out of the scope of this paper as it would need to re-analyze and fine-mapping each GWAS datasets using complementary tools (such as conditional and haplotype analyses).

2. The authors should undertake Mendelian randomization studies of CFHR5 levels, given that they have a cis-pQTL. They should also assay if their associated lead SNP is associate with other CFHR protein levels. This can be done in the deCODE paper data, I listed above. The benefit of the MR study is that it would allow for a tremendous reduction in potential confounding and they could assess the CFHR5 effect on VTE using the largest GWAS meta-analysis of VTE, providing a much larger sample size than the FARIVE and RETROVE studies.

Our response below was also provided to Reviewer #1 (point 3):

We assessed the impact on VTE risk of the lead polymorphisms that were detected in the extended GWAS on CFHR5 and conducted Colocalization and Mendelian Randomization (MR) analyses using the latest GWAS findings on VTE obtained by the INVENT-MVP consortium (Thibord et al., 2022a). Five of the six lead SNPs at the identified loci with VTE risk. showed marginal association ($p = 4E-04$ to $p = 0.026$) with VTE. However, for only two of them (*JMJD1C* rs7916868 and *DNAH10* rs7133378), the pattern of association with CFHR5 and with VTE risk was consistent. Of note, these two polymorphisms were those showing the stronger (but still moderate) probability of colocalization (PP4 ~40%) [Table S1, Tab 11]. As a consequence, the MR analysis does not provide strong support for causality between CFHR5 levels and VTE risk [Table S1, Tab 13]. The MR analysis was performed using genetic factors for VTE as a whole without distinguishing pulmonary embolism and deep vein thrombosis, provoked and unprovoked events. Such heterogeneity may have impacted our MR analysis.

We also used the results of the Decode proteogenomics scan (Ferkingstad et al., 2021) to assess whether the 6 lead SNPs of our metaGWAS for CFHR5 associate with plasma levels of other CFHR proteins. The lead SNP on chromosome 1 is associated with all CFHR proteins levels (Table 1 for review). The magnitude of its genetic effects appeared slightly stronger for CFHR2 and CFHR4 compared to CFHR5. Lead SNPs at JMJD1C, HNF1A and HNF4A additional exhibited suggestive statistical evidence ($p \sim 10^{-3}$) with CFHR4 only.

Associations of the CFHR5 levels associated lead SNPs with CFHR protein levels in the DECODE study													
CHR:POS:NEA:EA	RS	Localisation	GENE	CFHR1		CFHR2		CFHR3		CFHR4		CFHR5	
				BETA(SE)	P	BETA(SE)	P	BETA(SE)	P	BETA(SE)	P	BETA(SE)	P
1:196851676:T:G	rs10737681	Intergenic	CFHR1;CFHR4	-0.17(0.01)	2.31x10 ⁻⁹⁰	0.57(0.01)	<10 ⁻⁴⁰⁰	0.05(0.01)	2.64x10 ⁻⁸	0.50(0.01)	<10 ⁻⁴⁰⁰	0.30(0.01)	4.64x10 ⁻²⁸⁵
8:125487789:C:G	rs28601761	Intergenic	TRIB1;LINC00861	0.002(0.01)	0.811	-0.01(0.01)	0.152	-0.002(0.01)	0.816	-0.006(0.01)	0.319	-0.04(0.01)	7.48x10 ⁻⁸
10:63229171:A:T	rs7916868	Intronic	JMJD1C	-0.02(0.01)	0.032	0.001(0.01)	0.927	0.01(0.01)	0.152	-0.02(0.01)	3.51x10 ⁻³	0.05(0.01)	6.39x10 ⁻¹¹
12:120986603:A:G	rs2393776	Intronic	HNF1A*	-0.002(0.01)	0.838	0.01(0.01)	0.227	-0.01(0.01)	0.111	0.04(0.01)	1.14x10 ⁻⁸	0.06(0.01)	7.77x10 ⁻¹⁵
12:123924955:A:G	rs7133378	Intronic	DNAH10	-0.005(0.01)	0.505	-0.02(0.01)	0.017	-0.008(0.01)	0.329	-0.004(0.01)	0.500	-0.04(0.01)	6.67x10 ⁻⁶
20:44413724:T:C	rs1800961	Exonic	HNF4A	-0.02(0.01)	0.293	0.03(0.01)	0.062	-0.006(0.02)	0.758	0.07(0.01)	2.91x10 ⁻⁷	0.09(0.02)	4.00x10 ⁻⁶

Table 1 for review. Association of the 6 lead CFHR5 levels –associated SNPs with plasma levels of CFHR1-CFHR2-CFHR3-CFHR4-CFHR5 protein in the Decode study (N = 35,559)

As these results do not significantly add to our findings, in light of space restrictions, we have not added them into the revised version of the manuscript. Measuring all CFHR proteins in the same case-control samples would be extremely useful, and we plan to investigate this issue in further work.

We have added the following text to the results section [lines **380-389**]:

The extended meta-analysis identified 5 additional independent loci associated with CFHR5 levels: HNF1A (rs2393776, $p=1.48E-21$) on 12q24.31, JMJD1C (rs7916868, $p=4.61E-12$) on 10q21.3, TRIB1 (rs28601761, $p=4.39E-09$) on 8q24.13, DNAH10 (rs7133378, $p=2.43E-08$) also on 12q24.31 and HNF4A (rs1800961, $p=4.97E-08$) on 20q13.12 (Figure 4A and Table S1 Tab 12). All of the lead SNPs at these loci, except HNF1A rs2393776 ($p=0.17$), demonstrated marginal ($p<0.05$) association with VTE risk (Table S1, Tab 11). However, only two, JMJD1C_rs7916868 and DNAH10_rs7133378, showed patterns of association with VTE that are compatible with the association of increased CFHR5 levels with VTE risk. This explains why MR analyses are not supportive for a causal association between increased CFHR5 levels and VTE (Table S1, Tab 13)

We have added the following text to the discussion section [lines **483-503**]:

The CFHR5 locus maps to chromosome 1q31.3 at one end, a gene cluster that spans approximately 350 kb including (in order from CFHR5) the CFHR2, CFHR4, CFHR1, CFHR3, and CFH loci. The rs10737681 we identified with genome wide association with plasma CFHR5 level maps between the CFHR4 and CFHR1 genes. Of note, the CFHR2 locus has just been identified as a novel susceptibility locus for VTE in the recently published international effort on VTE genetics (Thibord et al., 2022b). The lead SNP at this locus is rs143410348, which is in moderate LD with the rs10737681 ($r^2 \sim 0.40$ in European population (Machiela and Chanock, 2015)), which here we found associated with CFHR5 plasma levels. In a combined meta-analysis of 37,770 individuals from 3 cohorts (EPIC/FARIVE/Omicscience) where it was imputed, rs143410348 was less strongly associated with CFHR5 levels than the lead rs10737681 ($\beta = -0.15$, $p = 1.9E-32$ vs $\beta = -0.27$, $p = 2.1E-95$). Altogether, the observations from the GWAS analyses emphasize the need for a deeper investigation of the genetic architecture of the CFHR1/CFHR4/CFHR2/CFHR5 locus with respect to CFHR5 levels and VTE risk. Five additional candidate loci were identified as participating to the genetic regulation of CFHR5 plasma levels: DNAH10, HNF1A, HNF4A, JMJD1C and TRIB1. All 5 loci have been reported to associate with various lipids traits (see GWAS catalogue (Sollis et al., 2022)) and 3 (HNF1A, HNF4A and TRIB1) have been reported to also associate with liver enzymes. DNAH10 is also known to be a locus involved in white & red blood cell biology (Chen et al., 2020) while the JMJD1C is a locus linked to platelet biology (Eicher et al., 2016; Johnson et al., 2010). Most of

these traits are well known risk factors for VTE, and this may suggest that the association of CFHR5 levels with VTE risk implies many additional biological players with pleiotropic effects. This may explain why the MR analyses did not provide causal evidence for a link between CFHR5 levels and VTE risk'

3. They should be particularly weary of potential horizontal pleiotropy in their MR study as I suspect that the lead SNP may also be driving association with other CFHR proteins.

The point raised in this comment is very interesting. Determining the exact genetic architecture that influences each CFHR protein and identifying which variants could have a shared influence on several would be particularly relevant. However, this is slightly out of the scope of the current work, requiring extensive additional investigations, including ideally having access to raw data, not only summary statistics. That said, the additional metaGWAS, colocalization and MR investigations suggest that, indeed, pleiotropy and strong genetic heterogeneity may underly the genetic association between CFHR5 levels and VTE risk. As mentioned, most of the loci we now identified as associated with CFHR5 levels are also associated with other candidate risk factors for VTE, suggesting some pleiotropic effects holds in the relationship between CFHR5 levels and VTE. However, the MR analysis suggests that strong heterogeneity holds, and could be a more important concern to be understood. Possible explanations for the presence of such heterogeneity could be related to the use of different assays to measure CFHR5 introducing heterogeneity in the measurements (the correlation between Somalogic and Olink CFHR5 measurements was only ~0.35, see section *Genome Wide Association Study on CFHR5 plasma levels* in Materials and Methods). It could also relate to the heterogeneity of the VTE phenotype that was analyzed in the GWAS context, as it includes both pulmonary embolism and deep vein thrombosis, provoked and unprovoked events.

4. In the GWAS, which is really helpful, the authors identify a hit near CFHR5. This is compelling and strongly suggests that their assay is measuring a CFHR protein. However, the region is gene rich and seems to have several genes that are also CFHR genes. Are these gene duplications? Is it possible that their assay is measuring other CFHR proteins?

As the reviewer points out, the *CFHR 1-5* genes in this cluster contain several repeating regions. These are believed to have resulted from genomic duplication events, leading to the production of CFHR proteins with partly similar domains. To verify that the quantitative assay we used to analyse the different cohorts specifically measure CFHR5, we used three additional verification strategies:

1. Immuno-capture mass spectrometry using the CFHR5 detection antibody

IC-MS using the monoclonal anti-CFHR5 detection antibody (MAB3845). We found that only CFHR5 (and none of the other CFHR1-4 proteins) was captured by this monoclonal detection antibody (Figure S2 for review A.i). MPG, which is not located on the same chromosomal region, achieved a borderline z-score, but this signal was also detected in the negative control mouse IgG (Figure 1 for review A.ii), and therefore reflects unspecific background binding. Thus, the three different detection antibodies used to create the dual binder assays in combination with this monoclonal anti-CFHR5 (MAB3845) capture antibody (original Figure 1G.i-iii), could only generate a signal in response to the recognition of the target CFHR5.

Figure 1 for review [new figure S2A in revised manuscript]. Immunocapture mass spectrometry analysis of monoclonal anti-CFHR5 detection antibody (MAB3845) and negative control (Mouse IgG).

We have added the following text to the results section [lines 200-202]:

'We confirmed that detection antibody (MAB3845) specifically bound CFHR5 in plasma, using IC-MS analysis (Figure S2A).'

We have added the following text to the methods section [lines 754-767]:

'Immunocapture mass spectrometry (IC-MS)

IC-MS was performed in triplicate, as previously described (Bruzelius et al., 2016a) using the HPA059937 antibody (Atlas Antibodies) or MAB3845 (RnD Biosystems) and rabbit or mouse immunoglobulin G (rlgG, AB-105-C [RnD] and PMP01X [Biorad], respectively) as respective negative controls. In brief, samples were treated in 10 mM dithiothreitol followed by 50 mM chloroacetamide. Overnight sample digestion at 37 °C using Trypsin, was quenched with 0.5% (v/v) trifluoroacetic acid. Digested samples were analyzed using an Ultimate 3000 RSLC nanosystem (Dionex) coupled to a Q-Exactive HF (Thermo). Resulting raw files were searched using the engine Sequest and Proteome Discoverer platform (PD, v1.4.0.339, Thermo Scientific and Uniprot whole human proteome [20180131, for HPA058337], or MaxQuant (Cox et al., 2011) (v. 2.1.4.0) against whole human proteome (UniProt,20210811, for MAB3845) using default settings and label-free quantification. An internal database containing the most common proteins detected by IC-MS in plasma was used to calculate Z-scores (Fredolini et al., 2019).'

2. Western blot for detection of recombinant CFHR5 and CFHR5 in plasma

We also performed Western blot using recombinant (r)CFHR5, normal plasma, and plasma depleted of the 14 most abundant plasma proteins (Figure 2 for review). The monoclonal detection antibody (MAB3845), and the capture anti-CFHR5 antibody used in the quantitative assay (HPA072446), bind both the monomeric and dimeric form of rCFHR5, and the anti-CFHR5 capture antibody (HPA072446) detected a band of corresponding size in plasma. The antibody used in the original discovery screen (with predicted target SULF1) detected the monomeric form of rCFHR5 under non-reducing, but not reducing, conditions, consistent with an off-target binding to an epitope created by a tertiary folded structure of CFHR5. Thus, we can confirm that the dual binder assay used in the quantification analyses measures both the recombinant protein standard and the CFHR5 in plasma.

Figure 2 for review [new Figure S2B in revised manuscript]. Immunodetection of CFHR5 by Western blot. Recombinant CFHR5 (rCFHR5, 100 ng), normal plasma (NP, 1 µl) and 14 most abundant proteins depleted plasma (DP, 10 µl) loaded on SDS PAGE in non-reducing (without DTT, NR) or reducing conditions (with DTT, R). After electrophoresis and transfer on PVDF membrane, protein was detected using antibodies HPA059937 (original screening antibody designed to detect SULF1) and HPA072446 and MAB3845 (anti-CFHR5). The arrow shows the band size corresponding to CFHR5 protein.

We have added the following text to the results section [lines 202-204]:

‘Western blot analysis showed that both MAB3845 and HPA073894 bound mono and homodimer of recombinant CFHR5, and that HPA073894 detected a band corresponding to CFHR5 size in plasma (Figure S2B)’

and [lines 212-215]:

‘Western blot analysis showed that the anti-SULF1 HPA059937 detected the monomeric form of recombinant CFHR5 (rCFHR5) under non-reducing, but not reducing, conditions (Figure S2B), suggesting an off-target binding to an epitope created by a tertiary folded structure of CFHR5’

We have added the following text to the methods section [lines 804-814]:

‘Recombinant CFHR5 (rCFHR5, 100 ng, R&D), normal human plasma (NP, 1 µl, George king) and plasma depleted of the 14 most abundant proteins by depletion spin column (Thermo scientific) (DP, 10 µl) were loaded on SDS PAGE 4-12% (Invitrogen) in non-reducing (without dithiothreitol [DTT], NR) or reducing conditions (with DTT, R). After electrophoresis and transfer onto PVDF membrane, protein was detected using antibodies HPA059937 (original target SULF1), HPA072446 and MAB3845 (both targeting CFHR5). After incubation with horseradish peroxidase (HRP)-coupled goat anti-rabbit or anti-mouse antibodies (1:2000, Dako), bands were detected using chemiluminescence (ECL, Biorad). Molecular weight attributed using PageRuler™ prestained protein ladder (Thermo scientific). WB analysis verified that HPA072446 and MAB3845 bind monomeric and homodimeric form of recombinant CFHR5, and that HPA072446 detects a band in plasma corresponding to CFHR5 (Figure S2B)’

3. Unlabeled MS (data-independent acquisition)

To further validate our quantitative dual binder assay data, we analysed 96 samples in the VEBIOS ER cohort using unlabeled data-independent acquisition mass spectrometry (MS) run on an LC-MS/MS system. This method relies on quantitation of protein sequence specific peptides, generated by trypsin cleavage of plasma proteins, identified through a search using EncyclopeDIA (Searle et al., 2018) against a spectral library generated with a deep learning network Prosit (integrated into ProteomicsDB (Gessulat et al., 2019)) generated with sequences from a set of 2000 *in silico* digested proteins that were previously detected in blood plasma by LC-MS/MS (Geyer et al., 2016). A whole human proteome (Homo Sapiens UniProt ID: #UP000009606, 20,205 entries, accessed 2017-09-18) was used as a background proteome to match the peptide sequences from library to protein IDs and to control for peptide uniqueness between proteins. Protein quantities were calculated using top3 method from the peptide intensities (Silva et al., 2006). Thus, this method provides a totally independent identification and quantification of CFHR5 target that is not reliant on antibody binding and/or specificity.

The quantitative values we obtained based on detection of CFHR5 specific peptides show a high correlation ($\rho = 0.75$, $p < 2.2E-16$) with data generated using our quantitative dual binder assay (Figure 3 for review), orthogonally validating the quantitative performance over the concentration range found in VEBIOS ER samples.

Figure 3 for review [new Figure S2C in revised manuscript]. Relationship between data generated using quantitative dual binding assay (CFHR5 ng/ml) and label-free quantitative data-independent acquisition mass spectrometry data (LFQ-intensity) (ρ =Spearman's correlation coefficient).

We have added the following text to the results section [lines **219-224**]:

'We used data independent acquisition mass spectrometry (DIA-MS) to perform orthogonal validation of the results obtained from the analysis of CFHR5 plasma levels in VEBIOS ER using the dual binder assay with capture antibody HPA072446 (Figure 1G.iii). Data from these two independent assays correlated well ($\rho=0.75$, $p<2.2E-16$), and so the dual binder assay was used for quantification of CFHR5 in VEBIOS ER and an extended sample set of the VEBIOS Coagulation study ($n=284$) (Table S2, Tab 1 for cohort descriptive data).'

We have added the following related text to the methods section [lines **769-802**]:

'Sample preparation

Blood plasma was diluted 10 times with 1x PBS, 1% sodium deoxycholate and amount corresponding to 0.5 μ l of raw plasma processed as described in (Kotol et al., 2021). In brief, samples were treated in 10mM dithiothreitol followed by 50mM chloroacetamide. Digestion was performed overnight at 37°C using Pierce Trypsin Protease (Thermo Scientific, CN: 90057) in enzyme:substrate ratio 1:50 and quenched with 0.5% (v/v) trifluoroacetic acid. The digest was desalted using in-house prepared StageTips packed with Empore C18 Bonded Silica matrix

(CDS Analytical, CN: 98-0604-0217-3) as described in (Rappsilber et al., 2003). Briefly, three layers of octadecyl membrane were placed in 200 μ l pipette tips. The membrane was activated by addition of 100% acetonitrile (ACN) and then equilibrated with 0.1% TFA. Approximately 15 μ g of peptides was added to the StageTip membrane and washed twice with 0.1% TFA. The peptides were eluted in two-step elution with 30 μ l of solvent containing 80% ACN, 0.1% formic acid (FA). In between every buffer addition during the desalting the samples were centrifuged for 2 minutes at 1000xg. Desalted peptides were vacuum-dried and stored at -20°C. Prior to LC-MS/MS analysis samples were dissolved in Solvent A (3% ACN, 0.1% FA) and amount corresponding to approximately 3 μ g of raw plasma subjected to LC-MS/MS analysis.

DIA-MS analysis

The LC-MS/MS analysis was performed using an online system of Ultimate 3000 LC (Thermo Scientific) connected to Q Exactive HF (Thermo Scientific) mass spectrometer. First, the amount corresponding to 3 μ g of raw plasma was loaded onto a trap column (CN: 164535, Thermo Scientific) and washed for 3 minutes at 7 μ l/minute with solvent A. Peptides were separated by a 25 cm analytical column (CN: ES802A, Thermo Scientific) following a linear 40-minute gradient ranging from 1% to 32% Solvent B (95% ACN, 0.1% FA) at 0.7 μ l/minute. The washout of analytical column was performed with 99% B for 1 minute followed by two seesaw gradients from 1% to 99% Solvent B over 4 minutes. Column was then equilibrated for 9 minutes with 99% Solvent A. The MS was operated in DIA mode with each cycle comprising of one full MS scan performed at 30,000 resolution (AGC target $3e^6$, mass range 300-1,200 m/z and injection time 105 ms) followed by 30 DIA MS/MS scans with 10 m/z windows with 1 m/z margin ranging 350-1,000 m/z at 30,000 resolution (AGC target $1e^6$, NCE 26, isolation window 12 m/z, injection time 55 ms).

Data processing

Resulting raw files were converted to mzML format using peak picking filter within ProteoWizard provided software tool msConvert. Resulting mzML files were searched using EncyclopeDIA (Chambers et al., 2012) against a spectral library generated with a deep learning network Prosit, which is integrated into ProteomicsDB (Gessulat et al., 2019). A whole human proteome (Homo Sapiens UniProt ID: #UP000009606, 20,205 entries, accessed 2017-09-18) was used as a background proteome. Finally, the quantification reports were saved, and the protein quantities calculated using top3 method from the peptide intensities (Silva et al., 2006)

Taken together, these results provide strong evidence that our assays specifically measured plasma CFHR5 in the different cohorts/studies, rather than any other of the CFHR (1-4) proteins.

In addition, using exomechip data available in ~1200 participants with measured CFHR5 levels, we observed that 3 rare variants located in *CFHR5* coding regions were associated with CFHR5 levels (see Figure 4 for review), while none of the 5 genotyped rare coding variants in *CFHR2* were associated with CFHR5 levels. This observation (Table S1, Tab 14) provides additional support that the measured protein is indeed CFHR5.

Figure 4 for review [new Figure 4C in revised manuscript]. CFHR5 plasma levels in carriers (N = 16) and non-carriers (N=1214) of rare coding CFHR5 variations. Carriers refer to MARTHA participants harboring the rs41299613-C, rs139017763-A or rs35662416-A rare alleles. Association was tested using linear regression models adjusted for age, sex, anticoagulant therapy, and the allele effect at the lead rs10737681. ***p = 2.45 10⁻⁷

Related to this, we have added the following text to the results section [lines 390-403]:

*Of note, 1230 MARTHA participants with CFHR5 plasma levels have also been typed with an Illumina exome12v1.2 DNA array (Lindström et al., 2019) dedicated to the genotyping of coding polymorphisms, mainly of low frequency. Capitalizing on this additional genetic resource, we investigated whether low-frequency coding variants at the CFHR5 locus (including the nearby CFHR1/CFHR2/CFHR3/CFHR4 genes) could contribute to the inter-individual variability of CFHR5 plasma levels. Twelve rare variants were found polymorphic at this locus in MARTHA participants (Table S1, Tab 14). Three of these variants showed evidence for association with CFHR5 plasma (in bold). These were three rare non-synonymous CFHR5 variants: **rs139017763 (G278S) p=4.75E-05**, **rs41299613 (C208R) p=1.6E-03** and **rs35662416 (R356H) p=7.1E-03**, where rare minor alleles were associated with decreased CFHR5 plasma levels. It is important to emphasize that these 3 CFHR5 non-synonymous variants were carried by 16 distinct individuals. Figure 4C illustrates the difference in CFHR5 plasma levels between the 16 carriers of rare CFHR5-associated variants and non-carriers. This difference remained significant (p=2.45E-07) after adjusting for the common rs10737681 variant identified in the GWAS analysis*

5. In the Biobanque Quebecois COVID-19 cohort (bqc19.ca), we have generated SomaScan v4 data on hundreds of individuals with detailed COVID-19 outcomes. Many samples are longitudinal. The authors can access this data through its data application process and might be helpful, as it will allow for testing of other CFHR proteins. They can also apply for access to the samples if helpful.

We thank the reviewer for this suggestion and this data could certainly be interesting as part of a separate manuscript reporting CFHR5 as prognostic marker of COVID19.

6. The authors state that they retained SNPs with an imputation quality criterion greater than 0.3. Do they mean the info score? If so, please state this.

We indeed retained for GWAS analyses all SNPs with imputation quality criterion r^2 (sometimes referred to as info score) greater than 0.3. This is clarified in the methods section [lines 948-950].

7. How did the authors control for ancestry and relatedness in their GWAS? Why was a random effects model used in GWAMA? Personally, I would use fixed effects.

To address this comment, we have now used a fixed effect model. In the FARIVE, RETROVE and MARTHA, only European ancestry individuals were studied as described in their seminal GWAS papers whose reference have now been provided in the Materials and Methods section. For the deCODE, Fenland and EPIC studies, we used the summary statistics of the corresponding published manuscripts. All these summary data were derived from GWAS adjusted for main principal components. As a consequence, our main finding, that is the association of CFHR5 levels and VTE risk, was mainly conducted in European ancestry populations and would thus deserve further investigations in additional populations of other ancestry origin (also see response below regarding this point).

8. The authors should clarify if their GWAS results are available for sharing. This is a community standard and all summary statistics should be shared through the GWAS Catalog. Could they confirm this will be done upon publication?

We confirm that the results of the extended meta-GWAS analysis will be available on GWAS catalog once the manuscript has been accepted for publication. Results have already been uploaded in GWAS catalog with registration number GCP ID: GCP000508 with an embargo till publication acceptance.

9. Given that ancestry is a known risk factor for VTE, it would be worth mentioning the ancestral background of five cohorts (and samples used for GWASs).

We have added the following text to the discussion section [lines **582-583**]:

‘Our proteomics and GWAS analyses were mainly conducted in European ancestry populations and should be further investigated in populations of other ancestry origin’

Epidemiology

10. Table 2 presents the Odds of VTE (not the risk). More importantly, it might be relevant to consider meta-analysis of these findings across cohorts. Further, in the legend to this table I suggest to provide the covariates that were used in the analysis

We have now included a meta-analysis of the 5 cohorts in (see new Table 2). As BMI data was not available in all of the cohorts, we adjusted for age and sex only, as now mentioned in the table legend.

When samples from cases and controls were stratified according to CFHR5 concentration, the association with VTE was most pronounced in the third tertile, in all 5 cohorts analysed individually and in a meta-analysis (Table 2). These associations remained significant in subgroup meta-analyses when stratified by thrombosis type (DVT or PE), sex, or cause (provoked/unprovoked) (Table S1, Tab 9, Table A-C). In subgroup analyses in the individual cohorts, the association of CFHR5 with VTE did not reach significance in females in VEBIOS Coagulation and FARIVE and in males in DFW-VTE. Furthermore, the association with provoked VTE in RETROVE and with unprovoked VTE in FARIVE were not significant. The results were consistent when further adjusting for BMI and/or CRP when this information was available (Table S1 Tab 10, Table A-D).

11. The narrative of the story could be improved. The authors start by outlining how diagnosis and prediction of VTE would be helpful, but then provide little data to demonstrate that their findings solve this problem. (No AUROC, no AUPRC, no estimates of precision etc...) To this end, such metrics should be reported.

This is an important point which we have now addressed in the revised manuscript. The response below was also provided to Reviewer #2 (point 1) as they raised a similar query:

This is indeed an important point the reviewer has raised. We therefore performed an analysis adding absolute concentrations levels of CFHR5 into a model based on the clinical decision rule (CDR) used in most ER settings today. In the current CDR algorithm, an assessment of clinical probability is first performed using a Wells score, with different score sets used for DVT and PE. In patients with Wells score <2 for DVT and <4 for PE, a VTE diagnosis is considered to be of low probability i.e., 'VTE unlikely'; in these patients a D-dimer test is performed. If this is negative (below age adjusted cut off, i.e., $0.5 \text{ mg/L} + (\text{age}-50)/100$), it is considered safe to rule out VTE without the need for further diagnostic imaging (high negative predictive value), but D-dimer is positive, imaging is used to confirm or exclude a VTE diagnosis. In patients with high probability according to Wells score (≥ 2 for DVT and ≥ 4 for PE), current guidelines dictate that diagnostic imaging should be performed without prior D-dimer testing, as several factors incorporated in the Wells score are associated with increased D-dimer. International studies have shown that less than 10-20% of computed tomography pulmonary angiogram (CTPA) performed on suspicion of PE confirms such a diagnosis (Mittadodla et al., 2013) and thus, particularly in the group 'VTE likely' there is a high overutilisation of imaging within current CDR guidelines.

We performed an analysis to determine the potential application of CRHR5 as a diagnostic biomarker in: (i) the full VEBIOS ER group, and (ii) where patients were stratified according to 'VTE likely' or 'VTE unlikely'. Adding CFHR5 to a base model of D-dimer alone (dichotomized according to age adjusted cutoff) in the full VEBIOS ER group resulted in a non-significant improvement in AUC (0.88 versus 0.82, $p=0.110$). A model based on D-dimer + CFHR5 concentration + Wells score did not perform better than D-dimer alone (AUC 0.86 versus 0.82, $p=0.197$). In the sub analysis, in the 'VTE unlikely' group ($n=43$), adding CFHR5 to D-dimer alone resulted in a non-significant increased accuracy compared to the base model (AUC 0.84 versus 0.81, $p=0.61$). However, in the group representing the main diagnostic challenge in the acute setting, the 'VTE likely' group, the combination of CFHR5 and D-dimer performed significantly better than D-dimer alone (AUC 0.92 vs 0.83; $p=0.035$).

All data from this analysis has been added to a new tab in Table S1 (Tab 5 UAC).

In VEBIOS ER we have not defined cut-off values for CFHR5 concentration to calculate sensitivity and specificity, as additional studies are needed to establish reference range and cut-off value in patients with suspected acute VTE.

We have added the following text to the results section [lines **253-269**]:

CFHR5 measurement can increase diagnostic accuracy in patients with likely VTE

To explore the potential usefulness of CFHR5 as a biomarker to be included in the diagnostic workup of suspected acute VTE, we assessed the discriminatory power of CFHR5 in VEBIOS ER using logistic regression in different models together, with D-dimer dichotomised using current Clinical Decision Rules (CDR) as 'positive' or 'negative' (below age adjusted cut-off (Douma et al., 2010)) and Wells score (VTE likely (≥ 2 for DVT and ≥ 4 for PE) or unlikely) (Table S1, Tab 5). In VEBIOS ER, D-dimer had negative predictive value (NPV) of 100% (0 false negatives) for VTE, while the specificity and positive predictive value (PPV) was only 62.8% and 74% respectively, with 16 false positive cases. Adding CFHR5 to the base model of D-dimer alone resulted in a non-significant improvement in AUC (0.88 versus 0.82, $p=0.110$), as did adding Wells score to the base model (AUC 0.85, $p=0.33$) (Table S1, Tab 5). D-dimer alone performed better than CFHR5 alone (AUC 0.73 versus 0.82, $p=0.128$). When stratifying patients

based on Wells score, in the group where VTE was considered unlikely based on Wells score (n=43), adding CFHR5 to the base model resulted in a non-significant increased accuracy compared to the base model (AUC 0.84 versus 0.81, p=0.61). However, in the group where VTE was considered likely (n=41), the addition of CFHR5 to the base model resulted in a significantly increased accuracy compared to D-dimer alone (AUC 0.92 vs 0.83; p=0.035) (Table S1, Tab 5)

We have added the following text to the discussion section [lines **618-627**]:

‘Current clinical decision rule (CDR) in diagnostic workup of suspected acute VTE is based on age adjusted D-dimer and Wells score. In VEBIOS ER we found that adding CFHR5 to D-dimer increased diagnostic accuracy of acute VTE in the VTE-likely group (Wells score ≥ 2 for DVT and PE). This group represents the major diagnostic challenge, as an elevated D-dimer is common in several of the conditions associated with increased risk for VTE, e.g., cancer and surgery, both of which are included in Wells score. Therefore, according to current CDR, patients with high clinical probability based on Wells score proceed to diagnostic imaging without prior D-dimer testing (Kline, 2020; Zarabi et al., 2021). Thus, adding CFHR5 concentration to D-dimer in the diagnostic work-up could potentially reduce number of negative imaging procedures, to the benefit of patients and health care system’

We have added the following text to the methods section [lines **876-886**]:

‘Discriminatory accuracy of plasma concentrations of CFHR5 and of D-dimer categorized as ‘positive’ or ‘negative’ using age adjusted D-dimer cutoff (Douma et al., 2010) in the different models was assessed using logistic regression analysis and presented as Area Under the Receiver Operating Characteristics curve (AUC). Statistical analyses were performed using R version 4.0.3. ROC curves for the different biomarker-based risk models based on plasma concentration CFHR5, dichotomized data on D-dimer (positive or negative) and Wells score (‘VTE likely’ (≥ 2 for DVT and ≥ 4 for PE) or ‘VTE unlikely’) were compared using the function roc.test (Delong’s test) in the RStudio attachment. All tests were two-tailed’

12. There is no reference for the “Respiratory Index”. While this score seems reasonable to me at face value, is this a score accepted by the community? If not, why not use another scoring system like the WHO COVID severity score?

The general consensus between the reviewers was that the section on the analysis of plasma from COVID-19 patients does not significantly add to the manuscript in its current form, and we have therefore removed it.

13. While I lack expertise in measurement of proteins, is it not somewhat concerning that SULF1 is detected by the discovery antibodies? Would it not be possible to assess publicly available data from different assays to see if the results are concordant? For example, if the SomaScan assay also found an association between CFHR5 and VTE, but a lack of association of SULF1 with VTE, that might be helpful.

We agree with the reviewer that such data could be helpful to rule out that SULF1 is also a VTE associated protein. Unfortunately, an assay for SULF1 is not included in the Olink and SomaScan panels that have been used to generate existing public datasets, and it is thus not possible to test this using these orthogonal methods. As described in the original manuscript, we attempted to establish dual binder assays for SULF1 using five different detection antibodies (three from commercial sources) together with HPA059937 as capture antibody, however no combination gave a quantitative signal, either in serial dilutions of plasma or recombinant SULF1 protein. While this would not exclude that SULF1 still could bind and contribute to the MFI values generated for HPA059937 in the original discovery screen, the strong correlation between MFI values obtained with the single binder assay in that screen and those obtained with dual binder assays using either HPA059937 or the anti-CFHR5

HPA072446 as capture antibodies together with anti-CFHR5 MAB384 as detection antibody (now confirmed by IC-MS to be CFHR5 specific, as described above), strongly suggest that SULF1 did not contribute to the VTE association obtained with HPA059937 in the discovery screen.

14. In general, the epidemiology work done requires further thought. The covariates used to calculate the ORs should be listed in the text and table, comparing CFHR5 levels between cases and controls. If no covariates were used, why aren't the authors worried about confounding? For example, if CFHR5 levels are associated with D-dimer levels in cases, could this not influence its association with VTE via confounding?

We agree with the reviewer that potentially confounding covariates such as BMI and CRP, are relevant to adjust for when available. However, we do not have the same information of covariates in all 5 cohorts listed in Table 2. In FARIVE and RETROVE, and in a large proportion of the controls in the DFW-VTE cohort (83 of 146), we lack information for CRP. In DFW-VTE, and a proportion of VEBIOS ER(33 of 96), we also lack information for BMI. Therefore, in Table 2, including the meta-analysis, we present results based on age and sex adjusted analyses only. However, to enable the reader to assess the effect of adjusting for BMI and/or CRP, we have now performed individual analyses (including sub-analyses stratifying for sex, PE/DVT, provoked/unprovoked) of each cohort adjusting for covariates when possible. Results are consistent compared to when only adjusting for age and sex, although statistical significance is reduced or lost in some analyses, likely due to the reduced number of samples for which information was available (new data can be found in Table S1 Tab 10, Table A-D).

In results we have added [line 346-347]:

“The results were consistent when further adjusting for BMI and/or CRP when this information was available (Table S1 Tab 10 Table A-D)”

In acute VTE, as a fibrin breakdown product, D-dimer becomes markedly elevated as a consequence of clot formation and fibrinolysis, with D-dimer levels heavily influenced by factors such as clot burden and stability. D-dimer is not known to have a causal role in thrombosis, and there are to our knowledge, no studies indicating that D-dimer levels directly induce or modulate expression of liver proteins and/or complement e.g., CFHR5. Indeed, the lack of correlation between D-dimer and CFHR5 in VEBIOS ER is not consistent with such mechanisms. Therefore, we did not consider it a potentially confounding factor in acute VTE (VEBIOS ER and DFW-VTE).

The point of the reviewer regarding D-dimer is however very relevant in the non-acute VTE cohorts. In the absence of an acute thrombosis, elevated D-dimer reflects increased activity of the coagulation system, consistent with that modestly increased D-dimer levels have been associated with increased risk of VTE recurrence (Palareti et al., 2006). Adjusting for D-dimer as a proxy marker for the combined effect of factors that result in increased activity of coagulation system, could potentially capture confounders that contribute to the procoagulant phenotype of blood. We performed analyses adjusting for D-dimer in the three cohorts where this data was available (Table 2 for revision).

	VEBIOS ER n = 48 cases + 48 controls					VEBIOS Coagulation n = 142 cases + 135 controls					DFW n = 54 cases + 146 controls				
	OR	(95% CI)	pval-glm	NA	DF	OR	(95% CI)	pval-glm	NA	DF	OR	(95% CI)	pval-glm	NA	DF
Case/Control + SA	2,5	(1.52-4.66)	1,1E-03	0	92	1,55	(1.20-2.01)	8,8E-04	7	273	1,8	(1.29-2.57)	7,7E-04	0	196
Case/Control + SA + D.Dimer	1,87	(0.96-4.03)	8,5E-02	8	83	1,44	(1.11-1.88)	7,7E-03	12	267	1,25	(0.77-2.00)	3,6E-01	0	195
Case/Control + SA	111	(13.61-2190.27)	2,5E-04	8	84	2,29	(1.36-4.22)	4,5E-03	12	268	8942	(287.88-6.7E+05)	4,6E-06	0	196
Case/Control + SA + CFHR5	59,4	(7.81-1136.27)	1,3E-03	8	83	2,04	(1.23-3.72)	1,3E-02	12	267	7028	(226.71-5.43E+05)	9,2E-06	0	195

Table 2 for revision. Association of CFHR5 or D-dimer with VTE when adjusting for each other. SA: Sex and age adjusted; OR: Odds ratio; pval-glm: p-value, general linear model; (95% CI): 95% confidence interval; NA: not available [number of samples for which data was lacking]; DF: degrees of freedom

When adjusting for D-dimer in VEBIOS Coagulation, CFHR5 was still significantly associated with VTE (OR 1.44 [1.105-1.88], $p=7.7E-03$). Vice versa, when adjusting for CFHR5, D-dimer levels remain associated with VTE (OR 2.04 [1.23-3.72], $p=0.013$).

When adjusting for D-dimer levels in the acute VTE cohorts, CFHR5 is no longer significantly associated with acute VTE diagnosis in VEBIOS ER (OR 1.868 [0.96-4.03] $p=0.085$) or DFW-VTE (OR 1.246 [0.77-2.00] $p=0.358$), which is not unexpected as D-dimer is positive in all (++) cases (i.e., above age adjusted cutoff in D-dimer concentration) and strongly associated with diagnosis of acute VTE also when analysed as continuous variable (see table above). D-dimer remain highly associated with acute VTE in both VEBIOS ER (OR 59.4 [7.8-1136] $p=0.00125$) and DFW-VTE (7028 [226.71-5.43E6], $p=9.15E-6$) when adjusting for CFHR5 levels.

We have in Table S1 Tab 10 Table A included amended results for adjusting for D-dimer together with age and sex in VEBIOS ER, DFW-VTE and VEBIOS Coagulation.

We have added the following text to the results section [lines **249-252**]:

'The association between CFHR5 and VTE remained significant in VEBIOS Coagulation when adjusted for CRP (OR=1.55 [CI 1.18-2.03] $p=1.50E-03$) or D-Dimer (OR=1.435 [CI 1.05-1.88] $p=7.72E-03$) (Table S1, Tab 10 Table A)'

15. In another example, on the MARTHA cohort, adjustments were made for family history of VTE. Why? Why adjust for provoked, or unprovoked VTE?

The unprovoked/provoked status of a first VTE event is known to be associated with the risk of recurrence, e.g., (Iorio et al., 2010; Kearon et al., 2016). Therefore, when we performed analyses to determine association between CFHR5 plasma concentrations and VTE recurrence, for which the MARTHA cohort was used, we adjusted for this variable.

16. In another example: In the "ANALYSIS OF PLASMA BY TARGETED PROTEOMICS" section, the authors did not adjust for BMI (lines 258–259). However, in other sections, such as "Analysis of CFHR5 in COVID-19 patients in the COMMUNITY study" (line 312) and "CFHR5 and risk of recurrent VTE" (line 581), they adjusted for BMI. I recommend that they consistently adjust for BMI and other covariables. If it was not possible for some analyses due to lack of data, they should clearly state that.

We did not adjust for BMI in the proteomics screening discovery phase analysis of VEBIOS ER as BMI data was not available for 7 cases and 26 controls (34% total) as this information is not registered for every patient admitted to the Emergency Room. As suggested by the reviewer, we have now more clearly stated this in the methods section [lines **864-865**]:

'BMI data was lacking for 33 of the 96 patients (7 cases and 26 controls), so we did not adjust for BMI in the discovery analysis'

For the purpose of the review process, we re-analysed data from the 41 cases and 22 controls for which BMI was available. When adjusted for age, sex and BMI, the signal generated by the antibody proposed to target SULF1 (later determined to bind CFHR5) remained significantly associated with VTE in both citrate and EDTA anticoagulated blood (both $p<0.001$). The other 3 targets we originally identified as VTE associated (Figure 1B and C): SELP1, CD47, ADORA2, were also significantly associated with VTE in both citrate and EDTA anticoagulated blood in this alternative analysis, albeit above the threshold we selected for classification ($p<0.01$). None of the signals from the other 752 antibodies passed the predefined threshold in the discovery phase. Thus, the antibody originally designed to target SULF1 would still emerge as the only candidate overlapping both VEBIOS ER and VEBIOS coagulation analysis when analysis was adjusted for BMI.

17. In the “CFHR5 is associated with VTE independent of C3” section (starting from line 534), they used 0.05 as an absolute cut-off to declare CFHR5’s dependent/independent association with C3, but I find this misleading. The p-value differences is minor (0.032 vs 0.0645), and $p = 0.0645$ does not support independence. It just means that it was not statistically significant. So I suggest wording this section more carefully, especially given the relatively small sample size. Besides, since the authors indicated that CFHR5 is associated with C3 in other sections (e.g., Fig2. protein-protein interaction analysis, line 605–606), the author should elaborate on these seemingly contradicting claims.

We agree with the reviewer that the wording should be more carefully handled here, so it is clear that CFHR5 is independently associated with acute VTE only (VEBIOS ER) and not prior VTE (VEBIOS Coagulation). We have amended the section title and text to clarify, presenting the results for the two cohorts separately, emphasizing that the association of C3 or CFHR5 with VTE in VEBIOS Coagulation is weakened when each is adjusted for the other, but we do not claim these relationships are independent from each other.

We have modified the text in the results section to clarify this [lines **303-323**]:

‘CFHR5 is associated with *acute VTE* independent of C3

Plasma levels of complement component C3 have previously been reported as associated with incident VTE (Norgaard et al., 2016). To determine if the association between CFHR5 and VTE we observed is dependent on the concentration of C3, we developed an in-house dual binder quantitative assay to measure C3 in the VEBIOS ER and VEBIOS Coagulation cohorts. In VEBIOS ER, plasma C3 was not elevated in cases, compared to controls (Figure 2D.i and S1 Tab 2, panel B), CFHR5 and C3 did not significantly correlate in either group (Figure 2D.ii) and C3 was not associated with VTE (OR 1.04 [CI 0.68-1.58], $p=0.86$) (Table S1, Tab 8). *Furthermore, the association with acute VTE for one SD increase in CFHR5 level remained unchanged (OR 2.65 [CI 1.53-5.01], $p=1.26E-03$) when including and adjusting for C3 concentration (together with age and sex), compared to when only adjusting for age and sex (OR 2.54 [1.52-4.56], $p=0.001$, [Table 2, Table S1, Tab 8]), demonstrating that CFHR5 is independently associated with acute VTE.* In VEBIOS Coagulation, C3 levels were higher in plasma from cases, compared to controls (Figure 1E.i), and CFHR5 and C3 correlated with each other in both ([controls $\rho = 0.46$ $p<0.0001$], [cases $\rho = 0.47$ $p<0.0001$]). *After adjusting for age and sex, one SD increase in C3 level was significantly associated with previous VTE (OR 1.52 [CI 1.18-2.01], $p=1.93E-03$). When adjusting for CFHR5 levels (together with age and sex), this no longer reached significance (OR 1.31 [CI 0.99-1.78], $p=0.064$). The association with previous VTE for one SD increase in CFHR5 level in VEBIOS Coagulation was still nominally significant when adjusting for C3 levels (OR 1.36 [CI 1.03-1.82], $p=0.032$), although weaker compared to adjusting only for age and sex (OR 1.55 [1.2-2.01], $p=8.85E-04$, Table 2, Table S1, Tab 8)‘*

In the discussion we have omitted the original sentence:

‘...where C3 is a proxy for CFHR5, where CFHR5 represents the functional link to VTE risk, rather than C3 levels per se. Consistent with this idea, in the VEBIOS coagulation study, the association between CFHR5 and previous VTE remained significant when adjusted for C3.’

...and modified the text to now state the following [lines **524-528**]:

‘It could be speculated that the association of C3 with VTE in individuals sampled pre-VTE (Norgaard et al., 2016) or following treatment for a prior VTE, reflects co-regulation of CFHR5 and C3 under basal conditions (but not in the acute phase), which would be consistent with our finding that in VEBIOS coagulation, the association with VTE for CFHR5 was weaker when adjusting for levels of C3, and vice versa.’

The STRING protein-protein interaction analysis is based on several parameters, including evidence of co-expression, but also incorporating other data types, such as evidence of interaction from experimental work or curated databases and text mining of the existing literature. Thus, while such information can be useful to identify possible protein-protein links, it cannot be used to infer relative plasma protein levels. Similarly, the predicted co-expression of C3 and CFHR5 mRNA in liver hepatocytes does not necessarily translate into a similarity in plasma levels, as this analysis does not account for several key post-transcriptional factors, such as translation efficiency, cellular release dynamics, protein stability, clearance etc.

We have modified the text in the results section to clarify this [lines **295-297**]:

'While these data indicate a degree of co-expression with CFHR5 at the transcriptional level, plasma concentrations of the encoded proteins are subject to several post-transcriptional variables, such as translation efficiency, cellular release dynamics, protein stability and clearance'

18. This is a very awkward phrase to read in English: we found CFHR5 levels at baseline were associated with short-time prognosis of disease severity. What does "short-time prognosis of disease severity" mean? I think you mean maximum level of respiratory support. If so, why not just say that?

As the general consensus between the reviewers was that the COVID-19 section did not significantly add to the manuscript in its current form, we have therefore removed it.

19. Figures. I suggest to have less dense figures. For example, Fig 1H iii presents data that is very small. Why not have more figures and let the reader be able to see them?

We are somewhat restricted in this respect, due to the figure limit of the publisher. However, we agree that Figure 1 is rather dense, and to facilitate the reader we have enlarged the minimum text size from point 4 to point 6.5 (affecting Figure H and J, panels ii and iii), to increase readability. In addition, we have ensured that all key information is also presented in the associated text figure legends or tables.

20. Please provide QC procedures for genotyping MARTHA, rather than a reference for this. Why were patients with SLE removed?

By original design, exclusion criteria for MARTHA patients included protein S, protein C or antithrombin deficiency, lupus anticoagulant, or homozygosity for the F5L or F2 G20210A mutations, to exclude participants with known thrombophilia.

We have added the following QC details to the methods section [lines **936-946**]:

'MARTHA participants were genotyped with the Illumina bead arrays (Sennblad et al., 2017). Quality control procedures were as previously described (Antoni et al., 2011; Germain et al., 2015; Germain et al., 2011). Briefly, SNPs showing genotyping call rate <99%, significant ($p < 10^{-5}$) deviation from Hardy-Weinberg Equilibrium (HWE), with minor allele frequency (MAF) less than 1% in were filtered out. Individuals were excluded based on the following criteria: (i) genotyping success rates <95%, (ii) close relatedness as detected by pairwise clustering of identity by state distances (IBS) and multi-dimensional scaling (MDS) using PLINK, (iii) genetic outliers using principal components approach as calculated by EIGENSTRAT. After application of quality control filters, 1525 participants remained for association testing with CFHR5 plasma levels. We imputed genotypes by using mach version 1.0.18.c and haplotypes from the 1000 Genomes Total European Ancestry (EUR) population (August 2010 release)'

Minors:

21. Line 163–164 (and other sections mentioning Well’s criteria): For Well’s criteria, the authors should clarify the cut-off they used.

We have included the cutoff used for DVT (≥ 2 points) and PE (≥ 4 points) in the relevant results and methods sections.

22. Line 176–177: Given matching cases and controls can greatly affect the results, especially when the sample size is relatively small, I recommend more elaboration on how the authors matched cases and controls.

Each control in VEBIOS ER was selected to match a case on sex and then as close as possible by age. In women, the mean difference in age between case and control was 0.95 years. In men the mean difference between cases and controls was 3.65 years.

We have clarified this in the methods section [line **666**]:

‘For the present study, 48 cases were available for analysis and 48 controls were matched by sex, and as closely as possible by age (mean age difference [years] cases vs. controls, women: 0.95, men: 3.65)’

23. For Table S1 Tab 1–S5, statistical testing is needed to evaluate baseline differences between cases and controls (adding a p-value column).

As the reviewer suggested, in Table S2 Tab 1-5 (former Table S1) we have added a column for p value to the characteristics of the cohorts. Baseline differences were evaluated by Student t-test and Pearson's Chi-squared Test for numerical and categorical variables, respectively.

24. It would also be helpful if the authors provided an additional table listing all five cohorts’ characteristics, comparing their differences.

We have added an additional tab to Table S2 (Tab 6), where data from all cohorts is collated into a single table.

25. Line 201: “All participants were free from cancer.” Is this correct? Or does this mean that they were not in the active treatment of cancer?

All patients were free of known or discovered active cancer at the time of the thromboembolic episode were excluded. Patients treated effectively by surgery, certified or radiotherapeutic for more than 5 years before inclusion in the study, and without recurrence, could be included.

We have added text to clarify this into the methods section [lines **691-693**]:

‘All patients were free of known or recently discovered cancer at the time of VTE diagnosis. Patients treated for cancer >5 years before the episode without recurrence could be included’

26. Line 312, 331 (and other sections mentioning computational tools): I recommend the authors include version information.

Version information has been added for R statistical computing software and GraphPad Prism [lines 870-871].

27. In the Methods section, the authors should clearly state the cut-off for each of the statistical testing performed, including the one for FDR.

We have added the following sentences to the respective method sections:

Analysis of plasma by targeted affinity proteomics [lines **752-753**]:

'A significance threshold of $p < 0.01$ in both EDTA and Citrate plasma was used as selection criteria'

Immunocapture mass spectrometry (IC-MS) [lines **766-767**]:

'A z-score of ≥ 3 , corresponding to a p-value < 0.01 [Confidence level 99%], was used as cut-off'

Gene ontology and reactome analysis, and the generation of respective FDR values, was performed using <http://geneontology.org/docs/go-enrichment-analysis/>. In Table S1, we reported the top six overrepresented groups and the associated PFDR values (all PFDR $< 1.0E-10$ for gene ontology and PFDR $< 1.0E-05$ for reactome), and thus didn't apply a 'cut off' threshold value as such.

We have added the following sentences to the method section [lines **913-916**]

'Gene ontology and reactome analysis was performed using (<http://geneontology.org/docs/go-enrichment-analysis/>) (Ashburner et al., 2000; Gene Ontology, 2021), PFDR values for the top six enriched over represented terms in each category are provided in Table S1, Tab 6'

28. Line 472–474: I suggest the authors include these data (quantitative signal) in Supplementary materials.

None of the dual binder assays for SULF1 produced a detection signal significantly above that of the background, when tested either with recombinant SULF1 or in serial dilutions of plasma, and so we do not have any quantitative data. We have clarified this in the results section [lines 209-212].

29. Line 518: Since FDR stands for false discovery rate (e.g., 5%), "FDR = $2.4E-16$ " may be inappropriate. It should instead be written as FDR-adjusted P-value (PFDR) or q-value.

We have modified the terminology as suggested [lines **284-285**].

30. Line 582–584: P-values for subgroup analyses should be shown next to HR and its confidence intervals. As none of them reached a conventional statistical threshold of 0.05, the authors should be careful in interpreting these associations. E.g., I found the term "this association is driven by" too strong since the primary analysis was not statistically significant.

We have modified the results section [lines **354-359**].

'The association was consistent between females (HR=1.1 [0.90 -1.38], $p=0.320$) and males (HR=1.14 [0.91 - 1.44], $p=0.260$) and between patients with DVT (HR =1.18 [0.98 -1.42], $p=0.080$) or PE as first event (HR=1.13 [0.80 - 1.61], $p=0.489$). This trend for association was driven by strongest in the subgroup of patients with unprovoked first VTE (HR=1.32 [0.99–1.77], $p=0.056$), as no association was observed when the first event was provoked (HR=1.01 [0.83–1.23], $p=0.900$)'

31. Line 587: Typo. Testhomogeneity -> test of homogeneity

This has now been corrected - thanks for the attention to detail.

32. Line 634: Was CFHR5 associated with any clinical measurements/features/symptoms that suggest thrombosis formation/VTE events in these COVID-19 patients? I understand sample size can be an issue, and if there are not sufficient data available, the authors do not have to address this.

The general consensus between the reviewers was that this section does not significantly add to the manuscript in its current form, and we have therefore removed it.

33. Line 748: The reference is not properly formatted.

This reference has now been removed, as it was related to the COVID-19 data.

Reviewer #4 (Remarks to the Author):

VTE is an important disorder with high morbidities and mortalities. Although several risk factors and mediators have been established its pathogenesis is still not completely understood. In the present study the authors provide several lines of evidence supporting an important role of CFHR5 in this disorder. The finding was first detected in a discovery proteomic study and thereafter confirmed in five replication cohorts including a total of 1137 patients and 1272 controls. A role of CFHR5 was also supported by GWAS analysis of 2967 individual. They also showed that recombinant CFHR5 promote platelet activation in vitro. Their findings are novel and the use of several replication cohorts and in vitro studies clearly strengthen their findings. Their study have also some important limitations.

1. Although mentioned is some of the study studies, the author should in all studies the relationship of CFHR5 to provoked and spontaneous VTE and to DVT and PE, separately.

We have included these analyses separately for each cohort into Table S1, Tab 9 (Table A and C) so the reader can assess the results in each separate cohort. However, it results in small sample sizes in some sub-analyses, and hence lacks power in there.

2. In some sub-studies they perform adjustment for other risk factor (e.g., BMI, aged, gender, CRP, co-morbidities), but this should be consistently be performed in all sub-analyses.

This point was also raised by reviewer 3. We agree with the reviewers that potentially confounding covariates such as BMI and CRP, are relevant to adjust for when available. However, we do not have the same information of covariates in all 5 cohorts listed in Table 2. In FARIVE and RETROVE, and in a large proportion of the controls in the DFW-VTE cohort (83 of 146), we lack information for CRP. In DFW-VTE, and a proportion of VEBIOS ER (33 of 96), we lack information for BMI. Therefore, in Table 2, including the meta-analysis, we present results based on age and sex adjusted analyses only. However, to enable the reader to assess the effect of adjusting for BMI and/or CRP, we have now performed individual analyses (including sub-analyses stratifying for sex, PE/DVT, provoked/unprovoked) of each cohort adjusting for covariates when possible. Results are consistent compared to when only adjusting for age and sex, although statistical significance is reduced or lost in some analyses, likely due to the reduced number of samples for which information was available (new data can be found in Table S1 Tab 10, Table A-D).

In results we have added [line **346-347**]:

'The results were consistent when further adjusting for BMI and/or CRP when this information was available (Table S1 Tab 10 Table A-D)'

3. In the statistical approach they could use multivariable Cox proportional hazards models to calculate HR. In fact, I cannot find a thorough description of the statistical methods.

We have added the following details to the methods section:

'Association of CFHR5 levels with VTE recurrence in the MARTHA cohort was assessed using a Cox survival model with left truncature at age at sampling. Analysis was adjusted for sex, familial history of VTE, provoked or unprovoked status of the first VTE, age at first VTE, and BMI, and were conducted using the Survival R package' [lines **872-875**]

'The heterogeneity of the association between CFHR5 and VTE (recurrence) according to specific subgroups was assessed using the Cochran-Mantel-Haenszel statistical test (MANTEL and HAENSZEL, 1959)' [lines 875-877]

4. As mentioned by the authors, CFHR5 will inhibit FH and thereby enhance activity of the alternative complement pathway. Have the authors measured the activity in the alternative and terminal pathway for example by measuring C3bc, detecting the C3b, iC3b and C3c fragments, the C3 convertase C3bBbP and C5b-9/TCC at least in a subgroup of patients.

This is an interesting idea. As suggested, we selected samples from a subgroup of VEBIOS ER samples, based on them having the (i) highest or (ii) lowest CFHR5 concentrations in the cohort. We then measured the concentration of complement factor C3c in each group. Mean C3c concentration was highest in the high CFHR5 group (C3c (µg/ml) ± std dev: CFHR5 low: 5.48±1.2 vs CFHR5 high: 6.48±1.2) (Figure 1 for review A), but there was a lack of statistical significance between the two groups by unpaired t-test (p<0.086). However, C3c and CFHR5 concentrations were positively correlated across the sample set (Figure X for review B).

Figure 1 for review [new Figure S6 in revised manuscript]. (A) C3c concentrations in the low CFHR5 group (<2500 ng/ml), vs. the high CFHR5 group (>3800 ng/ml) and **(B)** the correlation between C3c and CFHR5 concentration in all measured samples. (ρ=Spearman’s correlation coefficient).

We have added the following text to the methods [lines 845-848]:

Concentration of complement fragment 3c (C3c) in 20 VEBIOS ER cohort samples, selected based on low (<2500 ng/ml) or high (3800 ng/ml) plasma concentrations of CFHR5 were measured using a commercial sandwich C3c ELISA kit (Nordic BioSite, Sweden). Samples were measured in duplicate and C3c concentration calculated using a standard curve

We have added the following text to the results [lines 446-453]:

Complement fragment 3c concentration correlates with CFHR5 in VEBIOS ER subset
As a marker for C3 cleavage and activation in plasma, we measured complement fragment 3c (C3c) in a subset of plasma samples from VEBIOS ER, selected based on a low (<2500 ng/ml, 10 samples) or high (>3800 ng/ml, 10 samples) plasma CFHR5 concentration. Mean C3c concentration was greatest in the high CFHR5 group (C3c (µg/ml) ± std dev: CFHR5 low: 5.48±1.2 vs CFHR5 high: 6.48±1.2), although this difference failed to reach statistical significance (p<0.086) (Figure S6A). However, C3c and CFHR5 concentrations were positively correlated across this sample set (corr.0.51, p<0.02) (Figure S6B).

We have added the following text to the discussion [lines 517-524]:

Elevated plasma C3 in baseline samples has been shown to be associated with increased risk of future VTE (Norgaard et al., 2016). Consistent with these findings, C3 was associated with prior VTE in the VEBIOS coagulation study, but not with acute VTE in the VEBIOS ER study.

In both cases, and in the previous study by Nordgaard et. al. (Nordgaard et al., 2016), total C3 level, rather than the active form (C3a) is measured; it is possible that in acute VTE, regulation of C3 convertase (by CFHR5) is important, rather than absolute C3. Consistent with this, we observe a trend for higher plasma levels of complement C3c fragment, a marker of C3 activation, in samples with higher CFHR5 concentrations at VTE diagnosis.

5. The relevance of platelet activation for VTE is questionable. It would have strengthened their findings if they could show a pro-coagulant effect of recombinant CFHR5 on endothelial cells.

To address this point, we performed an assay we previously developed in house, to determine the thrombin generation potential on endothelial cells isolated from fresh tissue (human umbilical cord endothelial cells, HUVECs) (Figure 2 for review only). Briefly, HUVECs from 2 donors were cultivated until confluency in M199 on flat-bottom 96 well plates (Falcon, C-treated culture plate), stimulated with the inflammatory cytokine tumour necrosis factor alpha (TNF; 10 ng/ml), rCFHR5 (6 µg/ml), both or neither, for 24h before thrombin generation assay. After washing the cells with PBS and saturation with 3% BSA, thrombin formation was initiated in 120 µL reaction mixtures containing human citrated plasma (George King), 4 µM phospholipids, 16.6 mM Ca²⁺ + and 2.5 mM fluorogenic substrate (Z-Gly-Gly-Arg-AMC) (Thrombinoscope BV, Diagnostica Stago). As thrombin generation controls, plasma Tissue factor (1 pM, Dade Innovin, Siemens) and mouse monoclonal anti-TF antibody (12.5 µg/ml, HTF-1, BD Pharmingen) were added to plasma (Data not shown). All real time thrombin formation were run in duplicate. Thrombin generation was quantified using the Thrombinoscope software package (Version 3.0.0.29).

Figure 2 for review only. HUVECs were cultivated until confluency before they were treated with a PBS control or (TNF; 10 ng/ml), with or without rCFHR5 (6 µg/ml) for 24h. Thrombin generation was measured and (A) peak thrombin generation, (B) time to peak and (C) lag time were calculated.

In addition, we screened for mRNA expression level of a panel of endothelial cell genes with established roles in coagulation (under the same treatment conditions described above). We did not observe any marked differences in the expression of these genes in the presence of CFHR5 (Figure 3 for review only).

Figure 3 for review only. HUVECs were cultivated until confluency before they were treated with a PBS control or (TNF; 10 ng/ml), with or without rCFHR5 (6 µg/ml) for 24h qPCR was subsequently performed using target primers against endothelial cell genes (A) *F3* (tissue factor), (B) *THBD* (thrombomodulin), (C) *PLAT* (plasminogen activator, tissue type), (D) *TFPI* (tissue factor pathway inhibitor), (E) *VWF* (von Willebrand Factor), (F) *ICAM1* (intracellular adhesion molecule 1) or (G) *CXCL8* (interleukin 8).

Thus, we did not observe any results to indicate that rCFHR5 had a procoagulant effect on endothelial cells. Due to space restrictions, we did not include this data in the manuscript revision.

As also described in the response to reviewer #2 (point 3F), we have addressed the relevance of platelets in VTE in the discussion [lines 529-539]:

The mechanisms underlying venous and arterial thrombosis development differ; venous thrombi contain an abundance of red blood cells trapped in a fibrin clot together with platelets, a structure quite distinct from the vast platelet aggregates found in arterial thrombi (Koupenova et al., 2017). Thus, arterial thrombosis is treated with therapies that target platelet activation and/or aggregation while VTE is traditionally treated with drugs targeting the coagulation system. Historically, platelet function has attracted attention primarily in arterial thrombosis, however more recently the role of platelets in VTE has been recognised (Montoro-Garcia et al., 2016). Elevated levels of markers of platelet activation, such as P-selectin, are associated with acute VTE (Jacobs et al., 2016); a protein we also identified as one of four candidates associated with VTE in the discovery screen of VEBIOS ER. Furthermore, anti-platelet therapy with acetylic salicylic acid had a protective effect against VTE (Simes et al., 2014), and reduced the size of venous thrombus linked to inhibition of platelet activation in mice (Tarantino et al., 2016)

6. The relevance of their COVID-19 findings is questionable. Several studies have shown that a large number of factors are associated with the degree of respiratory failure including TCC and in the present study the authors present unadjusted analyses. They discuss the importance of VTE in the pathogenesis of severe respiratory failure, but they present no data on either PA or VTE in the included patients. This part of the study should either be deleted or markedly improved.

We agree with the consensus between the reviewers that this section does not significantly add to the manuscript in its current form, and we have therefore removed it.

REFERENCES

- Antoni, G., Morange, P.E., Luo, Y., Saut, N., Burgos, G., Heath, S., Germain, M., Biron-Andreani, C., Schved, J.F., Pernod, G., *et al.* (2010). A multi-stage multi-design strategy provides strong evidence that the BAI3 locus is associated with early-onset venous thromboembolism. *J Thromb Haemost* 8, 2671-2679.
- Antoni, G., Oudot-Mellakh, T., Dimitromanolakis, A., Germain, M., Cohen, W., Wells, P., Lathrop, M., Gagnon, F., Morange, P.E., and Tregouet, D.A. (2011). Combined analysis of three genome-wide association studies on vWF and FVIII plasma levels. *BMC Med Genet* 12, 102.
- Ashburner, M., Ball, C.A., Blake, J.A., Botstein, D., Butler, H., Cherry, J.M., Davis, A.P., Dolinski, K., Dwight, S.S., Eppig, J.T., *et al.* (2000). Gene ontology: tool for the unification of biology. The Gene Ontology Consortium. *Nature genetics* 25, 25-29.
- Bruzelius, M., Iglesias, M.J., Hong, M.-G., Sanchez-Rivera, L., Gyorgy, B., Souto, J.C., Frånberg, M., Fredolini, C., Strawbridge, R.J., Holmström, M., *et al.* (2016a). PDGFB, a new candidate plasma biomarker for venous thromboembolism: results from the VEREMA affinity proteomics study. *Blood* 128, 59-67.
- Bruzelius, M., Iglesias, M.J., Hong, M.G., Sanchez-Rivera, L., Gyorgy, B., Souto, J.C., Franberg, M., Fredolini, C., Strawbridge, R.J., Holmstrom, M., *et al.* (2016b). PDGFB, a new candidate plasma biomarker for venous thromboembolism: results from the VEREMA affinity proteomics study. *Blood* 128, e59-e66.
- Butler, L.M., Hallstrom, B.M., Fagerberg, L., Ponten, F., Uhlen, M., Renne, T., and Odeberg, J. (2016). Analysis of Body-wide Unfractionated Tissue Data to Identify a Core Human Endothelial Transcriptome. *Cell Syst* 3, 287-301 e283.
- Chambers, M.C., Maclean, B., Burke, R., Amodei, D., Ruderman, D.L., Neumann, S., Gatto, L., Fischer, B., Pratt, B., Egertson, J., *et al.* (2012). A cross-platform toolkit for mass spectrometry and proteomics. *Nat Biotechnol* 30, 918-920.
- Chauhan, G., Adams, H.H.H., Bis, J.C., Weinstein, G., Yu, L., Töglhofer, A.M., Smith, A.V., van der Lee, S.J., Gottesman, R.F., Thomson, R., *et al.* (2015a). Association of Alzheimer's disease GWAS loci with MRI markers of brain aging. *Neurobiol Aging* 36, 1765.e1767-1765.e1716.
- Chauhan, G., Adams, H.H.H., Bis, J.C., Weinstein, G., Yu, L., Toglhofer, A.M., Smith, A.V., van der Lee, S.J., Gottesman, R.F., Thomson, R., *et al.* (2015b). Association of Alzheimer's disease GWAS loci with MRI markers of brain aging. *Neurobiol Aging* 36, 1765 e1767-1765 e1716.
- Chen, M.H., Raffield, L.M., Mousas, A., Sakaue, S., Huffman, J.E., Moscati, A., Trivedi, B., Jiang, T., Akbari, P., Vuckovic, D., *et al.* (2020). Trans-ethnic and Ancestry-Specific Blood-Cell Genetics in 746,667 Individuals from 5 Global Populations. *Cell* 182, 1198-1213.e1114.
- Chen, Q., Manzke, M., Hartmann, A., Buttner, M., Amann, K., Pauly, D., Wiesener, M., Skerka, C., and Zipfel, P.F. (2016). Complement Factor H-Related 5-Hybrid Proteins Anchor Properdin and Activate Complement at Self-Surfaces. *J Am Soc Nephrol* 27, 1413-1425.
- Cho, J. (2013). Protein disulfide isomerase in thrombosis and vascular inflammation. *J Thromb Haemost* 11, 2084-2091.
- Conlan, M.G., Folsom, A.R., Finch, A., Davis, C.E., Sorlie, P., Marcucci, G., and Wu, K.K. (1993). Associations of factor VIII and von Willebrand factor with age, race, sex, and risk factors for atherosclerosis. The Atherosclerosis Risk in Communities (ARIC) Study. *Thromb Haemost* 70, 380-385.
- Cox, J., Neuhauser, N., Michalski, A., Scheltema, R.A., Olsen, J.V., and Mann, M. (2011). Andromeda: a peptide search engine integrated into the MaxQuant environment. *J Proteome Res* 10, 1794-1805.
- Cserhalmi, M., Papp, A., Brandus, B., Uzonyi, B., and Jozsi, M. (2019). Regulation of regulators: Role of the complement factor H-related proteins. *Semin Immunol* 45, 101341.

- Douma, R.A., le Gal, G., Sohne, M., Righini, M., Kamphuisen, P.W., Perrier, A., Kruip, M.J., Bounameaux, H., Buller, H.R., and Roy, P.M. (2010). Potential of an age adjusted D-dimer cut-off value to improve the exclusion of pulmonary embolism in older patients: a retrospective analysis of three large cohorts. *BMJ* *340*, c1475.
- Eicher, J.D., Xue, L., Ben-Shlomo, Y., Beswick, A.D., and Johnson, A.D. (2016). Replication and hematological characterization of human platelet reactivity genetic associations in men from the Caerphilly Prospective Study (CaPS). *J Thromb Thrombolysis* *41*, 343-350.
- Ferkingstad, E., Sulem, P., Atlason, B.A., Sveinbjornsson, G., Magnusson, M.I., Styrnisdottir, E.L., Gunnarsdottir, K., Helgason, A., Oddsson, A., Halldorsson, B.V., *et al.* (2021). Large-scale integration of the plasma proteome with genetics and disease. *Nat Genet* *53*, 1712-1721.
- Fredolini, C., Bystrom, S., Sanchez-Rivera, L., Ioannou, M., Tamburro, D., Ponten, F., Branca, R.M., Nilsson, P., Lehtio, J., and Schwenk, J.M. (2019). Systematic assessment of antibody selectivity in plasma based on a resource of enrichment profiles. *Sci Rep* *9*, 8324.
- Fukuoka, Y., and Hugli, T.E. (1988). Demonstration of a specific C3a receptor on guinea pig platelets. *J Immunol* *140*, 3496-3501.
- Gene Ontology, C. (2021). The Gene Ontology resource: enriching a GOld mine. *Nucleic Acids Res* *49*, D325-D334.
- Germain, M., Chasman, D.I., de Haan, H., Tang, W., Lindstrom, S., Weng, L.C., de Andrade, M., de Visser, M.C., Wiggins, K.L., Suchon, P., *et al.* (2015). Meta-analysis of 65,734 individuals identifies TSPAN15 and SLC44A2 as two susceptibility loci for venous thromboembolism. *Am J Hum Genet* *96*, 532-542.
- Germain, M., Saut, N., Greliche, N., Dina, C., Lambert, J.C., Perret, C., Cohen, W., Oudot-Mellakh, T., Antoni, G., Alessi, M.C., *et al.* (2011). Genetics of venous thrombosis: insights from a new genome wide association study. *PLoS One* *6*, e25581.
- Gessulat, S., Schmidt, T., Zolg, D.P., Samaras, P., Schnatbaum, K., Zerweck, J., Knaute, T., Rechenberger, J., Delanghe, B., Huhmer, A., *et al.* (2019). Prosit: proteome-wide prediction of peptide tandem mass spectra by deep learning. *Nat Methods* *16*, 509-518.
- Geyer, P.E., Kulak, N.A., Pichler, G., Holdt, L.M., Teupser, D., and Mann, M. (2016). Plasma Proteome Profiling to Assess Human Health and Disease. *Cell Syst* *2*, 185-195.
- Goicoechea de Jorge, E., Caesar, J.J., Malik, T.H., Patel, M., Colledge, M., Johnson, S., Hakobyan, S., Morgan, B.P., Harris, C.L., Pickering, M.C., *et al.* (2013). Dimerization of complement factor H-related proteins modulates complement activation in vivo. *Proc Natl Acad Sci U S A* *110*, 4685-4690.
- Heemskerk, J.W., Mattheij, N.J., and Cosemans, J.M. (2013). Platelet-based coagulation: different populations, different functions. *J Thromb Haemost* *11*, 2-16.
- Iorio, A., Kearon, C., Filippucci, E., Marcucci, M., Macura, A., Pengo, V., Siragusa, S., and Palareti, G. (2010). Risk of recurrence after a first episode of symptomatic venous thromboembolism provoked by a transient risk factor: a systematic review. *Arch Intern Med* *170*, 1710-1716.
- Jacobs, B., Obi, A., and Wakefield, T. (2016). Diagnostic biomarkers in venous thromboembolic disease. *J Vasc Surg Venous Lymphat Disord* *4*, 508-517.
- Janse van Rensburg, W.J., and van der Merwe, P. (2017). Comparison of Commercially Available Blood Collection Tubes Containing Sodium Citrate and Hirudin in Platelet Aggregation Testing. *Med Sci Monit Basic Res* *23*, 264-269.
- Johnson, A.D., Yanek, L.R., Chen, M.H., Faraday, N., Larson, M.G., Tofler, G., Lin, S.J., Kraja, A.T., Province, M.A., Yang, Q., *et al.* (2010). Genome-wide meta-analyses identifies seven loci associated with platelet aggregation in response to agonists. *Nat Genet* *42*, 608-613.
- Kearon, C., Ageno, W., Cannegieter, S.C., Cosmi, B., Geersing, G.J., Kyrle, P.A., Subcommittees on Control of, A., Predictive, and Diagnostic Variables in Thrombotic, D. (2016). Categorization of patients as having provoked or unprovoked venous thromboembolism: guidance from the SSC of ISTH. *J Thromb Haemost* *14*, 1480-1483.

- Kline, J. (2020). Response by Kline to Letter Regarding Article, "Over-Testing for Suspected Pulmonary Embolism in American Emergency Departments: The Continuing Epidemic". *Circ Cardiovasc Qual Outcomes* 13, e006588.
- Koprulu, M.C.-Z., J.; Wheeler, E.; Lockhart, S.; Kerrison, N.D.; Wareham, N.J.; Pietzner, M.; Langenberg, C. (2022). From genome to phenome via the proteome: broad capture, antibody-based proteomics to explore disease mechanisms. www.medrxiv.org.
- Kotol, D., Hober, A., Strandberg, L., Svensson, A.S., Uhlen, M., and Edfors, F. (2021). Targeted proteomics analysis of plasma proteins using recombinant protein standards for addition only workflows. *Biotechniques* 71, 473-483.
- Koupenova, M., Kehrel, B.E., Corkrey, H.A., and Freedman, J.E. (2017). Thrombosis and platelets: an update. *Eur Heart J* 38, 785-791.
- Laurent, P.A., Séverin, S., Hechler, B., Vanhaesebroeck, B., Payrastre, B., and Gratacap, M.P. (2015). Platelet PI3K β and GSK3 regulate thrombus stability at a high shear rate. *Blood* 125, 881-888.
- Lindström, S., Brody, J.A., Turman, C., Germain, M., Bartz, T.M., Smith, E.N., Chen, M.H., Puurunen, M., Chasman, D., Hassler, J., *et al.* (2019). A large-scale exome array analysis of venous thromboembolism. *Genet Epidemiol* 43, 449-457.
- Llobet, D., Vallve, C., Tirado, I., Vilalta, N., Carrasco, M., Oliver, A., Mateo, J., Fontcuberta, J., and Souto, J.C. (2021). Platelet hyperaggregability and venous thrombosis risk: results from the RETROVE project. *Blood Coagul Fibrinolysis* 32, 122-131.
- Machiela, M.J., and Chanock, S.J. (2015). LDlink: a web-based application for exploring population-specific haplotype structure and linking correlated alleles of possible functional variants. *Bioinformatics* 31, 3555-3557.
- MANTEL, N., and HAENSZEL, W. (1959). Statistical aspects of the analysis of data from retrospective studies of disease. *J Natl Cancer Inst* 22, 719-748.
- Mastellos, D.C., Yancopoulou, D., Kokkinos, P., Huber-Lang, M., Hajishengallis, G., Biglarnia, A.R., Lupu, F., Nilsson, B., Risitano, A.M., Ricklin, D., *et al.* (2015). Compstatin: a C3-targeted complement inhibitor reaching its prime for bedside intervention. *Eur J Clin Invest* 45, 423-440.
- Matic, L.P., Jesus Iglesias, M., Vesterlund, M., Lengquist, M., Hong, M.G., Saieed, S., Sanchez-Rivera, L., Berg, M., Razuvaev, A., Kronqvist, M., *et al.* (2018). Novel Multiomics Profiling of Human Carotid Atherosclerotic Plaques and Plasma Reveals Biliverdin Reductase B as a Marker of Intraplaque Hemorrhage. *JACC Basic Transl Sci* 3, 464-480.
- McRae, J.L., Duthy, T.G., Griggs, K.M., Ormsby, R.J., Cowan, P.J., Cromer, B.A., McKinstry, W.J., Parker, M.W., Murphy, B.F., and Gordon, D.L. (2005). Human factor H-related protein 5 has cofactor activity, inhibits C3 convertase activity, binds heparin and C-reactive protein, and associates with lipoprotein. *J Immunol* 174, 6250-6256.
- Mittadodla, P.S., Kumar, S., Smith, E., Badireddy, M., Turki, M., and Fioravanti, G.T. (2013). CT pulmonary angiography: an over-utilized imaging modality in hospitalized patients with suspected pulmonary embolism. *J Community Hosp Intern Med Perspect* 3.
- Montoro-Garcia, S., Schindewolf, M., Stanford, S., Larsen, O.H., and Thiele, T. (2016). The Role of Platelets in Venous Thromboembolism. *Semin Thromb Hemost* 42, 242-251.
- Nalls, M.A., Couper, D.J., Tanaka, T., van Rooij, F.J., Chen, M.H., Smith, A.V., Toniolo, D., Zakai, N.A., Yang, Q., Greinacher, A., *et al.* (2011). Multiple loci are associated with white blood cell phenotypes. *PLoS Genet* 7, e1002113.
- Norgaard, I., Nielsen, S.F., and Nordestgaard, B.G. (2016). Complement C3 and High Risk of Venous Thromboembolism: 80517 Individuals from the Copenhagen General Population Study. *Clin Chem* 62, 525-534.
- Palareti, G., Cosmi, B., Legnani, C., Tosetto, A., Brusi, C., Iorio, A., Pengo, V., Ghirarduzzi, A., Pattacini, C., Testa, S., *et al.* (2006). D-dimer testing to determine the duration of anticoagulation therapy. *N Engl J Med* 355, 1780-1789.

- Panova-Noeva, M., Wagner, B., Nagler, M., Koeck, T., Ten Cate, V., Prochaska, J.H., Heitmeier, S., Meyer, I., Gerdes, C., Laux, V., *et al.* (2020). Comprehensive platelet phenotyping supports the role of platelets in the pathogenesis of acute venous thromboembolism - results from clinical observation studies. *EBioMedicine* *60*, 102978.
- Pietzner, M., Wheeler, E., Carrasco-Zanini, J., Cortes, A., Koprulu, M., Worheide, M.A., Oerton, E., Cook, J., Stewart, I.D., Kerrison, N.D., *et al.* (2021a). Mapping the proteo-genomic convergence of human diseases. *Science* *374*, eabj1541.
- Pietzner, M., Wheeler, E., Carrasco-Zanini, J., Kerrison, N.D., Oerton, E., Koprulu, M., Luan, J., Hingorani, A.D., Williams, S.A., Wareham, N.J., *et al.* (2021b). Synergistic insights into human health from aptamer- and antibody-based proteomic profiling. *Nat Commun* *12*, 6822.
- Pittens, C.A., Bouman, H.J., van Werkum, J.W., ten Berg, J.M., and Hackeng, C.M. (2009). Comparison between hirudin and citrate in monitoring the inhibitory effects of P2Y12 receptor antagonists with different platelet function tests. *J Thromb Haemost* *7*, 1929-1932.
- Polley, M.J., and Nachman, R.L. (1983). Human platelet activation by C3a and C3a des-arg. *J Exp Med* *158*, 603-615.
- Puurunen, M.K., Hwang, S.J., O'Donnell, C.J., Tofler, G., and Johnson, A.D. (2017). Platelet function as a risk factor for venous thromboembolism in the Framingham Heart Study. *Thromb Res* *151*, 57-62.
- Rappsilber, J., Ishihama, Y., and Mann, M. (2003). Stop and go extraction tips for matrix-assisted laser desorption/ionization, nanoelectrospray, and LC/MS sample pretreatment in proteomics. *Anal Chem* *75*, 663-670.
- Sauter, R.J., Sauter, M., Reis, E.S., Emschermann, F.N., Nording, H., Ebenhoch, S., Kraft, P., Munzer, P., Mauler, M., Rheinlaender, J., *et al.* (2018). Functional Relevance of the Anaphylatoxin Receptor C3aR for Platelet Function and Arterial Thrombus Formation Marks an Intersection Point Between Innate Immunity and Thrombosis. *Circulation* *138*, 1720-1735.
- Searle, B.C., Pino, L.K., Egertson, J.D., Ting, Y.S., Lawrence, R.T., MacLean, B.X., Villén, J., and MacCoss, M.J. (2018). Chromatogram libraries improve peptide detection and quantification by data independent acquisition mass spectrometry. *Nat Commun* *9*, 5128.
- Sennblad, B., Basu, S., Mazur, J., Suchon, P., Martinez-Perez, A., van Hylckama Vlieg, A., Truong, V., Li, Y., Gadin, J.R., Tang, W., *et al.* (2017). Genome-wide association study with additional genetic and post-transcriptional analyses reveals novel regulators of plasma factor XI levels. *Hum Mol Genet* *26*, 637-649.
- Silva, J.C., Gorenstein, M.V., Li, G.Z., Vissers, J.P., and Geromanos, S.J. (2006). Absolute quantification of proteins by LCMSE: a virtue of parallel MS acquisition. *Mol Cell Proteomics* *5*, 144-156.
- Simes, J., Becattini, C., Agnelli, G., Eikelboom, J.W., Kirby, A.C., Mister, R., Prandoni, P., Brighton, T.A., and Investigators, I.S. (2014). Aspirin for the prevention of recurrent venous thromboembolism: the INSPIRE collaboration. *Circulation* *130*, 1062-1071.
- Smith, N.L., Chen, M.H., Dehghan, A., Strachan, D.P., Basu, S., Soranzo, N., Hayward, C., Rudan, I., Sabater-Lleal, M., Bis, J.C., *et al.* (2010). Novel associations of multiple genetic loci with plasma levels of factor VII, factor VIII, and von Willebrand factor: The CHARGE (Cohorts for Heart and Aging Research in Genome Epidemiology) Consortium. *Circulation* *121*, 1382-1392.
- Sollis, E., Mosaku, A., Abid, A., Buniello, A., Cerezo, M., Gil, L., Groza, T., Gunes, O., Hall, P., Hayhurst, J., *et al.* (2022). The NHGRI-EBI GWAS Catalog: knowledgebase and deposition resource. *Nucleic Acids Res.*
- Tarantino, E., Amadio, P., Squellerio, I., Porro, B., Sandrini, L., Turnu, L., Cavalca, V., Tremoli, E., and Barbieri, S.S. (2016). Role of thromboxane-dependent platelet activation in venous thrombosis: Aspirin effects in mouse model. *Pharmacol Res* *107*, 415-425.
- Terraube, V., O'Donnell, J.S., and Jenkins, P.V. (2010). Factor VIII and von Willebrand factor interaction: biological, clinical and therapeutic importance. *Haemophilia* *16*, 3-13.

Thibord, F., Klarin, D., Brody, J.A., Chen, M.H., Levin, M.G., Chasman, D.I., Goode, E.L., Hveem, K., Teder-Laving, M., Martinez-Perez, A., *et al.* (2022a). Cross-Ancestry Investigation of Venous Thromboembolism Genomic Predictors. *Circulation* *146*, 1225-1242.

Thibord, F., Klarin, D., Brody, J.A., Chen, M.H., Levin, M.G., Chasman, D.I., Goode, E.L., Hveem, K., Teder-Laving, M., Martinez-Perez, A., *et al.* (2022b). Cross-Ancestry Investigation of Venous Thromboembolism Genomic Predictors. *Circulation* *146*, 1225-1242.

Tsai, A.W., Cushman, M., Rosamond, W.D., Heckbert, S.R., Tracy, R.P., Aleksic, N., and Folsom, A.R. (2002). Coagulation factors, inflammation markers, and venous thromboembolism: the longitudinal investigation of thromboembolism etiology (LITE). *Am J Med* *113*, 636-642.

Weber, M., Gerdson, F., Gutensohn, K., Schoder, V., Eifrig, B., and Hossfeld, D.K. (2002). Enhanced platelet aggregation with TRAP-6 and collagen in platelet aggregometry in patients with venous thromboembolism. *Thromb Res* *107*, 325-328.

Willer, C.J., Li, Y., and Abecasis, G.R. (2010). METAL: fast and efficient meta-analysis of genomewide association scans. *Bioinformatics* *26*, 2190-2191.

Zarabi, S., Chan, T.M., Mercuri, M., Kearon, C., Turcotte, M., Grusko, E., Barbic, D., Varner, C., Bridges, E., Houston, R., *et al.* (2021). Physician choices in pulmonary embolism testing. *CMAJ* *193*, E38-E46.

REVIEWERS' COMMENTS

Reviewer #1 (Remarks to the Author):

The authors have done an excellent job responding to all reviewers comments questions and suggestions. This is an improved manuscript.
I have no further suggestions.

Reviewer #2 (Remarks to the Author):

The revised version of the manuscript has improved markedly, and the authors provide an impressive amount of additional analyses and novel experimentation. In general, the authors have addressed the concerns raised in an adequate way.

I appreciate the authors' thorough work with including the appropriate controls for the platelet experiments. Their results are interesting and now appear more definitive. As the authors acknowledge, the exact mechanism by which CFHR5 potentiates platelet activation as well as the in vivo relevance remains to be elucidated.

Some comments on the interpretation of these results:

-CFHR5-induced platelet activation is blocked by Compstatin, a key experiment that demonstrates specificity and a complement dependent mechanism. The same results are achieved with a C3a-blocking antibody (clone K13/16), which is interesting given the conflicting results on expression of receptors for C3a in platelets. In the reviewer's hands, there is no effect whatsoever of purified C3a on platelets. The source of the clone K13/16 is not stated, but according to a number of suppliers (Sigma, ThermoFisher etc) this antibody reacts with intact C3 in addition to C3a and C3a-desArg, raising the possibility that it blocks activation of C3 rather than neutralizing C3a. Compstatin blocks C3 convertase formation and thus the entire complement cascade downstream from C3, so alternative mechanisms are possible.

-The results comparing citrate and hirudin as anticoagulants for the platelet studies are actually quite revealing. Hirudin is an extremely potent and specific inhibitor of thrombin, the coagulation factor that converts fibrinogen to fibrin. It is not known to inhibit any other protease, and therefore is used as an anticoagulant that does not interfere with complement. Yet, in hirudin-PRP the effect of CFHR5 disappears. In contrast, CFHR5 potentiates platelet activation in citrated PRP, but not in washed platelets. It is important to note that active thrombin is also an extremely potent platelet agonist. It might be that CHFR5 somehow promotes the low degree of thrombin generation that occurs when platelets are activated by agonists, which explains why no effect is observed in the presence of hirudin, or in washed platelets where there is no plasma as a source of thrombin. Citrate chelates calcium ions that are needed for both coagulation and complement, but is a less effective chelator compared to e.g. EDTA, and residual thrombin generation may occur. The lower background activation of platelets in hirudin-anticoagulated PRP in Fig. 5A for review also demonstrates this fact.

-It's a bit speculative to explain the absence of a potentiating effect of CFHR5 on platelet aggregation by "platelet activation that is independent of platelet aggregation". More likely, it reflects the limited sensitivity of light aggregometry, or an effect that is below the threshold to trigger aggregation. Also, in Fig. 4 & 5 for review, CFHR5 promotes activation of GPIIb/IIIa (as measured by flow for PAC-1 binding), which is the primary receptor mediating platelet aggregation.

Reviewer #3 (Remarks to the Author):

The authors have undertaken considerable additional analyses and addressed my concerns with the original manuscript.

Reviewer #4 (Remarks to the Author):

The authors have adequately addressed my concern and manuscript has in general been significantly improve

I have only one minor point. They present data indicating that rCFHR5 had no procoagulant effect on endothelial cells. This observation as a negative finding is of interest and could be presented in 1-2 sentences with references to Supplemental file.

Reviewer #1 (Remarks to the Author):

The authors have done an excellent job responding to all reviewers comments questions and suggestions. This is an improved manuscript.

I have no further suggestions.

Reviewer #2 (Remarks to the Author):

The revised version of the manuscript has improved markedly, and the authors provide an impressive amount of additional analyses and novel experimentation. In general, the authors have addressed the concerns raised in an adequate way.

I appreciate the authors' thorough work with including the appropriate controls for the platelet experiments. Their results are interesting and now appear more definitive. As the authors acknowledge, the exact mechanism by which CFHR5 potentiates platelet activation as well as the in vivo relevance remains to be elucidated.

Some comments on the interpretation of these results:

-CFHR5-induced platelet activation is blocked by Compstatin, a key experiment that demonstrates specificity and a complement dependent mechanism. The same results are achieved with a C3a-blocking antibody (clone K13/16), which is interesting given the conflicting results on expression of receptors for C3a in platelets. In the reviewer's hands, there is no effect whatsoever of purified C3a on platelets. The source of the clone K13/16 is not stated, but according to a number of suppliers (Sigma, ThermoFisher etc) this antibody reacts with intact C3 in addition to C3a and C3a-desArg, raising the possibility that it blocks activation of C3 rather than neutralizing C3a. Compstatin blocks C3 convertase formation and thus the entire complement cascade downstream from C3, so alternative mechanisms are possible.

We agree with the reviewer that, as the antibody targets also intact C3, alternative mechanisms are possible. We have added the following text to discussion [line 577-580]

'As the anti-C3a monoclonal antibody used detects an epitope also present on intact C3, the observed effect could be due to the antibody blocking of C3 activation rather than neutralisation of C3a, thus alternative mechanisms are possible'.

-The results comparing citrate and hirudin as anticoagulants for the platelet studies are actually quite revealing. Hirudin is an extremely potent and specific inhibitor of thrombin, the coagulation factor that converts fibrinogen to fibrin. It is not known to inhibit any other protease, and therefore is used as an anticoagulant that does not interfere with complement. Yet, in hirudin-PRP the effect of CFHR5 disappears. In contrast, CFHR5 potentiates platelet activation in citrated PRP, but not in washed platelets. It is important to note that active thrombin is also an extremely potent platelet agonist. It might be that CHFR5 somehow promotes the low degree of thrombin generation that occurs when platelets are activated by agonists, which explains why no effect is observed in the presence of hirudin, or in washed

platelets where there is no plasma as a source of thrombin. Citrate chelates calcium ions that are needed for both coagulation and complement, but is a less effective chelator compared to e.g. EDTA, and residual thrombin generation may occur. The lower background activation of platelets in hirudin-anticoagulated PRP in Fig. 5A for review also demonstrates this fact.

R: We thank the reviewer for very helpful comments on the interpretation of our findings. In view of these, we decided to include the results from platelet activation in hirudin plasma (previously only in response letter), into the results section, [line 434-440].

'In PRP anticoagulated with hirudin, baseline detection of activated GP IIb/IIIa (PAC1+) on unstimulated and ADP stimulated platelets was significantly lower compared to when citrate was used as anticoagulant (mean % expression ± std dev. [unstimulated: citrate 9.6±4.8 vs. hirudin 1.6±1.2] and [ADP-stimulated: citrate 30±15.0 vs. hirudin 7.5±3.2]. We did not observe any effect of recombinant CFHR5 on activated GP IIb/IIIa (PAC1+) on platelets from hirudin anticoagulated blood following ADP stimulation (Figure S5d-e).

New supplementary Figure S5 d-e, with legend:

d) Baseline or ADP-induced platelet activated GP IIb/IIIa (PAC1+) expression (%) was measured on platelets in plasma isolated from (i) citrate- or (ii) hirudin- anticoagulated blood, following preincubation with either PBS control or recombinant CFHR5. The same data, normalized to each respective ADP-stimulated control, is presented in (e). US: unstimulated, control (PBS). Each experiment is represented by an individual point and paired experiments connected by a dotted line. (5a-e) ANOVA two-sided test were performed. *p<0.05 **p<0.01 ***p<0.001

We have amended the discussion [line 580-590]

'In PRP anticoagulated with hirudin the co-stimulatory effect of CFHR5 observed in PRP anticoagulated with citrate does not occur, in line with studies describing an inhibitory effect of hirudin on platelet activation [75]. Hirudin acts through an irreversible strong specific inhibition of thrombin, which is also is an extremely potent platelet activation agonist. Citrate, by chelating calcium ions, reduce the activity of several enzymes in the coagulation system and parts of the complement system (e.g. classical pathway), however not as efficiently as EDTA, and residual thrombin generation can occur, as demonstrated by the difference in background platelet activation observed between hirudin PRP and citrate PRP. One could speculate that CFHR5 promotes the low degree of thrombin generation that occurs when platelets are activated by agonists which would explain why no effect is observed in the presence of hirudin, or in washed platelets where there is no source of thrombin.'

-It's a bit speculative to explain the absence of a potentiating effect of CFHR5 on platelet aggregation by "platelet activation that is independent of platelet aggregation". More likely, it reflects the limited sensitivity of light aggregometry, or an effect that is below the threshold to trigger aggregation. Also, in Fig. 4 & 5 for review, CFHR5 promotes activation of GPIIb/IIIa (as measured by flow for PAC-1 binding), which is the primary receptor mediating platelet aggregation.

R: We agree with the reviewer that it is not unlikely that the results reflect limitation in the method used, and are now pointing this out in the discussion [line 597-600, blue text]

Thus, one could speculate that CFHR5 has a role in VTE-linked platelet activation that is independent of platelet aggregation, however the lack of an observable potentiating effect of CFHR5 on platelet aggregation could have methodological explanations (e.g. reflecting the limited sensitivity of light aggregometry), since we found CFHR5 to enhance activation of GPIIb/IIIa, the primary receptor mediating platelet aggregation.

Reviewer #3 (Remarks to the Author):

The authors have undertaken considerable additional analyses and addressed my concerns with the original manuscript.

Reviewer #4 (Remarks to the Author):

The authors have adequately addressed my concern and manuscript has in general been significantly improve.

I have only one minor point. They present data indicating that rCFHR5 had no procoagulant effect on endothelial cells. This observation as a negative finding is of interest and could be presented in 1-2 sentences with references to Supplemental file.

R: As the reviewer suggest, we have now included these negative findings in the manuscript, adding a paragraph in the results section [line 460-466]:

CFHR5 does not induce a procoagulant response to inflammation in endothelial cells
To investigate a potential effect of CFHR5 on endothelial cells, we treated primary human umbilical vein endothelial cells (HUVECs) with recombinant CFHR5 under unstimulated and TNF stimulated conditions and assessed the effect on coagulation and inflammation, using thrombin generation assay (TGA) and measurement of mRNA expression of several markers (F3, IL8, vWF, THBD, TFPI, PLAT, ICAM1). No differences were observed in the presence of rCFHR5 compared with buffer (PBS), under any of the conditions (Figure S7).

New supplementary Figure S7:

Figure S7

Figure S7: Effect of CFHR5 on HUVEC coagulability and inflammatory response. HUVEC were stimulated with rCFHR5 (6 µg/ml), the inflammatory cytokine tumour necrosis factor alpha (TNF, Sigma) (10 ng/ml), both or neither (PBS controls), for 24h before measurement of: **(a)** thrombin formation, assessed by real-time thrombin formation assay (n=2) and parameters of thrombin generation (Peak, Time to peak and Lag time) determined by the thrombinoscope software, or **(b)** relative gene expression of F3, THBD, PLAT, TFPI, vWF, ICAM1 and CXCL8, measured by real-time quantitative PCR (n=2). Data presented individual independent experiments with mean values. Source data are provided as a Source Data file (tab: Fig S7). F3: Tissue factor, THBD: thrombomodulin, PLAT: plasminogen

activator, tissue type, TFPI: tissue factor pathway inhibitor, vWF: von Willebrand Factor, ICAM1: intracellular adhesion molecule 1, CXCL8: interleukin 8.

New methods section (line 1073-1092):

Effect of CFHR5 on HUVECs. Human umbilical vein endothelial cells (HUVEC) were isolated from 2 anonymised umbilical cords collected from Karolinska University Hospital. Ethical approval for HUVEC isolation and subsequent experimentation was granted by Regional ethics committee in Stockholm (DNR 2015/1294-31/2). HUVEC were cultivated in M199 (M199, Gibco) supplemented with 20% foetal bovine serum (FBS), 100 U/ml penicillin, 0.1 mg/ml streptomycin, 1 µg/ml Hydrocortisone, 1 ng/ml Human Epidermal Growth Factor (all Sigma), and 1.25 µg/ml Amphotericin B (Invitrogen) and seeded on flat-bottom 96 well plates (Falcon, C-treated culture plate). HUVEC were stimulated with rCFHR5 (6 µg/ml), the inflammatory cytokine tumour necrosis factor alpha (TNF, Sigma) (10 ng/ml), both or neither (PBS controls), for 24h before measurement of thrombin generation or analysis of selected gene expression. HUVEC were blocked with 3% BSA and thrombin formation was initiated in 120 µL reaction mixtures containing human citrated plasma (George King), 4 µM phospholipids, 16.6 mM Ca²⁺ and 2.5 mM fluorogenic substrate (Z-Gly-Gly-Arg-AMC) (Thrombinoscope BV, Diagnostica Stago). All real time thrombin formation assays were run in duplicate. Thrombin generation was quantified using the Thrombinoscope software package (Version 3.0.0.29). For relative gene expression, cDNA was prepared using TaqMan Gene Expression Cells-to-Ct Kit (Ambion), and qPCR performed using Taqman Fast Universal PCR Master Mix and 18 s rRNA reference primer (4319413E), with target primers for ICAM1 (Hs00164932), CXCL8 (Hs00174103), F3 (Hs01076029), vWF (Hs01109446), THBD (Hs00264920), TFPI (Hs00196731) and PLAT (Hs00263492) using a StepOnePlus Real-Time PCR System (all Applied Biosystems).